# Deep Generative Models through the Lens of the Manifold Hypothesis: A Survey and New Connections

**Gabriel Loaiza-Ganem**                                                                 *gabriel@layer6.ai*
*Layer 6 AI*

**Brendan Leigh Ross**                                                                  *brendan@layer6.ai*
*Layer 6 AI*

**Rasa Hosseinzadeh**                                                                     *rasa@layer6.ai*
*Layer 6 AI*

**Anthony L. Caterini**                                                                 *anthony@layer6.ai*
*Layer 6 AI*

**Jesse C. Cresswell**                                                                   *jesse@layer6.ai*
*Layer 6 AI*

**Reviewed on OpenReview:** *https://openreview.net/forum?id=a90WpmSi0I*

## Abstract

In recent years there has been increased interest in understanding the interplay between deep generative models (DGMs) and the manifold hypothesis. Research in this area focuses on understanding the reasons why commonly-used DGMs succeed or fail at learning distributions supported on unknown low-dimensional manifolds, as well as developing new models explicitly designed to account for manifold-supported data. This manifold lens provides both clarity as to why some DGMs (e.g. diffusion models and some generative adversarial networks) empirically surpass others (e.g. likelihood-based models such as variational autoencoders, normalizing flows, or energy-based models) at sample generation, and guidance for devising more performant DGMs. We carry out the first survey of DGMs viewed through this lens, making two novel contributions along the way. First, we formally establish that numerical instability of likelihoods in high ambient dimensions is unavoidable when modelling data with low intrinsic dimension. We then show that DGMs on learned representations of autoencoders can be interpreted as approximately minimizing Wasserstein distance: this result, which applies to latent diffusion models, helps justify their outstanding empirical results. The manifold lens provides a rich perspective from which to understand DGMs, and we aim to make this perspective more accessible and widespread.

# Contents

# 1 Introduction

Learning the distribution that gave rise to observed data in $\mathbb{R}^D$ has long been a central problem in statistics and machine learning (Lehmann & Casella, 2006; Murphy, 2012). In the deep learning era, there has been a tremendous amount of research aimed at leveraging neural networks to solve this task, bringing forth *deep generative models* (DGMs). This effort has paid off, with state-of-the-art approaches such as diffusion models (Sohl-Dickstein et al., 2015; Ho et al., 2020; Song et al., 2021b) and their latent variants (Rombach et al., 2022) achieving remarkable empirical success (Ramesh et al., 2022; Nichol et al., 2022; Saharia et al., 2022). A natural question is why these latest models outperform previous DGMs, or more generally:

*What makes a deep generative model good?*

To answer this fundamental question, a line of research has emerged with the goal of understanding DGMs through their relationship with the *manifold hypothesis*, and using the obtained insights to drive empirical improvements. In its simplest form, this crucial hypothesis states that high-dimensional data of interest often lies on an unknown $d^*$-dimensional submanifold $\mathcal{M}$ of $\mathbb{R}^D$, with $d^* < D$. Studying the behaviour of DGMs when the underlying data-generating distribution is supported on an unknown low-dimensional submanifold of $\mathbb{R}^D$ has already proven fruitful: for example, it is precisely those models with the capacity to learn low-dimensional manifolds (such as diffusion models, latent or otherwise) that tend to work better in practice. Similarly, various pathologies of DGMs – such as mode collapse in variational autoencoders (Kingma & Welling, 2014; Rezende et al., 2014; Dai & Wipf, 2019), and numerical instabilities in the score function of diffusion models (Pidstrigach, 2022; Lu et al., 2023) or in normalizing flows (Dinh et al., 2015; 2017; Cornish et al., 2020; Behrmann et al., 2021), among others – can be explained as a failure to properly account for manifold structure within data. Despite the existence of various DGM review papers (Kobyzev et al., 2020; Papamakarios et al., 2021; Bond-Taylor et al., 2022; Yang et al., 2023), to the best of our knowledge none takes this "manifold lens". Here, we present the first survey of DGMs from this viewpoint.

The relevance and usefulness of studying DGMs through this lens hinges on a critical assumption: namely, that the manifold hypothesis actually holds. It is thus relevant to justify this hypothesis. There is a plethora of arguments supporting the existence of low-dimensional structure in most high-dimensional data of interest, which we summarize below:

- **Intuition** Informally, saying that a subset $\mathcal{M}$ of $\mathbb{R}^D$ is a low-dimensional submanifold is a mathematical way of capturing two important properties: a sense of sparsity in ambient space ($\mathcal{M}$ has volume, or Lebesgue measure, 0 in $\mathbb{R}^D$), and a notion of smoothness. These are properties that one can commonly expect of high-dimensional data of interest. For example, consider natural images in $[0,1]^D$ (assume they have been scaled to this range), where $D$ corresponds to the number of pixels. Imagine sampling uniformly from $[0,1]^D$ until a human deems the resulting sample to be a natural image. For all intents and purposes, such an image would never be sampled since natural images are "sparse" in their ambient space. Images are also "smooth" in that, given a natural image, one can always conceive of slightly deforming it in such a way that the result remains a natural image. One can also intuitively understand a $d^*$-dimensional submanifold $\mathcal{M}$ of $\mathbb{R}^D$ as a (smooth in some way) subset of *intrinsic dimension* $d^*$, which can be thought of as the number of factors of variation needed to characterize a point in $\mathcal{M}$. Again, most machine learning researchers or practitioners who have worked with natural images will have the tacit understanding that far fewer dimensions than the *ambient dimension*, $D$, are needed to describe an image: for example, images can be successfully synthesized from low-dimensional latent variables (Rombach et al., 2022; Sauer et al., 2023), and they can be effectively compressed without affecting how humans perceive them (Wallace, 1992; Townsend et al., 2019; Ruan et al., 2021). Through this line of thinking, the manifold hypothesis has been a motivating concept in the field of machine learning from its infancy. Some of the first autoencoders (Kramer, 1991) aimed to account for data with low intrinsic dimension. Early unsupervised algorithms such as Boltzmann machines (Ackley et al., 1985) were conceived as ways of learning the constraints that govern complex data distributions. Indeed, the manifold hypothesis is one of the core intuitions behind why neural networks are so successful at learning low-dimensional representations in the first place (Bengio et al., 2013).

- **Theory**  The manifold hypothesis helps explain the success of deep learning through more than just intuition. For example, under standard assumptions, the sample complexity of kernel density estimation in $\mathbb{R}^D$ is exponential in $D$ (Wand & Jones, 1994). This well-known result is a manifestation of the curse of dimensionality, and highlights the fundamental hardness of learning arbitrary high-dimensional distributions. Yet DGMs succeed at this task, suggesting that these standard assumptions are too loose and do not take into account relevant structure present in data of interest. Making the assumption that the data lies on a submanifold of $\mathbb{R}^D$ is a sensible way of incorporating more structure, and is consistent with theory: the sample complexity of kernel density estimation actually scales exponentially with *intrinsic* dimensionality (Ozakin & Gray, 2009; Berenfeld & Hoffmann, 2021) – even if the data is concentrated around a manifold rather than exactly on one (Divol, 2022). These results suggest that the task of learning distributions when $d^*$ is much smaller than $D$ is fundamentally more tractable than when there is no low-dimensional submanifold (i.e. $d^* = D$). This observation is not unique to density estimation: the difficulty of classification and manifold learning are also known to scale with intrinsic – rather than ambient – dimension (Narayanan & Niyogi, 2009; Narayanan & Mitter, 2010), making them effectively impossible for high-dimensional data without additional structure. These results are a sign that manifold structure in data is the reason why deep learning manages to avoid the curse of dimensionality. The triumph of modern algorithms on these tasks thus provides strong implicit justification for the manifold hypothesis.

- **Empiricism**  There are various works which use existing intrinsic dimension estimators (Levina & Bickel, 2004; MacKay & Ghahramani, 2005; Johnsson et al., 2014; Facco et al., 2017; Bac et al., 2021), or develop their own, and apply them on commonly-used image datasets (Pope et al., 2021; Tempczyk et al., 2022; Zheng et al., 2022; Brown et al., 2023). These works unanimously estimate the intrinsic dimension of images to be orders of magnitude smaller than their ambient dimension, and similar studies have been carried out on physics datasets with analogous findings (Cresswell et al., 2022). All these works provide explicit evidence supporting the manifold hypothesis.

Our goal in this survey is to present an accessible, yet mathematically precise, view of DGMs from the perspective of the manifold hypothesis. First, Section 2 characterizes the setup we will consider throughout, and provides a consistent set of notation and terminology with which to describe DGMs. Section 3 lays out relevant background, covering manifold learning and divergences between probability distributions – with a special focus on which ones provide an adequate minimization objective when manifolds are involved. Section 4 describes popular *manifold-unaware* DGMs – i.e. models which do not account for the manifold hypothesis – and in Section 4.1.1 we provide our first novel result, showing that high-dimensional likelihood-based models are unavoidably bound to suffer numerical instability when the manifold hypothesis holds. Section 5 covers *manifold-aware* DGMs – i.e. models which do account for the manifold hypothesis. We include both popular DGMs which happen to be manifold-aware and models which were explicitly designed to account for manifold structure. In Section 5.3.1 we provide a new perspective on *two-step models*, one of the predominant paradigms of manifold-aware DGMs: we show that, in addition to their common interpretation as jointly learning a manifold and a distribution, they minimize a (potentially regularized) upper bound – which can become tight at optimality – of the Wasserstein distance against the true data-generating distribution. In Section 6 we cover discrete DGMs, before concluding and discussing directions for future research in Section 7.

## 2  Notation and Setup

In this section we present the notation and setup that we will use throughout our work. We try to deviate as little as possible from standard notation, but we nonetheless prioritize precision and consistency across models. While knowledge of measure theory and differential geometry is needed to understand some technical details of how DGMs relate to manifolds, the core ideas, methods, and intuitions of this area do not require mathematics beyond what is typically known by machine learning researchers. Our intention in this survey is thus to remain accessible to readers with no background in these topics.

> **Advanced topics** In the interest of mathematically-inclined readers, we use grey boxes like this one throughout the manuscript to present content which does require familiarity with topics such as measure theory (Billingsley, 2012), topology (Munkres, 2014), or differential geometry (Lee, 2012; 2018). Providing a primer covering all the relevant material from these topics would be prohibitively lengthy and thus falls outside the scope of our work. Nonetheless, we do include a short summary of weak convergence of probability measures in Appendix A, as this is a particularly important tool from measure theory allowing us to formalize the intuition that a model learns its target distribution throughout training – even when this target is supported on a low-dimensional manifold. The material within these grey boxes is self-contained and is not necessary to understand the rest of our survey.

## 2.1 Notation

**Ambient and latent spaces** We denote the $D$-dimensional ambient space as $\mathcal{X}$, where depending on the particular model being discussed, $\mathcal{X}$ could be $\mathbb{R}^D$ or $[0,1]^D$. Many models use a latent space, which we denote as $\mathcal{Z}$, where $\mathcal{Z} = \mathbb{R}^d$. In most cases the latent space is low-dimensional, i.e. $d < D$, but in some instances this need not hold. We use lower-case letters $x \in \mathcal{X}$ and $z \in \mathcal{Z}$ to denote points, and upper-case letters $X \in \mathcal{X}$ and $Z \in \mathcal{Z}$ for random variables, on these respective spaces.

**Encoders and decoders** It will often be the case that we consider an encoder and a decoder between the aforementioned spaces, which we denote as $f : \mathcal{X} \to \mathcal{Z}$ and $g : \mathcal{Z} \to \mathcal{X}$, respectively.

**Probability** We use the letters $p$ and $q$ to denote densities; an exception being the Gaussian density, which we denote as $\mathcal{N}(x; \mu, \Sigma)$ when evaluated at $x$, where $\mu$ and $\Sigma$ correspond to its mean and covariance matrix, respectively. We write $X \sim p$ to indicate that $X$ is distributed according to $p$. Since we will often need various densities, we use superindices to identify them,[1] e.g. $p^X$ and $p^Z$ will denote densities on $\mathcal{X}$ and $\mathcal{Z}$, respectively. In particular, we write the true data-generating density on $\mathcal{X}$ as $p_*^X$. For a space $\mathcal{S}$ (e.g. $\mathcal{Z}$ or $\mathcal{X}$), we denote the set of all probability distributions on $\mathcal{S}$ as $\Delta(\mathcal{S})$. We use $p \circledast q$ to denote the convolution between the densities $p$ and $q$, i.e. if $X_1 \sim p$ and $X_2 \sim q$ are independent, $p \circledast q$ is the density of $X_1 + X_2$. We denote expectations with $\mathbb{E}$, and use a subindex to specify what density the expectation is taken with respect to, e.g. $\mathbb{E}_{X \sim p_*^X}[\cdot]$.

**Network parameters** All DGMs leverage neural networks, in one way or another, to define a density $p_\theta^X$ parameterized by the learnable parameters $\theta$ of the neural network(s). We will often abuse notation and use $\theta$ to parameterize all the generative components of the DGM; e.g. some models involve a decoder $g_\theta$ and a prior $p_\theta^Z$ on $\mathcal{Z}$, and we frequently subindex both components with $\theta$ even if they do not share parameters. In some instances it will be necessary to distinguish between generative parameters, in which case we will explicitly write $\theta = (\theta_1, \theta_2)$, using e.g. $g_{\theta_1}$ and $p_{\theta_2}^Z$ instead of $g_\theta$ and $p_\theta^Z$, respectively. We will sometimes overload notation by using subindices to denote a sequence of model parameters $(\theta_t)_{t=1}^\infty$; the meaning of parameter subindices will always be clear from context. We use $\phi$ for all auxiliary parameters; i.e. those that are not needed for generation. For example, an encoder $f_\phi$ could have been learned alongside $p_\theta^Z$ and $g_\theta$, but it might not be needed to sample from the model. We use $\theta^*$ and $\phi^*$ to denote optimal values of these parameters (with respect to the loss being optimized for the particular model being discussed).

**Calculus** We denote derivatives (gradients/Jacobians) with $\nabla$, and use a subindex to indicate which variable the differentiation is with respect to. For example, $\nabla_x f_\phi(x)$ and $\nabla_z g_\theta(z)$ respectively denote the Jacobians of the encoder and decoder with respect to their inputs (not their parameters), evaluated at $x$ and $z$; and $\nabla_\theta \log p_\theta^X(x)$ denotes, for a given $x$, the gradient of the log density of the model with respect to its parameters, evaluated at $\theta$.

**Linear algebra** We denote the identity matrix as $I$, with a corresponding subindex to indicate dimension. For example, $I_D$ corresponds to the $D \times D$ identity matrix. For a matrix $J$, we use $\det J$, $\operatorname{tr} J$, and $J^\top$

---

[1]That is, we do not use the common overloading of notation where $p(x)$ and $p(z)$ refer to different densities: in our notation these would correspond to the same density evaluated at two different points.

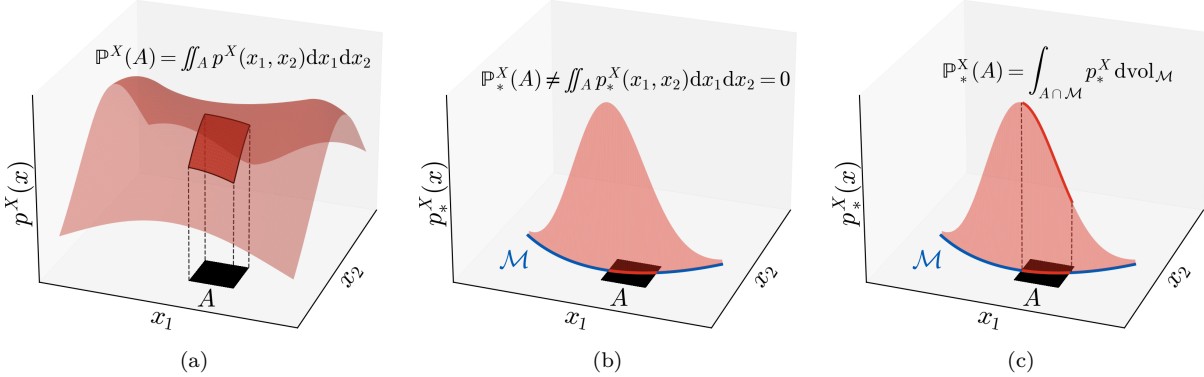

Figure 1: **(a)** Depiction of a full-dimensional density (i.e. a density in the "usual sense") $p^X$ on $\mathbb{R}^D$, with $D = 2$. The probability assigned by $p^X$ to a region $A$ of $\mathbb{R}^D$ is its integral over the region, i.e. $\iint_A p^X(x_1, x_2)\mathrm{d}x_1\mathrm{d}x_2$. **(b)** When the density $p_*^X$ is instead supported on a $d^*$-dimensional manifold $\mathcal{M}$ embedded in $\mathbb{R}^D$ (here, $d^* = 1$ and $\mathcal{M}$ is a curve), the integral evaluates to zero, and thus $p_*^X$ is not a density in the "usual sense". **(c)** Formally, in order to recover the probability assigned to $A$ by the manifold-supported density $p_*^X$, the density must be integrated only over $\mathcal{M}$ using a volume form $(\mathrm{dvol}_{\mathcal{M}})$ on the manifold, $\int_{A \cap \mathcal{M}} p_*^X \mathrm{dvol}_{\mathcal{M}}$ – which in this case simply corresponds to a line integral.

to denote its determinant, trace, and transpose, respectively. We denote the $\ell_1$ and $\ell_2$ norms in Euclidean space as $\|\cdot\|_1$ and $\|\cdot\|_2$, respectively.

> **Formal notation** In order to formally discuss probability distributions on manifolds, we need the language of measure theory. All the measures we will consider are defined over $\mathcal{X}$ or $\mathcal{Z}$, with their respective Borel $\sigma$-algebras. We remind the reader that, since they are Euclidean, $\mathcal{X}$ and $\mathcal{Z}$ are not arbitrary probability spaces. For a subset $A$ of $\mathcal{X}$, we denote its closure in $\mathcal{X}$ as $\mathrm{cl}_{\mathcal{X}}(A)$. We denote measures with the same letters as densities, but with uppercase blackboard style, i.e. $\mathbb{P}$ and $\mathbb{Q}$ – two exceptions being the Lebesgue and Gaussian measures, which we denote as $\lambda$ and $\mathcal{N}(\mu, \Sigma)$, respectively. We use a subindex on $\lambda$ to indicate its ambient dimension; e.g. $\lambda_D$ denotes the $D$-dimensional Lebesgue measure. We write $\mathbb{P} \ll \mathbb{Q}$ to indicate that $\mathbb{P}$ is absolutely continuous with respect to $\mathbb{Q}$. We use $\#$ as a subscript to denote pushforward measures; e.g. for a measurable function $g : \mathcal{Z} \to \mathcal{X}$ and a probability measure $\mathbb{P}^Z$ on $\mathcal{Z}$, $g_\# \mathbb{P}^Z$ is the pushforward probability measure (on $\mathcal{X}$) of $\mathbb{P}^Z$ through $g$. We will write the true data-generating distribution corresponding to $p_*^X$ as $\mathbb{P}_*^X$, and the model distribution corresponding to $p_\theta^X$ as $\mathbb{P}_\theta^X$. Finally, we use $\xrightarrow{\omega}$ to indicate weak convergence of probability measures.

## 2.2 Setup

The main goal of all the models $p_\theta^X$ considered in this paper is to learn $p_*^X$. Two main assumptions, which most of the works presented throughout our survey follow, are commonly made when attempting to understand DGMs through the manifold lens:

- **Manifold support** As mentioned in the introduction, there is a substantial body of work supporting the existence of low-dimensional structure in high-dimensional data of interest such as images. One way of mathematically expressing this structure is by taking a literal interpretation of the manifold hypothesis; that is, assuming that $p_*^X$ is supported on a $d^*$-dimensional submanifold $\mathcal{M}$ of $\mathcal{X}$, where $0 < d^* < D$ and both $\mathcal{M}$ and $d^*$ are unknown. Several aspects of this assumption warrant additional discussion. (*i*) $p_*^X$ being manifold-supported implies that it is not a full-dimensional density (i.e. with respect to the $D$-dimensional Lebesgue measure) and it is thus not a density in the "usual sense". See Figure 1 for an explanation. (*ii*) This assumption is a choice about how to

represent low-dimensional structure, and can be relaxed in various ways. For example, one could instead assume that the support of $p_*^X$ has varying intrinsic dimension (Brown et al., 2023), that it has singularities (Von Rohrscheidt & Rieck, 2023; Wang & Wang, 2024), or that $p_*^X$ simply concentrates most of its mass around a manifold – rather than all of it (Divol, 2022; Berenfeld et al., 2024). ($iii$) The assumption that $p_*^X$ is supported exactly on a manifold remains nonetheless very useful, even if we believe it to be "slightly off"; it serves as a first step towards understanding the interplay between DGMs and the low-dimensional structure of the data they are trained on, and as we will see, insights arising from this assumption explain various empirical observations. ($iv$) The requirement that $d^* > 0$ simply rules out $p_*^X$ being a probability mass function, which are better modelled with discrete DGMs. Our main focus is thus on continuous models, although we discuss their discrete counterparts in Section 6.

- **Nonparametric regime**   We use the term *nonparametric regime* to refer to the setting where we assume access to an infinite amount of data, arbitrarily flexible models, and exact optimization. More specifically, training any of the DGMs that we will consider requires minimizing a loss that depends on model parameters and which involves an expectation with respect to $p_*^X$; doing so is challenging for various reasons. ($i$) In practice, expectations with respect to $p_*^X$ cannot be computed, and are thus approximated through empirical averages over the dataset at hand. Writing the losses using expectations with respect to $p_*^X$ is thus assuming access to infinite data. ($ii$) Attempting to reason about optimal parameter values for any given neural network architecture quickly becomes essentially impossible in all but trivial cases. One way to circumvent this issue is to assume that all the neural networks involved, or any density model $p_\theta^X$ used, can represent any continuous function, or continuous density, respectively. This assumption of arbitrary flexibility is of course motivated by universal approximation properties of neural networks (Hornik, 1991; Koehler et al., 2021; Puthawala et al., 2022). ($iii$) Stochastic gradient-based optimization (Robbins & Monro, 1951) over mini-batches of the non-convex loss is used in practice (Kingma & Ba, 2015), which is not guaranteed to recover a global optimum. Once again to facilitate analysis, it is convenient to assume that all the optimization problems can be solved exactly.

Overall, even though these assumptions are optimistic, they remain popular as they allow for mathematical analysis of DGMs. The nonparametric regime also provides a necessary condition for DGMs to learn $p_*^X$ in practice; if a model fails to capture $p_*^X$ even in this idealistic regime – which as we will see happens surprisingly often for commonly-used DGMs – then there is no hope that the model can empirically recover $p_*^X$ in a realistic setting.

---

**Formal setup**   Throughout our work, $\mathcal{M}$ will be a $d^*$-dimensional embedded submanifold of $\mathbb{R}^D$. We previously mentioned that we will assume $\mathbb{P}_*^X$ is supported on $\mathcal{M}$, and that it admits a density $p_*^X$; in this grey box we formalize the meaning of these statements, the latter of which can be formalized either through measure theory or differential geometry (although we will not require this formal understanding of $p_*^X$, we nonetheless include it for completeness):

- **Formalizing manifold support**   Intuitively, the support of a distribution is the "smallest set of probability 1". To formally capture this intuition, the support $\mathrm{supp}(\mathbb{P})$ of a distribution $\mathbb{P}$ on $\mathcal{X}$ is defined as (Bogachev, 2007, Section 2 of Chapter 7, page 77)

$$\mathrm{supp}(\mathbb{P}) := \bigcap_{C \in \mathcal{C}(\mathbb{P})} C, \tag{1}$$

  where $\mathcal{C}(\mathbb{P})$ is the collection of closed (in $\mathcal{X}$) sets $C$ such that $\mathbb{P}(C) = 1$. It immediately follows that the support of a distribution is always a closed set in its ambient space. In general, $\mathcal{M}$ need not be closed in $\mathcal{X}$, in which case it would be impossible for $\mathbb{P}_*^X$ to be supported on $\mathcal{M}$. We nonetheless abuse language throughout our survey (both in the main text and in these formal boxes) and say $\mathbb{P}_*^X$ is supported on $\mathcal{M}$ when we actually mean that $\mathbb{P}_*^X(\mathcal{M}) = 1$ and that $\mathrm{supp}(\mathbb{P}_*^X) = \mathrm{cl}_{\mathcal{X}}(\mathcal{M})$.

- **Formalizing manifold-supported densities through measure theory** The manifold $\mathcal{M}$ can always be equipped with a Riemannian metric which it inherits from $\mathcal{X}$, making $\mathcal{M}$ into a Riemannian manifold. Riemannian manifolds admit a unique measure (defined over their Borel sets), $\lambda_{\mathcal{M}}$, which plays an analogous role to that of $\lambda_D$ on $\mathbb{R}^D$. The measure $\lambda_{\mathcal{M}}$ is called the Riemannian measure (or sometimes the Lebesgue measure) of $\mathcal{M}$. We refer readers interested in Riemannian measures to the treatments by Dieudonné (1973, Section 22 of Chapter 16) and Pennec (2006). Then $\mathbb{P}_*^X$ can be restricted to $\mathcal{M}$, resulting in the probability measure $\mathbb{P}_*^X|_{\mathcal{M}}$ defined over the Borel sets of $\mathcal{M}$ given by

$$\mathbb{P}_*^X|_{\mathcal{M}}(B) \coloneqq \mathbb{P}_*^X(B) \tag{2}$$

  for any Borel set $B$ of $\mathcal{M}$, and $p_*^X$ can then be defined as the density (i.e. Radon-Nikodym derivative) – assuming it exists – of this measure with respect to the corresponding Riemannian measure, i.e.

$$p_*^X \coloneqq \frac{\mathrm{d}\mathbb{P}_*^X|_{\mathcal{M}}}{\mathrm{d}\lambda_{\mathcal{M}}}. \tag{3}$$

  In other words $p_*^X$ is such that, for any Borel set $A$ of $\mathcal{X}$,

$$\mathbb{P}_*^X(A) = \int_{A \cap \mathcal{M}} p_*^X(x)\mathrm{d}\lambda_{\mathcal{M}}(x). \tag{4}$$

- **Formalizing manifold-supported densities through differential geometry** When the Riemannian manifold $\mathcal{M}$ is orientable, the manifold admits a Riemannian volume form $\mathrm{dvol}_{\mathcal{M}}$, and $p_*^X$ is defined as the function – assuming it exists – having the property that

$$\mathbb{P}_*^X(U) = \int_{U \cap \mathcal{M}} p_*^X \mathrm{dvol}_{\mathcal{M}} \tag{5}$$

  for every open set $U$ of $\mathcal{X}$, i.e. $\mathrm{dvol}_{\mathcal{M}}$ "plays the role" of the Riemannian measure $\lambda_{\mathcal{M}}$ in Equation 4. The technical tool allowing us to establish a correspondence between these two views of $p_*^X$ is known as the Riesz-Markov-Kakutani theorem (Rudin, 1987, Theorem 2.14) – which is sometimes also referred to as the Riesz representation theorem and should not be confused with a different theorem about Hilbert spaces bearing the same name. The Riesz-Markov-Kakutani theorem allows us to assign a unique measure to $\mathrm{dvol}_{\mathcal{M}}$, namely $\lambda_{\mathcal{M}}$, such that integrating continuous compactly-supported functions against them is equivalent. In this sense, $\lambda_{\mathcal{M}}$ is the natural measure to "extend" $\mathrm{dvol}_{\mathcal{M}}$. We point out that when $\mathcal{M}$ is non-orientable, even though $\mathrm{dvol}_M$ is not defined, integration on manifolds can still be carried out through the above measure-theoretic formulation.

# 3 Background

In this section we cover background topics and standard tools which we make use of throughout the survey. Expert readers may wish to skip to Section 4.

## 3.1 Deep Generative Models on Known Manifolds

This work focuses on data governed by the manifold hypothesis, wherein the dataset of interest is constrained to an *unknown* $d^*$-dimensional submanifold of $\mathcal{X}$. However, for some datasets, the manifold is known *a priori*, and the challenge lies in designing a generative model which can learn densities within the manifold. Generative modelling on known manifolds is a distinct task from our focus in this survey, but it is closely related, and we thus briefly summarize work on this topic below.

Gemici et al. (2016) first identified deep generative modelling on known manifolds as a problem of interest and showed that the change-of-variables formula (Section 3.3) used to train normalizing flows (Section 4.1.3) can be generalized to account for manifold structure. However, naïvely applying this idea can be numerically unstable, so past work has specifically focused on developing this approach further for manifolds such as tori, spheres, and hyperbolic spaces (Rezende et al., 2020; Bose et al., 2020; Sorrenson et al., 2023). Other works have designed generative models which preserve the symmetries of data on certain manifolds (Lie groups) of interest in the natural sciences (Kanwar et al., 2020; Boyda et al., 2021; Katsman et al., 2021). Using ordinary differential equations or stochastic differential equations to model data on known manifolds has also been proven to be effective (Mathieu & Nickel, 2020; Rozen et al., 2021; De Bortoli et al., 2022; Ben-Hamu et al., 2022; Lou et al., 2023; Chen & Lipman, 2024), with the advantage that these approaches can be defined independently of any parameterization of the manifold. Bonet et al. (2024) recently proposed an approach based on optimal transport (Section 3.5) for generative modelling on known manifolds.

### 3.2 Manifold Learning

Learning distributions whose support is an unknown manifold $\mathcal{M}$ implies learning $\mathcal{M}$ as well, at least implicitly. The field of manifold learning is thus closely related to the main topic of our work. The term "manifold learning" is often treated synonymously with dimensionality reduction, and refers to methods whose goal is to provide a useful representation of high-dimensional data by transforming it into a lower-dimensional space. That representation may provide information about the data such as its intrinsic dimension, yield a useful visualization in two or three dimensions, or serve as a simplified starting point for downstream supervised learning tasks. Generally, manifold learning methods fall into three categories: spectral methods (Pearson, 1901; Kruskal, 1964; Beals et al., 1968; Schölkopf et al., 1998; Roweis & Saul, 2000; Tenenbaum et al., 2000) which rely on the eigenvectors and eigenvalues of some matrix related to the data; probabilistic methods (Tipping & Bishop, 1999; van der Maaten & Hinton, 2008; McInnes et al., 2018), which treat datapoints as high-dimensional random vectors whose relevant information is contained in some low-dimensional latent variables; and bottleneck methods (Rumelhart et al., 1988; Kramer, 1991; Tishby et al., 2000; Kingma & Welling, 2014; Alemi et al., 2017), which rely on passing information through a low-dimensional bottleneck representation, often using neural networks. We refer the reader to the work of Ghojogh et al. (2023) for a comprehensive review of manifold learning using this tripartite classification.

The focus on manifold learning in this work is mostly on bottleneck methods such as autoencoders (Rumelhart et al., 1988) – and their many variants – whose core idea is to train an encoder-decoder pair of neural networks to ensure reconstruction, for example through a squared $\ell_2$ loss:[2]

$$\min_{\theta,\phi} \mathbb{E}_{X \sim p_*^X} \left[ \| X - g_\theta \left( f_\phi(X) \right) \|_2^2 \right]. \tag{6}$$

Here, the "bottleneck" refers to the $d$-dimensional encoder output $f_\phi(X)$, which the decoder $g_\theta$ uses to reconstruct $X$. Since $p_*^X$ is supported on $\mathcal{M}$, the objective in Equation 6 aims to learn the manifold in the sense that it encourages perfect reconstructions on it, or more formally, if $x \in \mathcal{M}$, then $x = g_{\theta^*}(f_{\phi^*}(x))$. We highlight that perfect reconstructions need not always be achievable – even under the nonparametric regime (Section 2.2) – due to topological constraints. We discuss this point, which we will revisit in Section 5.4, in the next grey box.

We finish this section with the observation that, despite what the term "manifold learning" might suggest, autoencoders do not by themselves formally learn $\mathcal{M}$ – even when perfect reconstructions are achieved. One might expect a perfectly trained autoencoder to characterize $\mathcal{M}$ as the set of possible decoder outputs. However, the encoder may not make use of the entire latent space, leaving some points that would not be seen by the decoder during training. As a result, points $z \in \mathcal{Z} \setminus f_{\phi^*}(\mathcal{M})$ might be decoded outside of $\mathcal{M}$, as illustrated in Figure 2. Alternatively, one might expect an autoencoder to characterize $\mathcal{M}$ as the set of points which are perfectly reconstructed. However, some points outside the manifold can in principle still be perfectly reconstructed, as also illustrated in Figure 2. Despite these observations, assuming perfect reconstructions, the set $f_{\phi^*}(\mathcal{M})$ together with the decoder $g_{\theta^*}$ jointly characterize $\mathcal{M}$, since $g_{\theta^*}(f_{\phi^*}(\mathcal{M})) =$

---

[2]Note that autoencoders are not by themselves generative models, but we nonetheless parameterize $g$ with $\theta$ (which we use for generative parameters) for consistency with other decoders in the rest of the paper.

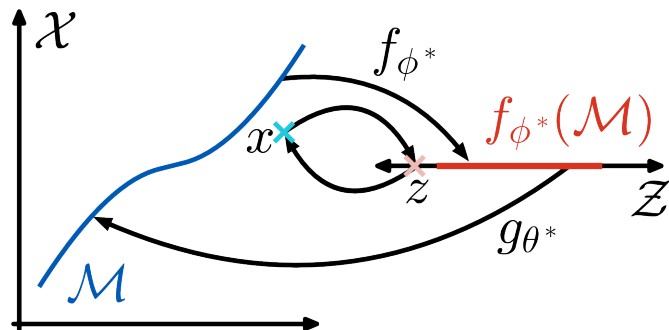

Figure 2: Illustration of why autoencoders, by themselves, do not characterize $\mathcal{M}$ even if they achieve perfect reconstructions on it. The illustrative point $z \in \mathcal{Z} \setminus f_{\phi^*}(\mathcal{M})$ is such that $x = g_{\theta^*}(z) \notin \mathcal{M}$, so that the set of possible decoder outputs does not match $\mathcal{M}$, i.e. $g_{\theta^*}(\mathcal{Z}) \neq \mathcal{M}$ – even though $\mathcal{M}$ is contained in $g_{\theta^*}(\mathcal{Z})$ due to the assumption of perfect reconstructions. Additionally, in this example $x \in \mathcal{X} \setminus \mathcal{M}$ is perfectly reconstructed, so that the set of perfectly reconstructed points does not match $\mathcal{M}$, i.e. $\{x \in \mathcal{X} \mid x = g_{\theta^*}(f_{\phi^*}(x))\} \neq \mathcal{M}$ – even though $\mathcal{M}$ is contained in this set whenever $x = g_{\theta^*}(f_{\phi^*}(x))$ for every $x \in \mathcal{M}$.

$\mathcal{M}$; we will revisit this point in Section 5.3 to show that training a DGM on data encoded by $f_{\phi^*}$ can characterize $\mathcal{M}$.

> **When are perfect reconstructions achievable?**  Ideally, the loss in Equation 6 achieves a value of 0 at optimality, which would directly imply that $x = g_{\theta^*}(f_{\phi^*}(x)), \mathbb{P}_*^X$-almost-surely. Under mild regularity conditions (Loaiza-Ganem et al., 2022a), this in turn implies that $x = g_{\theta^*}(f_{\phi^*}(x))$ for all $x \in \mathcal{M}$, in which case we say that the encoder-decoder pair $(f_{\phi^*}, g_{\theta^*})$ reconstructs $\mathcal{M}$ perfectly. When this condition is satisfied, the restriction $f_{\phi^*}|_{\mathcal{M}}$ is a (topological) embedding of $\mathcal{M}$ into $\mathcal{Z}$, as is evidenced by the existence of its continuous left-inverse, $g_{\theta^*}$. In other words, the existence of some continuous function $f$ that embeds $\mathcal{M}$ into $\mathcal{Z}$ is a necessary condition to achieve perfect reconstructions and thus to learn $\mathcal{M}$.
>
> In general, however, such an $f$ may not exist. For example, as we will see in Section 5.3, it is sometimes desirable to set $d$, the dimensionality of $\mathcal{Z}$, to be equal to $d^*$, the dimensionality of $\mathcal{M}$. This precludes the existence of $f$ for many manifolds $\mathcal{M}$, such as if $\mathcal{M}$ is a $d^*$-dimensional sphere, for which no embedding $f : \mathcal{M} \to \mathbb{R}^d$ is possible when $d = d^*$. In cases where no plausible embedding exists, even networks $(f_\phi, g_\theta)$ which come close to perfectly reconstructing $\mathcal{M}$ will incur numerical instability (Cornish et al., 2020). In some other cases, it is possible to resolve these topological issues by increasing $d$. For instance, a dimensionality of $d = 2d^* + 1$ is enough to topologically embed any manifold of dimension $d^*$ in $\mathbb{R}^d$ (Hurewicz & Wallman, 1948, Theorem V 3).

### 3.3 The Change-of-Variables Formula

It will often be the case that we have a density $p^Z$ on $\mathcal{Z}$ along with a decoder $g : \mathcal{Z} \to \mathcal{X}$. Together, these two components implicitly define the distribution of $X = g(Z)$, where $Z \sim p^Z$, and it will often be of interest to explicitly evaluate the density $p^X$ of $X$ (formally, $p^X$ is the pushforward density of $p^Z$ through $g$). The suitable tool is the change-of-variables formula, whose simplest form states that, when $\mathcal{Z} = \mathcal{X} = \mathbb{R}^D$, if $g$ is a diffeomorphism (i.e. a continuously differentiable function with a continuously differentiable inverse), then

$$p^X(x) = p^Z(z) \left|\det \nabla_z g(z)\right|^{-1} \tag{7}$$

$$= p^Z(f(x)) \left|\det \nabla_x f(x)\right|, \tag{8}$$

where $f = g^{-1}$ and $z = f(x)$, so that $\nabla_z g(z) \in \mathbb{R}^{D \times D}$ and $\nabla_x f(x) \in \mathbb{R}^{D \times D}$. An extension of this formula that will also be of use applies to the case where $\mathcal{Z} = \mathbb{R}^d$ with $d \leq D$. When $d < D$, $g : \mathcal{Z} \to \mathcal{X}$ cannot be

a diffeomorphism, but if it is injective it can be a diffeomorphism onto its image, $g(\mathcal{Z})$, in which case the density of $X$ is now given by

$$p^X(x) = p^Z(z) \left| \det \left( \nabla_z g(z)^\top \nabla_z g(z) \right) \right|^{-\frac{1}{2}}, \tag{9}$$

where again $z = f(x)$, but now $f : g(\mathcal{Z}) \to \mathcal{Z}$ is the left inverse of $g$ (i.e. $z' = f(g(z'))$ for all $z' \in \mathcal{Z}$), and $\nabla_z g(z) \in \mathbb{R}^{D \times d}$.

Several remarks about these formulas are worth making. (*i*) Computationally, it is often the case that Equation 8 is used, rather than Equation 7, as Equation 8 requires only a forward pass through the encoder $f$ (as well as computing its Jacobian determinant), whereas Equation 7 requires an additional forward computation through the decoder $g$; (*ii*) Equation 9, which is referred to as the injective change-of-variables formula, reduces to Equation 7 in the case where $d = D$, since the determinant distributes over products of square matrices; and (*iii*) when $d < D$, $p^X$ in Equation 9 is a manifold-supported density because it is only defined on a submanifold, $g(\mathcal{Z})$, of $\mathcal{X}$, much like $p_*^X$ which is only defined on $\mathcal{M}$. We refer the reader to the work of Köthe (2023) for a review of the uses the change-of-variables formula has within DGMs.

> Note that a more formal way of describing the change-of-variables formula is through the language of pushforward measures, where we have a measure $\mathbb{P}^Z$ on $\mathcal{Z}$ admitting a density $p^Z$ with respect to $\lambda_d$, along with the measurable map $g : \mathcal{Z} \to \mathcal{X}$. In the case of Equation 7 and Equation 8, $p^X$ corresponds to the density of $g_\# \mathbb{P}^Z$ with respect to $\lambda_D$; i.e. $p^X = \mathrm{d}g_\# \mathbb{P}^Z / \mathrm{d}\lambda_D$. In the case of Equation 9, when $g$ is a smooth embedding, $p^X$ is now a density with respect to the Riemannian measure on $g(\mathcal{Z})$, i.e. $p^X = \mathrm{d}g_\# \mathbb{P}^Z / \mathrm{d}\lambda_{g(\mathcal{Z})}$, where $g(\mathcal{Z})$ is treated as an embedded submanifold of $\mathcal{X}$.

### 3.4 Failures of KL Divergence

Despite the KL divergence being widely used throughout machine learning, it is most commonly used with the implicit assumption that the two involved densities are densities in the "same sense" (i.e. when they both admit the same dominating measure, see the grey box below). This assumption fails in the manifold setting when $p_\theta^X$ is full-dimensional, since $p_*^X$ is manifold-supported. We thus find it useful to provide the formal definition of the KL divergence in the grey box below, along with a discussion. In summary, the usual formula for computing KL divergence,

$$\mathbb{KL}\left(p_*^X \,\|\, p_\theta^X\right) = \mathbb{E}_{X \sim p_*^X}\left[\log \frac{p_*^X(X)}{p_\theta^X(X)}\right], \tag{10}$$

is only valid when $p_*^X$ and $p_\theta^X$ are such that for every subset $A$ of $\mathcal{X}$ that is assigned probability 0 by $p_\theta^X$, the density $p_*^X$ also assigns probability 0 to $A$. Whenever this property does not hold, $\mathbb{KL}(p_*^X \,\|\, p_\theta^X)$ is defined as infinity. It follows that in the manifold setting, $\mathbb{KL}(p_*^X \,\|\, p_\theta^X) = \infty = \mathbb{KL}(p_\theta^X \,\|\, p_*^X)$ when $p_\theta^X$ is full-dimensional, as illustrated in Figure 3(a). It also follows that $\mathbb{KL}(p_*^X \,\|\, p_\theta^X) = \infty$ even if $p_\theta^X$ is supported on a $d^*$-dimensional manifold – as long as $\mathcal{M}$ is not contained in the support of $p_\theta^X$ – as illustrated in Figure 3(b).

**KL divergence and maximum-likelihood** The most common way of attempting to minimize the KL divergence between the true distribution and the model is through maximum-likelihood:

$$\max_\theta \mathbb{E}_{X \sim p_*^X}\left[\log p_\theta^X(X)\right]. \tag{11}$$

When the KL divergence between $p_*^X$ and $p_\theta^X$ is not trivially equal to infinity, it can be written as

$$\mathbb{KL}(p_*^X \,\|\, p_\theta^X) = \int \log\left(\frac{p_*^X(x)}{p_\theta^X(x)}\right) p_*^X(x)\mathrm{d}x = \int p_*^X(x) \log p_*^X(x)\mathrm{d}x - \int p_*^X(x) \log p_\theta^X(x)\mathrm{d}x \tag{12}$$

$$= \mathbb{E}_{X \sim p_*^X}\left[\log p_*^X(X)\right] - \mathbb{E}_{X \sim p_*^X}\left[\log p_\theta^X(X)\right]. \tag{13}$$

Since $\mathbb{E}_{X \sim p_*^X}\left[\log p_*^X(X)\right]$ does not depend on $\theta$, this common derivation shows that as long as $\left|\mathbb{E}_{X \sim p_*^X}\left[\log p_*^X(X)\right]\right| < \infty$, maximum-likelihood optimization is equivalent to minimizing KL divergence.

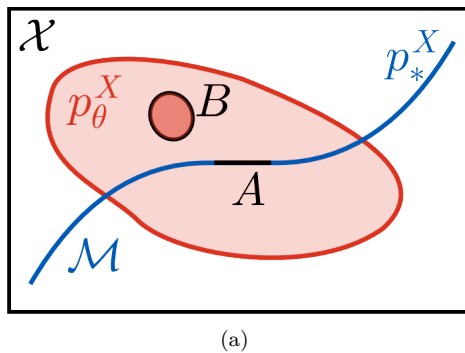 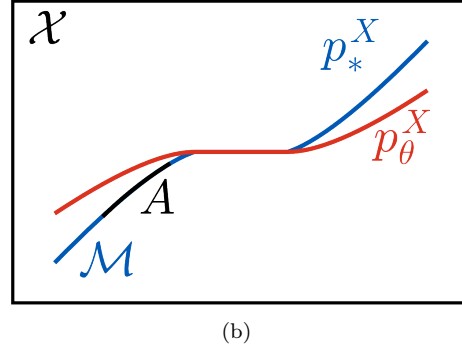

(a)             (b)

Figure 3: Illustration of why KL divergences can be infinite in the manifold setting. **(a)** $p_\theta^X$ has full-dimensional support (light red region), while $p_*^X$ is supported on a lower-dimensional manifold $\mathcal{M}$ (blue curve). The model $p_\theta^X$ assigns probability 0 to $A$, i.e. $\int_A p_\theta^X \, \mathrm{d}x = 0$, because the region $A$ has zero volume in $\mathcal{X}$. However, $p_*^X$ does not, since $\int_{A\cap\mathcal{M}} p_*^X \mathrm{dvol}_\mathcal{M} > 0$. We conclude that $\mathbb{KL}\left(p_*^X \,\|\, p_\theta^X\right) = \infty$. Meanwhile, $\int_{B\cap\mathcal{M}} p_*^X \mathrm{dvol}_\mathcal{M} = 0$ because $B \cap \mathcal{M} = \emptyset$, yet we have $\int_B p_\theta^X \, \mathrm{d}x > 0$, entailing that $\mathbb{KL}\left(p_\theta^X \,\|\, p_*^X\right) = \infty$. **(b)** Analogous example where now $p_\theta^X$ and $p_*^X$ are both supported on low-dimensional manifolds. Since $\mathcal{M}$ is not contained in the support of $p_\theta^X$, there exists a set $A$ to which $p_\theta^X$ assigns probability 0 despite having positive probability under $p_*^X$, so that $\mathbb{KL}\left(p_*^X \,\|\, p_\theta^X\right) = \infty$.

However, a key step in this derivation (the first equality) is the assumption that the KL divergence between $p_*^X$ and $p_\theta^X$ is not trivially infinite. As previously mentioned, in the manifold setting we will generally have $\mathbb{KL}(p_*^X \,\|\, p_\theta^X) = \infty$. It follows that in this setting, maximum-likelihood is not equivalent to KL divergence minimization, a point that we will later revisit.

---

Formally, the KL divergence between two probability measures $\mathbb{P}$ and $\mathbb{Q}$, $\mathbb{KL}(\mathbb{P} \,\|\, \mathbb{Q})$, is defined as

$$\mathbb{KL}(\mathbb{P} \,\|\, \mathbb{Q}) := \begin{cases} \int \log\left(\dfrac{\mathrm{d}\mathbb{P}}{\mathrm{d}\mathbb{Q}}(x)\right) \mathrm{d}\mathbb{P}(x), & \text{if } \dfrac{\mathrm{d}\mathbb{P}}{\mathrm{d}\mathbb{Q}} \text{ exists} \\ \infty, & \text{otherwise} \end{cases}, \tag{14}$$

where $\mathrm{d}\mathbb{P}/\mathrm{d}\mathbb{Q}$ denotes the Radon-Nikodym derivative of $\mathbb{P}$ with respect to $\mathbb{Q}$. By the Radon-Nikodym theorem, $\mathrm{d}\mathbb{P}/\mathrm{d}\mathbb{Q}$ exists if and only if $\mathbb{P} \ll \mathbb{Q}$. Finally, when both $\mathbb{P}$ and $\mathbb{Q}$ are dominated by the same measure $\eta$ – i.e. $\mathbb{P} \ll \eta$ and $\mathbb{Q} \ll \eta$ – with corresponding densities $p$ and $q$ with respect to $\eta$, the KL divergence between them simplifies to

$$\mathbb{KL}(\mathbb{P} \,\|\, \mathbb{Q}) = \mathbb{KL}(p \,\|\, q) = \int \log\left(\frac{p(x)}{q(x)}\right) p(x) \mathrm{d}\eta(x), \tag{15}$$

which recovers the commonly-used expressions for KL divergence (Equation 13) whenever $\eta$ is either the Lebesgue measure or the counting measure.

---

## 3.5 Wasserstein Distances

In Section 3.4, we summarized why $\mathbb{KL}(p_*^X \,\|\, p_\theta^X)$ does not provide a useful notion of divergence between $p_*^X$ and $p_\theta^X$ in the manifold setting, and the same is true of many other common divergences between distributions (see the discussion in Section 4.2). Wasserstein distances, which are based on the optimal transport problem (Villani, 2009; Peyré & Cuturi, 2019), provide a distance between distributions that remains meaningful even in the manifold setting. Despite the fact that accurately estimating Wasserstein distances is challenging (Arora et al., 2017), DGMs based on minimizing these distances tend to work very well in practice (e.g. Section 5.2.1 and Section 5.3.1).

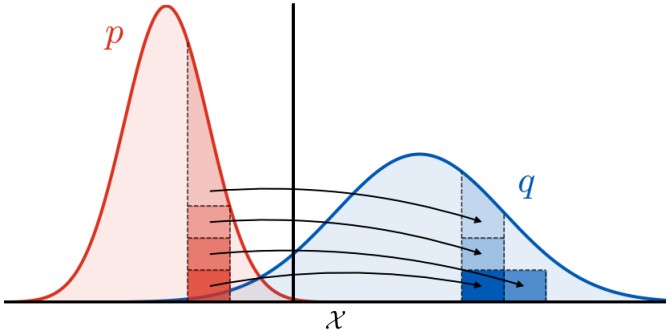

Figure 4: The optimal transport problem can be visualized as the minimum cost of "transporting" the density $p$ over to the density $q$. Picturing $p$ and $q$ as piles of dirt, each dirt particle from $p$ must be moved so that it becomes part of $q$. Moving dirt from $x$ to $y$ incurs a cost given by $c(x, y)$. The joint distribution $\gamma$ of $(X, Y)$ can be thought of as specifying the "transport plan": the constraint that its $X$-marginal matches $p$ ensures the starting pile of dirt is $p$; the constraint that its $Y$-marginal matches $q$ ensures the final pile of dirt is $q$; and its $(Y|X = x)$-conditional – illustrated with the black arrows in the figure – specifies how the dirt at $x$ from $p$ is (potentially stochastically) allocated to dirt from $q$. The most efficient plan possible for shifting all the dirt has an overall cost $\mathbb{W}^c(p, q)$. This analogy explains why the Wasserstein-1 distance is sometimes called the earth mover's distance.

The optimal transport problem between two densities $p$ and $q$ on $\mathcal{X}$ is given by

$$\mathbb{W}^c(p, q) := \inf_{\gamma \in \Pi(p,q)} \mathbb{E}_{(X,Y) \sim \gamma}[c(X, Y)], \tag{16}$$

where $c : \mathcal{X} \times \mathcal{X} \to \mathbb{R}$ is called the cost function, and $\Pi(p, q)$ is the set of distributions on $\mathcal{X} \times \mathcal{X}$ whose marginals match $p$ and $q$, respectively. Intuitively, the optimal transport problem can be understood as the cost (as measured by $c$) of "transporting" $p$ to $q$, as illustrated in Figure 4. When $c$ is given by the $\ell_1$ distance (i.e. $c(x, y) = \|x - y\|_1$) $\mathbb{W}^c$ is called the Wasserstein-1 distance, and is denoted as $\mathbb{W}_1$. The $\mathbb{W}_1$ metric admits the following well-known dual formulation:

$$\mathbb{W}_1(p, q) = \sup_{h \in \mathcal{H}} \mathbb{E}_{X \sim p}[h(X)] - \mathbb{E}_{X \sim q}[h(X)], \tag{17}$$

where $\mathcal{H} := \{h : \mathcal{X} \to \mathbb{R} \mid h \text{ is Lipschitz and } \mathrm{Lip}(h) \leq 1\}$, and $\mathrm{Lip}(h)$ denotes the Lipschitz constant of $h$. Analogously, when $c$ is given by the squared $\ell_2$ distance (i.e. $c(x, y) = \|x - y\|_2^2$) $\sqrt{\mathbb{W}^c}$ is called the Wasserstein-2 distance, and is denoted as $\mathbb{W}_2$. The Wasserstein distances $\mathbb{W}_1(p_*^X, p_\theta^X)$ and $\mathbb{W}_2(p_*^X, p_\theta^X)$ remain meaningfully defined even in the manifold setting (formally, this is because they metrize weak convergence, which we discuss in the grey box below), and thus provide sensible optimization objectives. For $\mathbb{W}_1$, this property can be informally understood through Equation 17, which essentially says that two distributions are close in Wasserstein distance if no Lipschitz function can discriminate between them. Intuitively, if no such function can discern between $p_*^X$ and $p_\theta^X$, then they must be "truly" close, even if one is manifold-supported and the other full-dimensional (or if both are supported on non-overlapping manifolds).

First, we point out that the optimal transport problem in Equation 16 applies to arbitrary probability measures $\mathbb{P}$ and $\mathbb{Q}$ on $\mathcal{X}$, not only densities:

$$\mathbb{W}^c(\mathbb{P}, \mathbb{Q}) := \inf_{\Gamma \in \Pi(\mathbb{P},\mathbb{Q})} \mathbb{E}_{(X,Y) \sim \gamma}[c(X, Y)], \tag{18}$$

where $\Pi(\mathbb{P}, \mathbb{Q}) := \{\Gamma \in \Delta(\mathcal{X} \times \mathcal{X}) \mid \Gamma(A \times \mathcal{X}) = \mathbb{P}(A) \text{ and } \Gamma(\mathcal{X} \times A) = \mathbb{Q}(A)$ for every measurable set $A \subset \mathcal{X}\}$, and $c : \mathcal{X} \times \mathcal{X} \to \mathbb{R}$ is measurable.

If $c$ is the $\ell_1$ (or $\ell_2$) distance, then convergence in $\mathbb{W}^c$ is equivalent to convergence in distribution plus convergence in first (or second) moments (Villani, 2009, Theorem 6.9). In particular, this

implies that if $\mathcal{X}$ is bounded (and $c$ is either the $\ell_1$ or $\ell_2$ distance), then $\mathbb{W}^c$ metrizes weak convergence, meaning that given a sequence $(\mathbb{P}^X_{\theta_t})^\infty_{t=1}$ of probability measures, $\mathbb{W}^c(\mathbb{P}^X_{\theta_t}, \mathbb{P}^X_*) \to 0$ as $t \to \infty$ if and only if $\mathbb{P}^X_{\theta_t} \xrightarrow{\omega} \mathbb{P}^X_*$ as $t \to \infty$. Arjovsky et al. (2017) identified that metrizing weak convergence is a desirable property in an optimization objective for training DGMs, as it ensures that "getting closer and closer" to the target distribution is properly quantified, even in the presence of dimensionality mismatch (see Appendix A for a more detailed discussion of this point).

Note that the KL divergence does not metrize weak convergence. Let us illustrate why this is problematic through an example by letting $\mathbb{P}^{X_\sigma}_* = \mathbb{P}^X_* \circledast \mathcal{N}(0, \sigma^2 I_D)$, where $\mathbb{P}^X_*$ is supported on $\mathcal{M}$. As $\sigma \to 0^+$, $\mathbb{P}^{X_\sigma}_*$ gets closer to $\mathbb{P}^X_*$, yet this is not reflected in the KL divergence, since $\mathbb{KL}(\mathbb{P}^X_* \| \mathbb{P}^{X_\sigma}_*) = \infty$ for every $\sigma > 0$ (this is because $\mathbb{P}^X_* \ll \mathbb{P}^{X_\sigma}_*$ does *not* hold). Thus $\mathbb{KL}(\mathbb{P}^X_* \| \mathbb{P}^{X_\sigma}_*) \to \infty$ as $\sigma \to 0^+$. This means that, despite $\mathbb{KL}(\mathbb{P}^X_* \| \mathbb{P}^{X_\sigma}_*)$ being minimized at $\sigma = 0$, the KL divergence provides no learning signal, and the same holds for the reverse KL. On the other hand, $\mathbb{W}^c(\mathbb{P}^X_*, \mathbb{P}^{X_\sigma}_*) \to 0$ as $\sigma \to 0^+$, provided that $\mathbb{W}^c$ metrizes weak convergence.

## 3.6 Maximum Mean Discrepancy

Similarly to Wasserstein distances (Section 3.5), the maximum mean discrepancy (MMD; Gretton et al., 2006) provides a notion of distance between probability distributions which remains mathematically meaningful in the manifold setting. The MMD between two probability densities $p$ and $q$ on $\mathcal{X}$, $\mathbb{MMD}_k(p, q)$, is given by

$$\mathbb{MMD}_k(p, q) \coloneqq \left( \mathbb{E}_{X, X' \sim p}[k(X, X')] - 2\mathbb{E}_{X \sim p, Y \sim q}[k(X, Y)] + \mathbb{E}_{Y, Y' \sim q}[k(Y, Y')] \right)^{\frac{1}{2}}, \tag{19}$$

where $X, X', Y, Y'$ are independent, and $k : \mathcal{X} \times \mathcal{X} \to \mathbb{R}$ is a symmetric positive semi-definite kernel (i.e. a function having the property that, for any $n \in \mathbb{N}$ and $x_1, \dots, x_n \in \mathcal{X}$, the $n \times n$ matrix $K$ given by $K_{ij} = k(x_i, x_j)$ is symmetric positive semi-definite), which is set as a hyperparameter.

The MMD has several desirable mathematical properties. ($i$) Under some regularity conditions which are satisfied by many commonly-used kernels, the MMD is a metric in the space of probability distributions over $\mathcal{X}$. ($ii$) $\mathbb{MMD}^2_k(p, q)$ can be straightforwardly estimated in an unbiased manner through Monte Carlo sampling, making it particularly amenable to gradient-based optimization. ($iii$) Conditions on $k$ which make $\mathbb{MMD}_k$ meaningfully defined in the manifold setting (formally, conditions under which MMD metrizes weak convergence) are known (Simon-Gabriel & Schölkopf, 2018; Simon-Gabriel et al., 2023). Provided that $\mathcal{X}$ is compact (which is the case for images in $[0, 1]^D$), these conditions hold for most commonly-used kernels, meaning that MMD can be used to compare distributions regardless of their support.

# 4 Manifold-Unaware Deep Generative Models

In this section we describe popular deep generative modelling frameworks which were not developed with the manifold setting in mind, and discuss their inability to learn $p^X_*$ in this setting.

## 4.1 The Problem with Likelihood-Based Approaches: Manifold Overfitting

Likelihood-based deep generative models are a broad and popular class of models, which includes variational autoencoders (Kingma & Welling, 2014; Rezende et al., 2014), normalizing flows (Dinh et al., 2015; 2017), energy-based models (Xie et al., 2016; Du & Mordatch, 2019), continuous autoregressive models (Uria et al., 2013), and more (Bond-Taylor et al., 2022). At a high-level, these models leverage neural networks to construct a full-dimensional density $p^X_\theta$. The models are trained by maximizing, sometimes approximately, the log-likelihood:

$$\max_\theta \mathbb{E}_{X \sim p^X_*}[\log p^X_\theta(X)]. \tag{20}$$

When the underlying density $p^X_*$ is full-dimensional, this objective is equivalent to minimizing the KL divergence between $p^X_*$ and $p^X_\theta$ (Equation 13). However, in our setting of interest $p^X_*$ is manifold-supported,

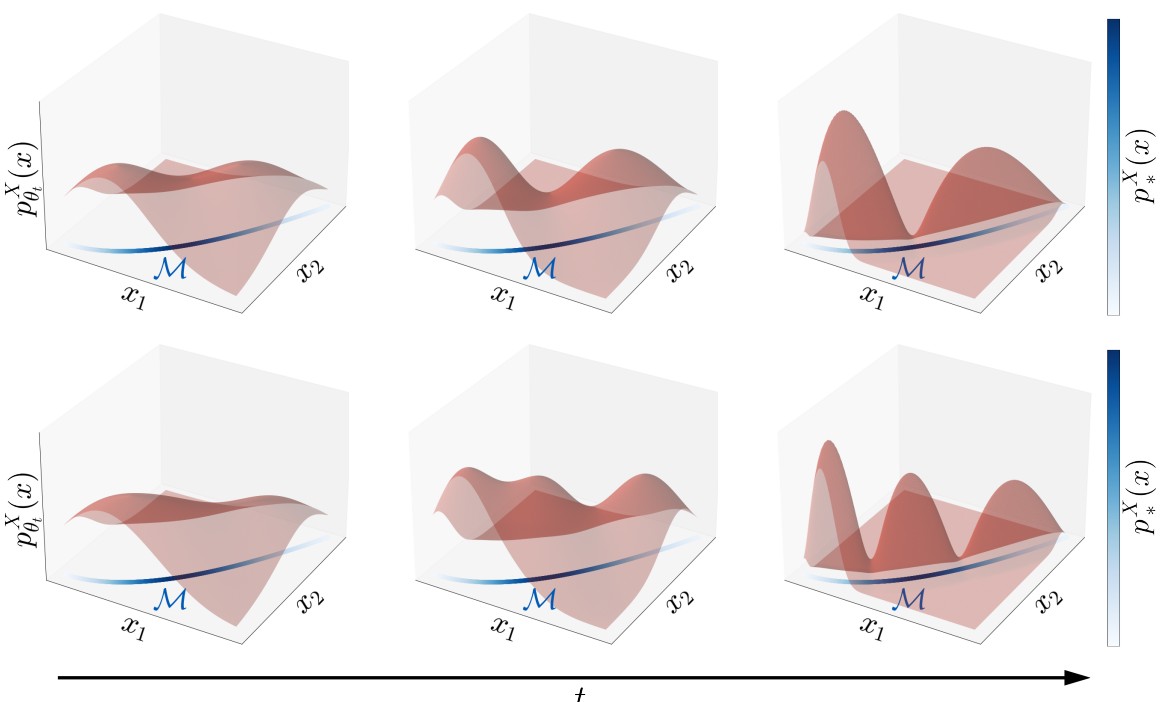

Figure 5: Illustration of manifold overfitting, where the 1-dimensional $p_*^X$ (shades of blue) along a curve $\mathcal{M}$ in 2-dimensional ambient space is improperly approximated. Each row shows a sequence of full-dimensional densities $p_{\theta_t}^X$ (red surfaces) having the property that their likelihood diverges to infinity on all of $\mathcal{M}$, yet each sequence approximates a different manifold-supported density $p_\dagger^X$ on $\mathcal{M}$: the top sequence will recover a bimodal distribution on $\mathcal{M}$ and the bottom sequence a trimodal one, despite $p_*^X$ being unimodal.

and as mentioned in Section 3.4, this equivalence breaks down, leading to the natural question: what happens if the likelihood is optimized when $p_\theta^X$ is full-dimensional but $p_*^X$ is not?

The first consequence of this dimensionality misspecification is that the log-likelihood does not admit a maximum as it can be made arbitrarily large. To see this, consider a sequence of full-dimensional models $(p_{\theta_t}^X)_{t=0}^\infty$ which concentrate more and more mass around $\mathcal{M}$ during training, as depicted in Figure 5. If $\mathcal{M}$ was full-dimensional, it would be impossible to have $p_{\theta_t}^X(x) \to \infty$ as $t \to \infty$ for all $x \in \mathcal{M}$, as doing so would quickly violate the requirement that the densities integrate to 1. However, when $\mathcal{M}$ is low-dimensional, it is "infinitely thin" in $\mathbb{R}^D$, and thus the model densities can be made to diverge to infinity along the entire manifold. This phenomenon is illustrated twice in Figure 5.

At a first glance, the fact that the likelihood does not admit a maximum might seem inconsequential, as one might hope that as long as $\mathbb{E}_{X \sim p_*^X}[\log p_{\theta_t}^X(X)] \to \infty$ as $t \to \infty$, then $p_*^X$ is still being learned. However, this is not the case, and the reason is once again illustrated in Figure 5: there are many ways in which the likelihood can diverge to infinity. Loaiza-Ganem et al. (2022a) formalized this intuition by proving that under mild regularity conditions, for any manifold-supported density $p_\dagger^X$ on $\mathcal{M}$, there always exists a sequence of full-dimensional densities which simultaneously (*i*) becomes arbitrarily large on the entire manifold, in turn maximizing likelihood, yet (*ii*) approximates $p_\dagger^X$ rather than the true data-generating density $p_*^X$. The latter condition is formalized using weak convergence in the grey box below, but can be intuitively understood as saying that samples from $p_{\theta_t}^X$ and $p_\dagger^X$ become indistinguishable as $t \to \infty$.[3]

---

[3]Note that approximating $p_\dagger^X$ does not imply that $p_{\theta_t}^X(x) \to p_\dagger^X(x)$ as $t \to \infty$ for all $x$ because $p_{\theta_t}^X$ is full-dimensional, whereas $p_\dagger^X$ is manifold-supported.

An immediate consequence of this result is that maximum-likelihood is an ill-posed objective in the manifold setting, as it simply encourages models to concentrate mass around $\mathcal{M}$ with no concern for the distribution within it. Loaiza-Ganem et al. (2022a) thus call this behaviour *manifold overfitting*. Several consequences of manifold overfitting are worth discussing. (*i*) Manifold overfitting does *not* imply that model densities diverge to infinity on *all* of $\mathcal{M}$; as long as these densities diverge to infinity on a subset of non-zero probability under $p_*^X$ and do not converge to zero on the rest of $\mathcal{M}$, the log-likelihood will still be "maximized", i.e. $\mathbb{E}_{X \sim p_*^X}[\log p_{\theta_t}^X(X)] \to \infty$ as $t \to \infty$. Analogously, densities could diverge to infinity on a superset of $\mathcal{M}$ – such as a manifold of dimension higher than $d^*$ but lower than $D$ (Koehler et al., 2022). Similarly, the log-likelihood can be made to diverge to infinity in such a way that the sequence of models $p_{\theta_t}^X$ does not learn any distribution $p_\dagger^X$ (Loaiza-Ganem et al., 2022a). In other words, the behaviour of models which "maximize" likelihood in the manifold setting can be pathological beyond $p_{\theta_t}^X(x)$ diverging to infinity if and only if $x \in \mathcal{M}$. (*ii*) One might be hopeful that in practice these pathological scenarios are avoided through the optimization dynamics of gradient descent so that $p_*^X$ is properly learned, yet this is not the case (Koehler et al., 2022). (*iii*) Manifold overfitting *cannot* be detected by using test likelihoods: as long as the test data is generated from $p_*^X$, then it lies on $\mathcal{M}$ with probability 1, and thus test likelihoods are also subject to degenerate behaviour. This observation highlights that, in the manifold setting, test log-likelihoods should be avoided as a DGM evaluation metric, and that sample-based metrics (Heusel et al., 2017; Borji, 2019; Stein et al., 2023) should be favoured instead. This unreliability of test log-likelihoods is consistent with the fact that they are not always correlated with sample quality when modelling images (Theis et al., 2016).

---

The manifold overfitting result of Loaiza-Ganem et al. (2022a) can be more formally stated as saying that, under some regularity conditions and provided that $d^* < D$, for any distribution $\mathbb{P}_\dagger^X$ on $\mathcal{X}$ supported on $\mathcal{M}$, there exists a sequence of distributions $(\mathbb{P}_{\theta_t}^X)_{t=1}^\infty$ such that:

- $\mathbb{P}_{\theta_t}^X$ is full-dimensional, i.e. $\mathbb{P}_{\theta_t}^X \ll \lambda_D$, for every $t$.

- For every $x \in \mathcal{M}$, it holds that $p_{\theta_t}^X(x) \to \infty$ as $t \to \infty$, where $p_{\theta_t}^X$ is a density of $\mathbb{P}_{\theta_t}^X$ with respect to $\lambda_D$.

- For every $x \in \mathcal{X} \setminus \mathrm{cl}_{\mathcal{X}}(\mathcal{M})$, it holds that $p_{\theta_t}^X(x) \to 0$ as $t \to \infty$.

- $\mathbb{P}_{\theta_t}^X \xrightarrow{\omega} \mathbb{P}_\dagger^X$ as $t \to \infty$.

Loaiza-Ganem et al. (2022a) proved this result under the assumption that $\mathcal{M}$ is analytic. Despite their other regularity conditions being very mild, this is a strong assumption. However, as we now argue, the result actually holds for arbitrary smooth submanifolds $\mathcal{M}$ of $\mathcal{X}$. Gray (1974, Theorem 3.1) proved a result which immediately implies that, if $\mathcal{M}$ is an analytic Riemannian manifold, then for $x \in \mathcal{M}$, as $\varepsilon \to 0$,

$$\lambda_{\mathcal{M}}\left(B_\varepsilon^{\mathcal{M}}(x)\right) = v(d^*)\, \varepsilon^{d^*} \left(1 + \mathcal{O}(\epsilon^2)\right), \tag{21}$$

where $\lambda_{\mathcal{M}}$ is the Riemannian measure on $\mathcal{M}$, $B_\varepsilon^{\mathcal{M}}(x)$ denotes a geodesic ball in $\mathcal{M}$ of radius $\varepsilon$ centred at $x$, and $v(d^*)$ is the volume of a $d^*$-dimensional Euclidean ball of radius 1. Loaiza-Ganem et al. (2022a) used the assumption that $\mathcal{M}$ is analytic only to apply Equation 21 by evoking the result of Gray (1974). However, Equation 21 is known to hold for arbitrary smooth Riemannian manifolds (Gallot et al., 2004, Theorem 3.98). It immediately follows that the result of Loaiza-Ganem et al. (2022a) indeed holds for arbitrary smooth submanifolds $\mathcal{M}$ of $\mathcal{X}$, even if they are not analytic.

---

### 4.1.1 The Unavoidable Numerical Instability of High-Dimensional Likelihoods

The manifold overfitting result of Loaiza-Ganem et al. (2022a) described in Section 4.1 establishes that maximum-likelihood is an ill-posed objective for high-dimensional densities in the manifold setting. Before continuing our review of existing work, we point out that their result does not rule out the possibility of somehow addressing the pathological behaviour of maximum-likelihood, for example by adding a regularizer. Here we prove that it is actually impossible to do so, by showing that for any "infinitely thin" subset $M$ of $\mathcal{X}$ (of which $\mathcal{M}$ is an example, but here we do not require $M$ to be a manifold), any density $p_\dagger^X$ supported

on $M$, and any sequence of $D$-dimensional models $p_{\theta_t}^X$ which learn $p_\dagger^X$, the following holds: ($i$) for any $x \in \mathcal{X}$ outside of $M$, $p_{\theta_t}^X(x)$ gets arbitrarily close to 0 as $t \to \infty$; and ($ii$) for any $x \in M$ and any $L > 0$, for large enough $t$ it holds that $p_{\theta_t}^X(x') > L$ for some $x' \in \mathcal{X}$ arbitrarily close to $x$. We formally state our theorem and include a technical discussion in the grey box below.

Technicalities aside, our result shows that likelihoods become arbitrarily close to 0 outside $M$, and that they become arbitrarily large on it (or arbitrarily close to it). The problem here is twofold: likelihoods are unstable not only because they become arbitrarily large around $M$, but also because they must change very rapidly to approach 0 outside of it. In particular, this implies that if $p_{\theta_t}^X$ is Lipschitz, the corresponding Lipschitz constant must blow up as $t \to \infty$.

One way to interpret our result is as a "soft generalization" of the manifold overfitting result of Loaiza-Ganem et al. (2022a); whereas they show that for any target $p_\dagger^X$ there exists a sequence of models which approximates it while exploding on $\mathcal{M}$ and converging to 0 elsewhere, we show that *any* sequence recovering $p_\dagger^X$ will exhibit similar pathological behaviour on $M$. Two implications of our result are worth discussing:

- **Numerical instability of likelihood evaluation**  Our theorem applies even if the models were not trained through maximum-likelihood, so that if the target distribution is correctly recovered through any means, density evaluation will remain numerically unstable – even when training itself does not involve likelihoods and is numerically stable. To see this, simply apply our theorem with $p_\dagger^X = p_*^X$, which immediately yields that any sequence of $D$-dimensional models which learn $p_*^X$ will do so with numerically unstable likelihoods. We can gain intuition as to why this should indeed be the case through Figure 5: the only way for the red surfaces $(p_{\theta_t}^X)$ to recover the density on the blue curve $(p_*^X)$ is by spiking to infinity around it, and by not assigning mass elsewhere.

- **Unfixability of maximum-likelihood**  Another consequence of our result is that maximum-likelihood cannot be "fixed"; for example, any regularizer added to it which ensures that $p_*^X$ is learned (rather than some arbitrary $p_\dagger^X$) would not circumvent the aforementioned numerical instabilities – provided it does not obviate the need to compute likelihoods (or any surrogates used) during training (e.g. by cancelling out the log-likelihood, at which point it would not fit the description of a regularizer anymore). Analogously, any regularizer or architecture guaranteeing numerical stability of likelihoods would be such that $p_*^X$ is not learned. In other words, our result ensures that learning $p_*^X$ and having numerically stable likelihoods cannot happen simultaneously.

---

**Theorem 1** (Likelihood Instability of Deep Generative Models). *Let $M \subset \mathcal{X}$ be a Borel set such that $\lambda_D(\mathrm{cl}_{\mathcal{X}}(M)) = 0$, and let $\mathbb{P}_\dagger^X$ be a probability measure on $\mathcal{X}$ such that $\mathbb{P}_\dagger^X(M) = 1$ and $\mathrm{supp}(\mathbb{P}_\dagger^X) = \mathrm{cl}_{\mathcal{X}}(M)$. Let $(\mathbb{P}_{\theta_t}^X)_{t=1}^\infty$ be a sequence of probability measures on $\mathcal{X}$ such that $\mathbb{P}_{\theta_t}^X \xrightarrow{\omega} \mathbb{P}_\dagger^X$ as $t \to \infty$ and $\mathbb{P}_{\theta_t}^X \ll \lambda_D$, with corresponding densities $p_{\theta_t}^X$. Then:*

- $\displaystyle \liminf_{t \to \infty} p_{\theta_t}^X(x) = 0$, $\lambda_D$*-almost-everywhere on* $\mathcal{X} \setminus \mathrm{cl}_{\mathcal{X}}(M)$.

- $\displaystyle \sup_{x' \in B_\varepsilon(x)} p_{\theta_t}^X(x') \to \infty$ *as* $t \to \infty$ *for every* $x \in \mathrm{cl}_{\mathcal{X}}(M)$ *and every* $\varepsilon > 0$*, where* $B_\varepsilon(x) := \{x' \in \mathcal{X} \mid \|x' - x\|_2 < \varepsilon\}$.

*Proof.* See Appendix B.1. □

We now make some relevant observations about the Likelihood Instability Theorem. ($i$) Note that we only require the closure of $M$ to have Lebesgue measure 0, so it need not be a manifold. Our result thus applies in settings beyond the standard manifold hypothesis, such as when $M$ is given by a union of manifolds (Brown et al., 2023), or by a non-manifold set with singularities (Von Rohrscheidt & Rieck, 2023; Wang & Wang, 2024). ($ii$) $\liminf_{t \to \infty} p_{\theta_t}^X(x)$ cannot in general be replaced by $\lim_{t \to \infty} p_{\theta_t}^X(x)$ since the limit need not exist, but as an immediate corollary, if the limit exists, then it must be 0 $\lambda_D$-almost-everywhere on $\mathcal{X} \setminus \mathrm{cl}_{\mathcal{X}}(M)$. ($iii$) We also point out that $\sup_{x' \in B_\varepsilon(x)} p_{\theta_t}^X(x')$ cannot be in general replaced by $p_{\theta_t}^X(x)$ either, despite our conclusion holding for every $\varepsilon > 0$. Intuitively, this is

> because for any given $x \in M$ the divergence to infinity of the density might not happen at $x$, but rather on a sequence converging to it: we provide an illustrative example in Appendix B.1. ($iv$) We do emphasize that despite not concluding that $p^X_{\theta_t}(x) \to 0$ outside of $M$ nor that $p^X_{\theta_t}(x) \to \infty$ in $M$, our result does unequivocally ensure numerical instability of the involved densities.

### 4.1.2 Variational Autoencoders

Variational autoencoders (VAEs; Kingma & Welling, 2014; Rezende et al., 2014) are a class of likelihood-based models. The continuous VAEs that we consider here specify a fixed prior density $p^Z$ (commonly a standard Gaussian) on $\mathcal{Z}$, along with a learnable conditional full-dimensional likelihood $p^{X|Z}_\theta$ on $\mathcal{X}$, which is often a parameterized Gaussian,

$$p^{X|Z}_\theta(x|z) = \mathcal{N}\left(x; g_\theta(z), \Sigma^{X|Z}_\theta(z)\right), \tag{22}$$

where $\Sigma^{X|Z}_\theta : \mathcal{Z} \to \mathbb{R}^{D \times D}$ is symmetric positive definite, often given by $\gamma I_D$, where $\gamma > 0$ is treated as a free parameter rather than the output of a neural network. The conditional likelihood $p^{X|Z}_\theta(\cdot|z)$ in Equation 22 can be understood as a stochastic decoder, whose mean is given by the deterministic decoder $g_\theta(z)$. Together, the prior and the conditional likelihood implicitly define the marginal likelihood over data:

$$p^X_\theta(x) = \int p^Z(z) p^{X|Z}_\theta(x|z) \mathrm{d}z. \tag{23}$$

Since computing the marginal likelihood involves an intractable integral, a variational posterior density $q^{Z|X}_\phi$ on $\mathcal{Z}$ is introduced, and the following objective, called the evidence lower bound (ELBO), is jointly maximized over $\theta$ and $\phi$:

$$\mathcal{E}_{p^X_*}(\theta, \phi) := \mathbb{E}_{X \sim p^X_*}\left[\mathbb{E}_{Z \sim q^{Z|X}_\phi(\cdot|X)}[\log p^{X|Z}_\theta(X|Z)] - \mathbb{KL}\left(q^{Z|X}_\phi(\cdot|X) \,\middle\|\, p^Z\right)\right] \tag{24}$$

$$= \mathbb{E}_{X \sim p^X_*}\left[\log p^X_\theta(X) - \mathbb{KL}\left(q^{Z|X}_\phi(\cdot|X) \,\middle\|\, p^{Z|X}_\theta(\cdot|X)\right)\right] \leq \mathbb{E}_{X \sim p^X_*}[\log p^X_\theta(X)], \tag{25}$$

where $p^{Z|X}_\theta$ denotes the true posterior density, which is implicitly defined by the prior $p^Z$ and the conditional likelihood $p^{X|Z}_\theta$. Note that Equation 24 is used as the optimization objective, since the true posterior density cannot be tractably evaluated. While not directly usable as an objective, Equation 25 shows that the ELBO lower-bounds the log-likelihood $\mathbb{E}_{X \sim p^X_*}[\log p^X_\theta(X)]$, and differs from it only by the error incurred by $q^{Z|X}_\phi$ to approximate the true posterior: this is often used as a justification for using the ELBO as an objective, as it simultaneously encourages learning $\theta$ through maximum-likelihood, and $\phi$ so that $q^{Z|X}_\phi$ matches the true posterior. It is common to also specify $q^{Z|X}_\phi$ as a Gaussian,

$$q^{Z|X}_\phi(z|x) = \mathcal{N}\left(z; f_\phi(x), \Sigma^{Z|X}_\phi(x)\right), \tag{26}$$

where $\Sigma^{Z|X}_\phi : \mathcal{X} \to \mathbb{R}^{d \times d}$ is also symmetric positive definite, and often given by a diagonal matrix with positive entries along its diagonal. In an analogous manner to the conditional likelihood, the variational posterior $q^{Z|X}_\phi(\cdot|x)$ in Equation 26 can be interpreted as a stochastic encoder, whose mean is given by the deterministic encoder $f_\phi(x)$.

An issue which commonly affects VAEs is posterior collapse (Chen et al., 2017; Wang et al., 2021), where the learned variational posterior $q^{Z|X}_{\phi^*}$ partially collapses to the prior $p^Z$. We will shortly explain posterior collapse in VAEs through the manifold lens, and thus we briefly summarize the phenomenon here: in the case where $\Sigma^{Z|X}_{\phi^*}$ is taken as a diagonal matrix, a subset of the diagonal entries of $\Sigma^{Z|X}_{\phi^*}(x)$ collapses to 1 for $x \in \mathcal{M}$, and the corresponding entries of $f_{\phi^*}(x)$ collapse to 0, matching the standard Gaussian prior $p^Z$; whereas the remaining diagonal entries of $\Sigma^{Z|X}_{\phi^*}(x)$ are extremely close to 0, essentially losing stochasticity along these coordinates. In other words, VAEs tend to only "use" a subset of the coordinates of their latent space $\mathcal{Z}$ to obtain data samples and default the rest to the prior.

**VAEs through the lens of the manifold hypothesis**   Despite the autoencoder-like structure of VAEs that leverages low-dimensional representations, VAEs as presented above are full-dimensional models. This is a direct consequence of the choice of $p_\theta^{X|Z}$, which always assigns strictly positive density to all of $\mathcal{X}$, i.e. $p_\theta^{X|Z}(x|z) > 0$ for all $x \in \mathcal{X}$ and $z \in \mathcal{Z}$. This in turn implies that $p_\theta^X(x) > 0$ for all $x \in \mathcal{X}$, so that the model density is not supported on a low-dimensional manifold. Thus, since the ELBO is maximized as a proxy for the log-likelihood, intuitively VAEs should be subject to manifold overfitting. Dai & Wipf (2019) formally show that this is indeed the case by proving that, subject to some regularity conditions and assuming that $d \geq d^*$, for any manifold-supported density $p_\dagger^X$ on $\mathcal{M}$, there exists a sequence of VAE models parameterized by $(\theta_t, \phi_t)_{t=1}^\infty$ such that: $(i)$ $\mathcal{E}_{p_*^X}(\theta_t, \phi_t) \to \infty$ as $t \to \infty$; $(ii)$ the VAE models $p_{\theta_t}^X$ learn $p_\dagger^X$ instead of $p_*^X$; and $(iii)$ $\mathbb{KL}(q_{\phi_t}^{Z|X}(\cdot|x) \,\|\, p_{\theta_t}^{Z|X}(\cdot|x)) \to 0$ as $t \to \infty$ for every $x \in \mathcal{M}$. Although this result predates the manifold overfitting result of Loaiza-Ganem et al. (2022a) from Section 4.1, it can be understood as saying that maximizing the ELBO instead of the log-likelihood does not prevent manifold overfitting. It is also worth noting that while the work of Loaiza-Ganem et al. (2022a) extends the result from Dai & Wipf (2019) to non-VAE models and VAE models with flexible variational posteriors (Rezende & Mohamed, 2015; Kingma et al., 2016; van den Berg et al., 2018; Caterini et al., 2021a) – since if the variational posterior is flexible enough, optimizing the ELBO becomes equivalent to maximizing the log-likelihood in the nonparametric regime (Section 2.2) – it does not immediately imply that manifold overfitting can happen when $q_\phi^{Z|X}$ is Gaussian, which Dai & Wipf (2019) do prove.

Well-known training instabilities of VAEs can be understood as consequences of manifold overfitting. For example, as previously mentioned, it is common to use $\Sigma_\theta^{X|Z}(z) = \gamma I_D$ for every $z \in \mathcal{Z}$, where $\gamma$ is a free parameter ($\gamma$ is also often treated as a non-learnable hyperparameter instead). This purposely simplistic choice is made for the sake of training stability, e.g. parameterizing $\Sigma_\theta^{X|Z}$ as a diagonal matrix whose non-zero entries are given by a neural network can easily result in divergent training (Lin et al., 2019; Rybkin et al., 2021); the added flexibility of the stochastic decoder of this VAE can be understood as making it more prone to experience manifold overfitting. We also highlight that, when modelling images, the best empirically performing VAEs do not use a Gaussian conditional likelihood $p_\theta^{X|Z}$. Instead, they treat pixels as discrete and use a categorical $p_\theta^{X|Z}$ (Vahdat & Kautz, 2020). Mathematically, these discrete models are not afflicted by manifold overfitting (we discuss why in Section 6), which helps explain why they outperform VAEs with continuous conditional likelihoods.

Dai & Wipf (2019) provide further insights into the interplay between VAEs and the manifold hypothesis beyond manifold overfitting. In particular, they also show that under appropriate conditions, Gaussian VAEs such as the ones presented above achieve perfect reconstructions, in the sense that encoding and then decoding any $x \in \mathcal{M}$ recovers $x$. This result justifies using VAEs as autoencoders despite the fact that they suffer from manifold overfitting. Additionally, Dai & Wipf (2019) also show that only $d^*$ latent dimensions are needed to achieve these perfect reconstructions, suggesting that the posterior over the remaining $d - d^*$ latent dimensions defaults to the standard Gaussian prior $p^Z$ due to the KL term in Equation 24. In other words, the manifold lens helps elucidate why posterior collapse happens.

---

We now formalize the discussion on the results of Dai & Wipf (2019), which assume Gaussian VAEs as described above, with the decoder covariance given by $\Sigma_\theta^{X|Z}(z) = \gamma I_D$, where $\gamma > 0$ is a free parameter that does not depend on $z$. They also assume throughout that $\mathcal{M}$ is diffeomorphic to $\mathbb{R}^{d^*}$, which is a much stronger assumption than required by Loaiza-Ganem et al. (2022a).

The first result of Dai & Wipf (2019) assumes some regularity conditions, that $d^* < D$, that $d \geq d^*$, and that $\gamma$ is learnable (i.e. it is part of $\theta$). The result then states that for any distribution $\mathbb{P}_\dagger^X$ on $\mathcal{X}$ supported on $\mathcal{M}$, there exist a sequence of VAE models parameterized by $(\theta_t, \phi_t)_{t=1}^\infty$ such that:

- For every $x \in \mathcal{M}$, it holds that $p_{\theta_t}^X(x) \to \infty$ and $\mathbb{KL}\left(q_{\phi_t}^{Z|X}(\cdot|x) \,\middle\|\, p_{\theta_t}^{Z|X}(\cdot|x)\right) \to 0$ as $t \to \infty$.

  Note that together, these two limits imply not only that $\mathbb{E}_{X \sim \mathbb{P}_*^X}[\log p_{\theta_t}^X(X)] \to \infty$, but also that $\mathcal{E}_{p_*^X}(\theta_t, \phi_t) \to \infty$ as $t \to \infty$.

- $\mathbb{P}^X_{\theta_t} \xrightarrow{\omega} \mathbb{P}^X_{\dagger}$ as $t \to \infty$, where $\mathbb{P}^X_{\theta_t}$ is the distribution corresponding to the model density $p^X_{\theta_t}$.

As previously mentioned, Dai & Wipf (2019) also show that Gaussian VAEs achieve perfect reconstructions, and they link this behaviour to posterior collapse. To do so, they first consider $\gamma > 0$ as a fixed hyperparameter instead of being learnable, and write the corresponding ELBO as $\mathcal{E}_{p^X_*}(\theta, \phi; \gamma)$, with corresponding maximizers $\theta^*(\gamma)$ and $\phi^*(\gamma)$. They then prove that, if $d \geq d^*$ and under similar regularity conditions to the result above, for every $\gamma > 0$ there exists $\gamma' \in (0, \gamma)$ such that:

$$\mathcal{E}_{p^X_*}\left(\theta^*(\gamma'), \phi^*(\gamma'); \gamma'\right) > \mathcal{E}_{p^X_*}\left(\theta^*(\gamma), \phi^*(\gamma); \gamma\right). \tag{27}$$

In particular this suggests that, when $\gamma$ is learnable, it must converge to 0 to make the ELBO diverge to infinity. An intuitive way to understand this is as saying that, for a small enough decoder variance $\gamma$, it is preferable to maximize the $\log p^{X|Z}_{\theta}(X|Z)$ term in Equation 24 – which in this case boils down to an $\ell_2$ reconstruction error weighted by $1/(2\gamma)$ – instead of minimizing the KL term. Dai & Wipf (2019) then leverage this result to show that VAEs achieve perfect reconstructions in the sense that

$$\lim_{\gamma \to 0} g_{\theta^*(\gamma)}\left(f_{\phi^*(\gamma)}(x)\right) = x, \quad \mathbb{P}^X_*\text{-almost-surely.} \tag{28}$$

Finally, since $\mathbb{P}^X_*$ is supported on a $d^*$-dimensional manifold $\mathcal{M}$, perfectly reconstructing a point $x \in \mathcal{M}$ requires $d^*$ dimensions, and not the full $d$ of the latent space $\mathcal{Z}$: this means that $d^*$ latent dimensions are used to achieve perfect reconstructions, and thus the respective approximate posterior variances are sent to 0; whereas the remaining $d - d^*$ dimensions in the approximate posterior default to a standard Gaussian to minimize the KL term in the ELBO. This explanation of posterior collapse was suggested by Dai & Wipf (2019), and it was further formalized by Zheng et al. (2022).

### 4.1.3 Normalizing Flows

Normalizing flows (NFs; Dinh et al., 2015; 2017; Papamakarios et al., 2017; Kingma & Dhariwal, 2018; Durkan et al., 2019; Kobyzev et al., 2020; Papamakarios et al., 2021) are a class of DGMs that leverage the change-of-variables formula to enable maximum-likelihood training. NFs construct a bijective neural network $g_\theta : \mathcal{Z} \to \mathcal{X}$, where $\mathcal{Z} = \mathcal{X} = \mathbb{R}^D$, such that both $g_\theta$ and its inverse $f_\theta$ are differentiable.[4] A prior density $p^Z$ is specified (often a standard Gaussian), and sampling from the model $p^X_\theta$ is achieved through $X = g_\theta(Z)$ where $Z \sim p^Z$ (formally, $p^X_\theta$ is the pushforward of $p^Z$ through $g_\theta$). The change-of-variables formula from Equation 8 provides the maximum-likelihood objective:

$$\max_\theta \mathbb{E}_{X \sim p^X_*} \left[\log p^Z\left(f_\theta(X)\right) + \log |\det \nabla_x f_\theta(X)|\right]. \tag{29}$$

Constructing the Jacobian $\nabla_x f_\theta(x)$ through automatic differentiation to compute the log-likelihood and then backpropagate with respect to $\theta$ is computationally prohibitive. To circumvent this issue, NFs are constructed in such a way that ensures that $\det \nabla_x f_\theta(x)$ can be efficiently evaluated in closed-form (or at least approximated), thus enabling gradient optimization with respect to $\theta$.

A relevant variant of NFs are continuous NFs, where $f_\theta$ is defined implicitly through an ordinary differential equation (ODE) rather than explicitly constructed (Chen et al., 2018; Salman et al., 2018; Grathwohl et al., 2019). More specifically, an auxiliary neural network $v_\theta : \mathcal{X} \times [0, T] \to \mathcal{X}$ is used to specify the ODE:

$$\begin{aligned} \mathrm{d}x_t &= v_\theta(x_t, t)\mathrm{d}t, \\ x_0 &\in \mathcal{X}. \end{aligned} \tag{30}$$

---

[4]Note that in NFs, the "encoder" $f_\theta$ is uniquely determined by the "decoder" $g_\theta$. Thus, the "encoder" does not require auxiliary parameters $\phi$, which is why we parameterize it with the generative parameters $\theta$. We do nonetheless highlight that moving away from this restriction by parameterizing the encoder and decoder networks separately and making them learn to invert each other has been attempted (Draxler et al., 2024).

Under standard regularity conditions this ODE has a unique and smooth solution on $[0, T]$ (Khalil, 2002),[5] i.e. it characterizes the trajectory $(x_t)_{t \in [0,T]}$, and thus implicitly defines a mapping from the initial condition to the final point in the trajectory, namely

$$
\begin{aligned}
f_\theta &: \mathcal{X} \to \mathcal{Z}, \\
x_0 &\mapsto x_T,
\end{aligned}
\tag{31}
$$

where again $\mathcal{Z} = \mathcal{X}$. Furthermore, under the same conditions that guarantee a unique solution to Equation 30, $f_\theta$ is invertible and its inverse $g_\theta$ can be computed by solving the reverse ODE

$$
\begin{aligned}
\mathrm{d}y_t &= -v_\theta(y_t, T - t)\mathrm{d}t, \\
y_0 &= x_T \in \mathcal{Z},
\end{aligned}
\tag{32}
$$

whose solution is the reversed trajectory $(y_t)_{t \in [0,T]} = (x_{T-t})_{t \in [0,T]}$, so that $g_\theta$ corresponds to the map

$$
\begin{aligned}
g_\theta &: \mathcal{Z} \to \mathcal{X}, \\
y_0 &\mapsto y_T.
\end{aligned}
\tag{33}
$$

Like standard NFs, continuous NFs are sampled from by first obtaining $Y_0 \sim p^Z$, and then computing $Y_T = g_\theta(Y_0)$ – which is now done by numerically solving Equation 32 initialized at $y_0 = Y_0$. In this case the log det term in the change-of-variables formula takes the form (Chen et al., 2018)

$$
\log \det \nabla_{x_0} f_\theta(x_0) = \int_0^T \mathrm{tr}\left(\nabla_{x_t} v_\theta(x_t, t)\right) \mathrm{d}t,
\tag{34}
$$

thus enabling maximum-likelihood training of continuous NFs through

$$
\max_\theta \mathbb{E}_{X_0 \sim p_*^X}\left[\log p^Z\left(f_\theta(X_0)\right) + \int_0^T \mathrm{tr}\left(\nabla_{x_t} v_\theta(X_t, t)\right) \mathrm{d}t\right],
\tag{35}
$$

where $X_t$ corresponds to $x_t$ when Equation 30 is initialized at $x_0 = X_0 \sim p_*^X$. While the computations used to train continuous NFs through Equation 35 are significantly different than those used for standard NFs, both models are fundamentally doing the same thing: modelling the data as the distribution obtained by mapping a simple distribution $p^Z$ such as a Gaussian through a bijective function $g_\theta$ (defined explicitly or implicitly), and training the model via maximum-likelihood.

**Normalizing flows through the lens of the manifold hypothesis** By construction, NFs – continuous or not – are full-dimensional models and are thus susceptible to manifold overfitting (Section 4.1). There are however other pathologies associated with using NFs in the manifold setting. For example, Cornish et al. (2020) show that if the supports of $p^Z$ and $p_*^X$ are not homeomorphic,[6] then any normalizing flow which approximates $p_*^X$ must have exploding bi-Lipschitz constant. In the manifold setting, the support of $p_*^X$ is the $d^*$-dimensional manifold $\mathcal{M}$, which is not homeomorphic to the support of $p^Z$, i.e. $\mathbb{R}^D$. The exploding bi-Lipschitz constant implied by the result of Cornish et al. (2020) entails that, if NFs converge to a distribution on a low-dimensional manifold (whether this is $p_*^X$ or some other distribution), they must do so in a numerically unstable way, which is completely consistent with our result in Section 4.1.1. These theoretical insights are borne out in practice; for example, Behrmann et al. (2021) show that trained NFs are numerically non-invertible. While perhaps initially surprising, this phenomenon is neatly explained by considering NFs through the lens of the manifold hypothesis, which we return to in Section 5.3.3.

---

[5]The most notable of these conditions is $v_\theta$ being Lipschitz in $t$ (with the Lipschitz constant not depending on $x$), which will become relevant when we discuss diffusion models in Section 5.1.2.

[6]Recall that two spaces are homeomorphic when there exists a homeomorphism – i.e. a continuous invertible function with continuous inverse – between them.

### 4.1.4 Energy-Based Models

Energy-based models (EBMs; Xie et al., 2016; Du & Mordatch, 2019) are likelihood-based DGMs which construct a density $p_\theta^X$ by specifying it up to proportionality. More specifically, a neural network $E_\theta : \mathcal{X} \to \mathbb{R}$, called the energy function, is used to define $p_\theta^X$ through

$$p_\theta^X(x) \propto e^{-E_\theta(x)}, \tag{36}$$

where $p_\theta^X$ is assumed to be well-defined (i.e. its normalizing constant is presumed finite: $\int_\mathcal{X} e^{-E_\theta(x)}\mathrm{d}x < \infty$). While the likelihood of EBMs is unavailable due to the normalizing constant of $p_\theta^X$ being intractable, the observation that

$$\nabla_\theta \log p_\theta^X(x) = \mathbb{E}_{X \sim p_\theta^X}[\nabla_\theta E_\theta(X)] - \nabla_\theta E_\theta(x) \tag{37}$$

enables gradient optimization of the log-likelihood, since

$$\nabla_\theta \mathbb{E}_{X \sim p_*^X}\left[\log p_\theta^X(X)\right] = \mathbb{E}_{X \sim p_\theta^X}[\nabla_\theta E_\theta(X)] - \mathbb{E}_{X \sim p_*^X}[\nabla_\theta E_\theta(X)], \tag{38}$$

and the expectation with respect to $p_\theta^X$ can be estimated through Markov chain Monte Carlo (MCMC) methods such as Langevin dynamics (Welling & Teh, 2011). In practice, the log-likelihood is maximized by implementing the loss

$$\min_\theta \mathbb{E}_{X \sim p_*^X}[E_\theta(X)] - \mathbb{E}_{X \sim p_{\theta'}^X}[E_\theta(X)], \tag{39}$$

where $\theta' = \texttt{stopgrad}(\theta)$, as it provides the correct gradient with respect to $\theta$.[7]

**Energy-based models through the lens of the manifold hypothesis** Since by construction $p_\theta^X(x) > 0$ for all $x \in \mathcal{X}$, EBMs are full-dimensional models and are thus susceptible to manifold overfitting (Section 4.1). EBM training and sampling are known to be difficult. A common trick to sample from a trained EBM is to initialize MCMC chains not from noise, but from previous chains held in a replay buffer. The buffer is a set of MCMC samples from chains that have been advanced by the EBM throughout the entire training process (Du & Mordatch, 2019; Grathwohl et al., 2020). Alternatively, a mixture of historical training checkpoints can be used for sampling (Du & Mordatch, 2019). These tricks are necessary because EBMs are prone to mode collapse, especially in high-dimensional ambient spaces (Arbel et al., 2021; Loaiza-Ganem et al., 2022a). This mode collapse behaviour is often blamed on Langevin dynamics and the multimodality of the target distribution, but can be further explained by the large or unstable gradients in the density landscape of a model that has undergone manifold overfitting.

One particularly interesting exception is the normalized autoencoder proposed by Yoon et al. (2021), which employs the reconstruction error of an autoencoder to define the energy function; i.e.

$$E_\theta(x) = \frac{\|x - g_\theta(f_\theta(x))\|_2^2}{T}, \tag{40}$$

where $T > 0$ is a hyperparameter.[8] Much like variational autoencoders (Section 4.1.2), despite using an autoencoder-like structure, normalized autoencoders remain full-dimensional models; this is true of EBMs regardless of the choice of energy function. An interesting observation, which to the best of our knowledge has not been previously made, is that the energy function in Equation 40 is lower-bounded by 0, which in turn implies that $e^{-E_{\theta_t}(x)}$ is upper-bounded, meaning that $p_{\theta_t}^X(x)$ cannot be sent to infinity by making $E_{\theta_t}(x)$ arbitrarily negative for $x \in \mathcal{M}$. In particular, manifold overfitting can only occur when the normalizing constant $\int_\mathcal{X} e^{-E_{\theta_t}(x)}\mathrm{d}x$ goes to 0. While it is of course possible for this to happen with arbitrarily flexible networks, we hypothesize that EBMs with lower-bounded energy functions might have an inductive bias which helps them avoid manifold overfitting in practice, albeit likely at the cost of generative quality. This might explain their empirical success at density-based out-of-distribution detection (Yoon et al., 2023).

---

[7]$\texttt{stopgrad}$ is a computational operator, commonly available in automatic differentiation libraries such as PyTorch (Paszke et al., 2019), which leaves the forward pass unchanged ($\theta' = \theta$) while ignoring gradients when differentiating ($\nabla_\theta \theta' = 0$), i.e. $\mathbb{E}_{X \sim p_{\theta'}^X}[E_\theta(X)] = \mathbb{E}_{X \sim p_\theta^X}[E_\theta(X)]$, yet $\nabla_\theta \mathbb{E}_{X \sim p_{\theta'}^X}[E_\theta(X)] = \mathbb{E}_{X \sim p_\theta^X}[\nabla_\theta E_\theta(X)]$ even though $\nabla_\theta \mathbb{E}_{X \sim p_\theta^X}[E_\theta(X)]$ is not in general equal to $\mathbb{E}_{X \sim p_\theta^X}[\nabla_\theta E_\theta(X)]$.

[8]Note that the distribution of normalized autoencoders depends on the encoder $f_\theta$, which is why we parameterize it with $\theta$ instead of auxiliary parameters $\phi$; this encoder need not share parameters with the decoder $g_\theta$.

### 4.2 Generative Adversarial Networks

Generative adversarial networks (GANs; Goodfellow et al., 2014; Radford et al., 2015) use a neural network $g_\theta : \mathcal{Z} \to \mathcal{X}$ called the generator, along with a latent distribution $p^Z$ (usually a standard Gaussian), to specify the model distribution $p_\theta^X$. Similarly to normalizing flows (Section 4.1.3), samples $X = g_\theta(Z)$ from a GAN are obtained by sampling $Z \sim p^Z$ and transforming the result through $g_\theta$ (as in NFs, $p_\theta^X$ is formally given by the pushforward of $p^Z$ through $g_\theta$), although unlike NFs, $d = D$ is not required, and rather $d < D$ is the standard choice for GANs, so that $g_\theta$ need not be invertible. GANs are not likelihood-based models, and in order to train $g_\theta$, they introduce a binary classifier $h_\phi : \mathcal{X} \to (0, 1)$, which is trained to distinguish between real samples $X \sim p_*^X$ and generated samples $X \sim p_\theta^X$. The generator $g_\theta$ is trained alongside $h_\phi$, so as to make the classifier unable to successfully differentiate between real and generated samples:

$$\min_\theta \max_\phi \mathbb{E}_{X \sim p_*^X} \left[ \log h_\phi(X) \right] + \mathbb{E}_{Z \sim p^z} \left[ \log \left( 1 - h_\phi(g_\theta(Z)) \right) \right]. \tag{41}$$

Assuming that the classifier is arbitrarily flexible, it can be shown that the above objective is equivalent to minimizing the Jensen-Shannon divergence, $\mathbb{JS}$, between the true distribution and the model,

$$\min_\theta \mathbb{JS} \left( p_*^X \, \| \, p_\theta^X \right), \tag{42}$$

where $\mathbb{JS}(p \, \| \, q) \coloneqq \frac{1}{2} \mathbb{KL}(p \, \| \, \frac{1}{2}p + \frac{1}{2}q) + \frac{1}{2} \mathbb{KL}(q \, \| \, \frac{1}{2}p + \frac{1}{2}q)$. This result was first shown by Goodfellow et al. (2014) under the unstated assumption that $p_\theta^X$ and $p_*^X$ are densities supported on the same manifold for all $\theta$. While this assumption is unrealistic in the manifold setting and should not be expected to hold in practice, the proof provided by Goodfellow et al. (2014) is "correct in spirit", and was later formalized and generalized by Donahue et al. (2017).

**Generative adversarial networks through the lens of the manifold hypothesis** The Jensen-Shannon divergence $\mathbb{JS}(p \, \| \, q)$ is meaningfully defined even when $\mathbb{KL}(p \, \| \, q) = \infty$. At a first glance this might suggest that optimizing the Jensen-Shannon divergence circumvents issues such as manifold overfitting (Section 4.1), which arise from attempting to minimize the KL divergence. However, the gradients of the Jensen-Shannon divergence $\mathbb{JS}(p_*^X \, \| \, p_\theta^X)$ with respect to model parameters $\theta$ will be 0 whenever the supports of $p_*^X$ and $p_\theta^X$ do not overlap (formally, the Jensen-Shannon divergence does not metrize weak convergence). This property makes gradient optimization futile whenever the support of the GAN has no overlap with the underlying data manifold, which Arjovsky et al. (2017) identify as a cause of training instabilities for GANs. The Jensen-Shannon divergence is a particular instance of an $f$-divergence (Polyanskiy & Wu, 2022), and there is work generalizing GANs to minimize $f$-divergences (Nowozin et al., 2016). Yet, Arjovsky et al. (2017) also show that various other $f$-divergences, such as the total variation distance and KL divergence, suffer from similar pathologies as the Jensen-Shannon divergence when there is mismatch between the supports of $p_\theta^X$ and $p_*^X$. In other words, although GANs do not suffer from manifold overfitting, they can still struggle to model manifold-supported data. It is however worth highlighting that the manifold-related woes of GANs are fundamentally different than those of likelihood-based models: the former use a proper low-dimensional model (whenever $d < D$), and the resulting problems are due only to the optimization objective; whereas the latter are full-dimensional models, and are thus misspecified. Still, GANs can remain topologically misspecified, e.g. when $\mathcal{M}$ is disconnected (Section 5.4.3), but again, this is an inherently different situation than the dimensional misspecification of likelihood-based models.

### 4.3 Score Matching

Score matching (Hyvärinen, 2005) is a method to learn full-dimensional densities $p_*^X$. The main idea is to learn the (Stein) score function, $\nabla_x \log p_*^X$, rather than $p_*^X$ itself.[9] In order to achieve this, a model $p_\theta^X$ is implicitly characterized by $s_\theta^X : \mathcal{X} \to \mathcal{X}$, whose goal is to approximate the unknown true score function. The Fisher divergence, which is sometimes referred to as the Fisher information distance (DasGupta, 2008), and which is defined as

$$\mathbb{F}(p, q) \coloneqq \mathbb{E}_{X \sim p} \left[ \| \nabla_x \log q(X) - \nabla_x \log p(X) \|_2^2 \right], \tag{43}$$

---

[9] While in machine learning $\nabla_x \log p_\theta^X$ is often called the score function of a model, in the statistics literature the score function refers to $\nabla_\theta \log p_\theta^X$, whereas $\nabla_x \log p_\theta^X$ is called the Stein score.

is leveraged for this goal. Ideally the model would be trained by minimizing $\mathbb{F}(p_*^X, p_\theta^X)$ as

$$\min_\theta \mathbb{E}_{X \sim p_*^X} \left[ \|s_\theta^X(X) - \nabla_x \log p_*^X(X)\|_2^2 \right], \tag{44}$$

but naïvely doing so requires evaluating the unknown $\nabla_x \log p_*^X$. Hyvärinen (2005) showed that, under mild regularity conditions,

$$\mathbb{F}(p, q) = \mathbb{E}_{X \sim p} \left[ \|\nabla_x \log q(X)\|_2^2 + 2 \operatorname{tr} \nabla_x^2 \log q(X) \right] + c(p), \tag{45}$$

where $c(p)$ is a term which depends only on $p$. Since $c(p_*^X)$ is a constant with respect to $\theta$, the objective in Equation 44 is thus equivalent to

$$\min_\theta \mathbb{E}_{X \sim p_*^X} \left[ \|s_\theta^X(X)\|_2^2 + 2 \operatorname{tr} \nabla_x s_\theta^X(X) \right], \tag{46}$$

which can actually be minimized. Once a model is trained, Markov chain Monte Carlo methods such as Langevin dynamics can be used to sample from it, similarly to energy-based models (Section 4.1.4).

**Score matching through the lens of the manifold hypothesis**  As mentioned above, score matching is derived under the assumption that the underlying data distribution $p_*^X$ is full-dimensional. While score matching has been extended to known manifolds (Mardia et al., 2016), we are not aware of any work theoretically studying dimensional mispecification within score matching in an analogous manner to how Loaiza-Ganem et al. (2022a) characterize manifold overfitting (Section 4.1) within likelihood-based models. Nonetheless, we should intuitively expect score matching to fail under the manifold setting due to this dimensional misspecification. To see why this is the case, we begin by noting that $p_\theta^X$ is indeed full-dimensional since $s_\theta^X$ takes inputs from all of $\mathcal{X}$ rather than just $\mathcal{M}$ ($s_\theta^X$ is evaluated at potentially any point in $\mathcal{X}$ during sampling when using procedures such as Langevin dynamics). The score functions $s_\theta^X$ and $\nabla_x \log p_*^X$ are thus different types of objects – the former is a full-dimensional score function and the latter is a manifold-supported one. Comparing the values of dimensionally-mismatched densities is not meaningful, and the comparison remains equally meaningless between the corresponding score functions. Consequently, there is no reason to expect Equation 44 to succeed at matching $p_\theta^X$ to $p_*^X$ in the presence of dimensional misspecification. This issue was identified by Song & Ermon (2019), who empirically confirm that score matching struggles in the manifold setting.

## 5 Manifold-Aware Deep Generative Models

As covered throughout Section 4, many commonly-used DGMs struggle to learn distributions on unknown manifolds. There are various (not always mutually exclusive) approaches that enable manifold-awareness, including judiciously adding noise to the target distribution; using support-agnostic optimization objectives (e.g. those which metrize weak convergence); and two-step models, which carry out generative modelling on a low-dimensional latent space and then map back to data space. We review these approaches in Section 5.1, Section 5.2, and Section 5.3, respectively. In Section 5.3.1 we show that (*i*) two-step models can be interpreted as (potentially regularized) minimizers of an upper bound of the Wasserstein distance, thus establishing a link between these different approaches for achieving manifold-awareness, and that (*ii*) the upper bound becomes tight at optimality whenever an autoencoder can achieve perfect reconstructions. Finally, in Section 5.4 we cover methods which make an explicit attempt at properly capturing the topology of $\mathcal{M}$. We take a lax interpretation of manifold-awareness throughout, and discuss not only DGMs which are formally manifold-aware, but also those which, while mathematically manifold-unaware, leverage some inductive bias towards manifold-awareness.

### 5.1 Manifold-Awareness by Adding Noise

When manifold-unawareness arises due to the mismatch between the dimension of the model and that of the true distribution – as is the case for likelihood-based models (Section 4.1) – adding noise to the training data seems like a natural solution; this can make the target distribution full-dimensional (e.g. by

convolving the true distribution with a Gaussian) and thus hopefully avoids manifold-related problems. Indeed, dequantization – i.e. the practice of adding noise to data that was discretized so as to be able to fit a continuous density model – is very common (Theis et al., 2016; Dinh et al., 2017; Ho et al., 2019), and can be further justified as a way to avoid manifold overfitting. Unfortunately, it has been shown that just adding Gaussian noise is not enough to empirically avoid manifold overfitting (Zhang et al., 2020a; Loaiza-Ganem et al., 2022a;b). Even though the theoretical conditions for manifold overfitting do not hold anymore, the new (noisy) target density will be extremely peaked around $\mathcal{M}$ (Section 4.1.1), and thus still numerically exposed to manifold-related woes. This observation is consistent with known convergence rates at which DGMs trained on noisy data recover $p_*^X$ (Chae et al., 2023). The lesson here is that adding noise can enable DGMs to learn unknown manifolds, but the noise has to be added carefully. Various methods doing so have been proposed, which we now review.

### 5.1.1  Denoising Score Matching

As previously mentioned, Song & Ermon (2019) showed that score matching (Section 4.3) struggles to model manifold-supported data, and they thus advocate for adding noise and using denoising score matching (Vincent, 2011) instead. In denoising score matching, the target distribution is not $p_*^X$ anymore, but rather the distribution $p_*^{X_\sigma}$ obtained by adding independent Gaussian noise $\mathcal{N}(\,\cdot\,;0,\sigma^2 I_D)$ to samples from $p_*^X$, where $\sigma^2$ is a hyperparameter. More formally, $p_*^{X_\sigma} := p_*^X \circledast \mathcal{N}(\,\cdot\,;0,\sigma^2 I_D)$. Importantly, adding full-dimensional Gaussian noise ensures that $p_*^{X_\sigma}$ is always full-dimensional, regardless of the support of $p_*^X$. Score matching can then be applied to learn a network $s_\theta^X$ to approximate $\nabla_{x_\sigma} \log p_*^{X_\sigma}$ through Equation 46 (with $p_*^X$ replaced by $p_*^{X_\sigma}$). However, Vincent (2011) shows that this objective is equivalent to

$$\min_\theta \mathbb{E}_{X_0 \sim p_*^X}\left[\mathbb{E}_{X_\sigma \sim p_*^{X_\sigma|X_0}(\cdot|X_0)}\left[\|s_\theta^X(X_\sigma) - \nabla_{x_\sigma}\log p_*^{X_\sigma|X_0}(X_\sigma|X_0)\|_2^2\right]\right], \tag{47}$$

where $p_*^{X_\sigma|X_0}(x_\sigma|x_0) = \mathcal{N}(x_\sigma;x_0,\sigma^2 I_D)$ is the density of noisy data (denoted $X_\sigma$) given the (un-noised) datapoint $X_0 = x_0$. Equation 47 is much easier to optimize than the usual score matching objective (Equation 46), since there is no need to backpropagate through the trace of the Jacobian of $s_\theta^X$. Saremi & Hyvärinen (2019) use the loss function in Equation 47 to learn an energy-based model (Section 4.1.4) on the noised-out data, but derive the loss from the perspective of empirical Bayes (Robbins, 1956); their approach can in principle be applied to other noising processes, but only the Gaussian case is implemented in their work.

Despite mathematically avoiding manifold-related pathologies, denoising score matching as presented above faces a tradeoff; setting $\sigma$ to a very small value means that the target density $p_*^{X_\sigma}$ is closer to the actual manifold-supported data density $p_*^X$, but doing so also means the target density is highly peaked around $\mathcal{M}$ and thus might be harder to properly learn (Section 4.1.1). As a way of being able to use small amounts of noise while still efficiently learning the resulting distribution, Song & Ermon (2019) propose to use various noise levels. More specifically, they consider fixed noise levels $0 < \sigma_1 < \sigma_2 < \cdots < \sigma_T$, and modify the score function to take the noise level as input; i.e. $s_\theta^X : \mathcal{X} \times [\sigma_1, \sigma_T] \to \mathcal{X}$ is now such that it aims to approximate the score function at all the corresponding noise levels: $s_\theta^X(\,\cdot\,,\sigma_t) \approx \nabla_{x_{\sigma_t}}\log p_*^{X_{\sigma_t}}$ for $t = 1,\ldots,T$. This new score function is trained with a weighted sum of the corresponding denoising score matching objectives,

$$\min_\theta \sum_{t=1}^T w(t)\mathbb{E}_{X_0 \sim p_*^X}\left[\mathbb{E}_{X_{\sigma_t} \sim p_*^{X_{\sigma_t}|X_0}(\cdot|X_0)}\left[\|s_\theta^X(X_{\sigma_t},\sigma_t) - \nabla_{x_{\sigma_t}}\log p_*^{X_{\sigma_t}|X_0}(X_{\sigma_t}|X_0)\|_2^2\right]\right], \tag{48}$$

where $w(t) > 0$ is a pre-specified weight coefficient which aims to keep the $T$ terms in the sum at roughly equal magnitudes. The intuition behind using varying noise levels is twofold. $(i)$ Learning the score function for larger values of $\sigma$ is easier, and thanks to parameter sharing ($\theta$ is the same for all noise levels), doing so is helpful for learning the score function for small values of $\sigma$. $(ii)$ Once the model is trained, different noise levels are also used within an annealed sampling scheme. $s_{\theta^*}^X(\,\cdot\,,\sigma_T)$ is used alongside Markov chain Monte Carlo to generate a sample, which is then used to initialize another Markov chain that now uses $s_{\theta^*}^X(\,\cdot\,,\sigma_{T-1})$; this process is repeated until $s_{\theta^*}^X(\,\cdot\,,\sigma_1)$ is used – and works much better than only using $s_{\theta^*}^X(\,\cdot\,,\sigma_1)$. Although this scheme produces approximate samples from $p_*^{X_{\sigma_1}}$ rather than from $p_*^X$, as long as $\sigma_1$ is small enough, the difference is negligible in practice.

### 5.1.2 Score-Based Diffusion Models

Song et al. (2021b) proposed score-based diffusion models as an extension of denoising score matching (Section 5.1.1) where there is a continuum of noise levels. Formally, they achieve this by constructing $(X_t)_{t\in[0,T]}$, where $X_t \in \mathcal{X}$ for every $t \in [0,T]$, as an Ornstein–Uhlenbeck process given by the Itô stochastic differential equation (SDE),[10]

$$\mathrm{d}X_t = -\frac{\beta(t)}{2}X_t\mathrm{d}t + \sqrt{\beta(t)}\mathrm{d}B_t,$$
$$X_0 \sim p_*^X, \tag{49}$$

where $\beta : [0,T] \to \mathbb{R}_+$ is a hyperparameter (often an affine function, i.e. $\beta(t) = \beta_{\min} + (\beta_{\max} - \beta_{\min})t/T$, where $0 < \beta_{\min} < \beta_{\max}$), and $(B_t)_{t\in[0,T]}$ denotes a $D$-dimensional Brownian motion. Other choices of SDE are possible, but we focus on the one above – which is often referred to as a *variance preserving* SDE – since it is assumed in some of the theoretical results that we will shortly discuss. Under mild regularity conditions, the SDE admits a unique solution (Øksendal, 2003), thus characterizing the density $p_*^{X_t}$ of $X_t$ for every $t \in [0,T]$, and prescribes how to progressively transform the data density $p_*^{X_0} = p_*^X$ into the noisier density $p_*^{X_T}$. Note that, due to the added Gaussian noise (from the Brownian motion in Equation 49), $p_*^{X_t}$ is a full-dimensional density for every $t \in (0,T]$ *regardless of the support* of $p_*^X$.

Reversing $(X_t)_{t\in[0,T]}$ provides a way to transform samples from $p_*^{X_T}$ into samples from $p_*^X$. The reverse process $(Y_t)_{t\in[0,T]} := (X_{T-t})_{t\in[0,T]}$ also obeys an SDE (Anderson, 1982; Haussmann & Pardoux, 1986):[11]

$$\mathrm{d}Y_t = \beta(T-t)\left(\frac{Y_t}{2} + \nabla_{y_t}\log p_*^{X_{T-t}}(Y_t)\right)\mathrm{d}t + \sqrt{\beta(T-t)}\mathrm{d}B_t,$$
$$Y_0 \sim p_*^{X_T}. \tag{50}$$

The main idea of score-based diffusion models is to leverage this reverse SDE to build a generative model. In order to achieve this, some approximations are needed. First, since $p_*^{X_T}$ is not known exactly, Equation 50 is initialized at a known distribution $p_*^{X_\infty}$. Formally, $(Y_t)_{t\in[0,T]}$ is approximated by $(\tilde{Y}_t)_{t\in[0,T]}$, where

$$\mathrm{d}\tilde{Y}_t = \beta(T-t)\left(\frac{\tilde{Y}_t}{2} + \nabla_{\tilde{y}_t}\log p_*^{X_{T-t}}(\tilde{Y}_t)\right)\mathrm{d}t + \sqrt{\beta(T-t)}\mathrm{d}B_t,$$
$$\tilde{Y}_0 \sim p_*^{X_\infty}. \tag{51}$$

We denote the density of $\tilde{Y}_t$ as $\tilde{p}^{X_{T-t}}$, and will shortly explain how $p_*^{X_\infty}$ is chosen so as to be close to $p_*^{X_T}$. The score $\nabla_{\tilde{y}_t}\log p_*^{X_{T-t}}(\tilde{Y}_t)$ is also unknown, and thus must be approximated as well. Score-based diffusion models leverage neural networks to construct $s_\theta^X : \mathcal{X} \times (0,T] \to \mathcal{X}$ with the goal of approximating this function, i.e. $s_\theta^X(x,t) \approx \nabla_x\log p_*^{X_t}(x)$ for all $x \in \mathcal{X}$ and $t \in (0,T]$. We will also soon explain how this network is trained, but for a given $s_\theta^X$, $(\tilde{Y}_t)_{t\in[0,T]}$ is approximated by $(\hat{Y}_t)_{t\in[0,T]}$, where

$$\mathrm{d}\hat{Y}_t = \beta(T-t)\left(\frac{\hat{Y}_t}{2} + s_\theta^X(\hat{Y}_t, T-t)\right)\mathrm{d}t + \sqrt{\beta(T-t)}\mathrm{d}B_t,$$
$$\hat{Y}_0 \sim p_*^{X_\infty}. \tag{52}$$

We denote the density of $\hat{Y}_t$ as $\hat{p}_\theta^{X_{T-t}}$. Ideally, a model sample would be obtained by perfectly solving Equation 52. In practice this SDE must be discretized and a numerical solver must be used. The model distribution $p_\theta^X$ is thus given by the approximate solution of Equation 52 at time $T$.

---

[10]Readers unfamiliar with SDEs can understand Equation 49 through its Euler-Maruyama discretization: split $[0,T]$ into $n$ sub-intervals of equal length $\Delta_{t,n} = T/n$, sample $X_{0,n} \sim p_*^X$, and set $X_{t_{k+1},n} = X_{t_k,n} - \frac{\beta(t_k)}{2}X_{t_k,n}\Delta_{t,n} + \sqrt{\beta(t_k)}\Delta_{B_{t_k},n}$ for $k = 0,\ldots,n-1$, where $t_k = k\Delta_{t,n}$, and where $\Delta_{B_{t_k},n} = B_{t_{k+1}} - B_{t_k} \sim \mathcal{N}(\,\cdot\,;0,\Delta_{t,n}I_D)$ are independent. This procedure characterizes $X_{t,n}$ at the times $t_k$ for $k = 0,\ldots,n$, and linearly interpolating between them yields a continuous stochastic process $(X_{t,n})_{t\in[0,T]}$, the limit of which as $n \to \infty$ corresponds to the process specified by the SDE.

[11]Note that the Brownian motions in Equation 49 and Equation 50 are not in general the same Brownian motion (they just have the same distribution), but we do not differentiate between them for notational simplicity. Note also that Equation 50 differs from the corresponding equation in (Song et al., 2021b) since $\mathrm{d}t$ in our notation corresponds to $-\mathrm{d}t$ in theirs.

In summary, diffusion models aim to solve Equation 50, since perfectly doing so would yield samples from $p_*^X$, but this is impossible and three sources of error have to be introduced to approximately solve this equation. $(i)$ $p_*^{X_T}$ is unknown, and is thus approximated by $p_*^{X\infty}$; $(ii)$ the true score $\nabla_x \log p_*^{X_{T-t}}(x)$ is also unknown, and is thus approximated by $s_\theta^X(x, T-t)$; and $(iii)$ the resulting SDE in Equation 52 must be solved numerically, inducing discretization error.

We have not yet discussed how diffusion models are trained. Before doing so, we point out that Equation 49 has the known transition kernel

$$p_*^{X_t|X_0}(x_t|x_0) = \mathcal{N}\left(x_t; \sqrt{1-\sigma_t^2}x_0, \sigma_t^2 I_D\right), \tag{53}$$

where $p_*^{X_t|X_0}(\cdot|x_0)$ is the conditional density of $X_t$ given that $X_0 = x_0$, and where

$$\sigma_t^2 = 1 - e^{-\int_0^t \beta(s)\mathrm{d}s}. \tag{54}$$

Thanks to Equation 53, $\nabla_{x_t} \log p_*^{X_t|X_0}(x_t|x_0)$ can be evaluated, and sampling from $p_*^{X_t|X_0}(\cdot|x_0)$ is very straightforward. Together, these points imply that denoising score matching (Section 5.1.1) provides a tractable objective for training diffusion models:[12]

$$\min_\theta \int_0^T w(t)\mathbb{E}_{X_0 \sim p_*^X}\left[\mathbb{E}_{X_t \sim p_*^{X_t|X_0}(\cdot|X_0)}\left[\|s_\theta^X(X_t, t) - \nabla_{x_t} \log p_*^{X_t|X_0}(X_t|X_0)\|_2^2\right]\right]\mathrm{d}t, \tag{55}$$

where $w : [0, T] \to \mathbb{R}_+$ is a weighting function (set as a hyperparameter).

Additionally, as long as $\beta$ is such that $\sigma_T^2 \to 1$ as $T \to \infty$, Equation 53 also implies that $p_*^{X_T|X_0}(\cdot|x_0)$ stops depending on $x_0$ in the sense that it converges to $\mathcal{N}(\cdot; 0, I_D)$ as $T \to \infty$: this observation provides an avenue for approximately sampling from $p_*^{X_T}$, namely by setting $p_*^{X\infty}$ to a standard Gaussian.

Finally, Song et al. (2021b) also show that the SDE in Equation 49 is intimately linked to the ODE

$$\mathrm{d}x_t = -\frac{\beta(t)}{2}\left(x_t + \nabla_{x_t} \log p_*^{X_t}(x_t)\right)\mathrm{d}t, \tag{56}$$
$$x_0 \in \mathcal{X}.$$

These equations are related in that, under some regularity conditions, if the ODE is initialized at $x_0 = X_0 \sim p_*^X$, then $x_T$ will have the same distribution as $X_T$. Equation 56 can of course not be solved because the true score function is unknown, but it can be approximated by replacing it with the learned score function, resulting in the new ODE:

$$\mathrm{d}\hat{x}_t = -\frac{\beta(t)}{2}\left(\hat{x}_t + s_{\theta^*}^X(\hat{x}_t, t)\right)\mathrm{d}t, \tag{57}$$
$$\hat{x}_0 \in \mathcal{X}.$$

This equation allows us to interpret diffusion models as continuous normalizing flows (Section 4.1.3): $v_{\theta^*}(x, t)$ in Equation 30 is given by $-\frac{\beta(t)}{2}(x + s_{\theta^*}^X(x, t))$. The connection between diffusion models and continuous NFs has two relevant consequences. $(i)$ It allows for an alternative way of sampling from them: instead of solving Equation 52, the ODE in Equation 57 can be reversed in time as in Equation 32 to obtain

$$\mathrm{d}\hat{y}_t = \frac{\beta(T-t)}{2}\left(\hat{y}_t + s_{\theta^*}^X(\hat{y}_t, T-t)\right)\mathrm{d}t, \tag{58}$$
$$\hat{y}_0 \in \mathcal{X}.$$

---

[12]While the target conditional densities in Equation 48 and Equation 55 – i.e. $\mathcal{N}(x_{\sigma_t}; x_0, \sigma_t^2 I_D)$ and $\mathcal{N}(x_t; \sqrt{1-\sigma_t^2}x_0, \sigma_t^2 I_D)$, respectively – differ in that the mean of the latter is scaled by $\sqrt{1-\sigma_t^2}$, the result of Vincent (2011) which justifies denoising score matching can be easily adapted to this scaled setting, so that Equation 55 is minimized when $s_\theta^X(\cdot, t)$ matches the true score function $\nabla_{x_t} \log p_*^{X_t}$.

Initializing this ODE at $\hat{y}_0 = \hat{Y}_0 \sim p_*^{X_\infty}$ (which plays the role of $p^Z$ in continuous NFs) and solving it will then result in samples from $p_*^X$ if the score function was properly learned, and provided that Equation 57 and Equation 58 admit unique solutions which are inverses of each other. (*ii*) The connection to continuous NFs is also used to justify using the change-of-variables formula (Equation 34) for evaluating the density $\hat{p}_\theta^{X_0}$ implicitly defined by $s_\theta^X$.

**Diffusion models through the lens of the manifold hypothesis**    There are deep connections between diffusion models and manifolds. First, score-based diffusion models are linked to maximum-likelihood. For example, Sohl-Dickstein et al. (2015) and Ho et al. (2020) formulate diffusion models not through SDEs, but through variational inference. In this formulation, the time interval $[0, T]$ is discretized, $X_0$ still corresponds to data, and $X_t$ for $t > 0$ is treated as a latent variable. Then, the (discretized) forward process from Equation 49 corresponds to a fixed variational approximation to the posterior distribution (of latents given data), and the backward process from Equation 52 provides a likelihood term (of data given latents). The resulting model, which is reminiscent of a variational autoencoder (Section 4.1.2), can be trained either by maximizing an ELBO (similar to Equation 24) or a reweighted version of it. Song et al. (2021a) establish another connection between score-based diffusion models and maximum-likelihood: under the assumption that $p_*^X$ is a full-dimensional density (which does not hold in the manifold setting) and some other regularity conditions, the denoising score matching objective from Equation 55, with a specific choice of weighting function $w$, becomes equivalent to minimizing an upper bound of $\mathbb{KL}(p_*^X \,\|\, \hat{p}_\theta^{X_0})$ which becomes tight at optimality.[13] In other words, when $p_*^X$ is full-dimensional and for a particular choice of $w$, diffusion models are likelihood-based models (provided that one ignores the approximation error between $p_*^{X_T}$ and $p_*^{X_\infty}$, and the discretization error, but intuitively both of these errors can be made small by choosing a large $T$ and making the discretization sufficiently fine, respectively).

Naïvely, both of these views of diffusion models suggest that they are a likelihood-based method and thus susceptible to manifold overfitting (Section 4.1). However, a competing intuition is that the denoising carried out through the backward SDE (Equation 50 or its approximation Equation 52) effectively projects noisy data onto $\mathcal{M}$ by removing the noise (Kadkhodaie & Simoncelli, 2021). Fortunately, it is the latter intuition that turns out to be correct: note that diffusion models do not only aim to learn the true manifold-supported data density $p_*^X = p_*^{X_0}$, but also its noisy full-dimensional versions $p_*^{X_t}$ for every $t \in (0, T]$. Indeed, the objective in Equation 55 recovers the score function for all $t \in (0, T]$, and stopping the reverse process from Equation 50 at time $T - t$ instead of $T$ results in a sample from $p_*^{X_t}$. Since $p_*^{X_t}$ is full-dimensional for $t > 0$, we should not expect maximum-likelihood to fail at learning it, and by continuity of the solutions of Equation 50, we should expect $p_*^{X_0}$ to be properly learned, *even under the manifold setting.* Pidstrigach (2022) formalizes this intuition, proving that under mild assumptions, $\tilde{p}^{X_0}$ and $p_*^X$ have the same support, and that $\mathbb{KL}(p_*^X \,\|\, \tilde{p}^{X_0}) \leq \mathbb{KL}(p_*^{X_T} \,\|\, p_*^{X_\infty})$.[14] Since $\tilde{p}^{X_0}$ is the distribution of the model under the assumptions of no discretization error and having perfectly recovered the true score function (justified by the nonparametric regime assumption from Section 2.2), the fact that $\mathbb{KL}(p_*^{X_T} \,\|\, p_*^{X_\infty}) \to 0$ as $T \to \infty$ then implies that perfectly trained score-based diffusion models properly learn $p_*^X$. Pidstrigach (2022) also shows that if the score function is well approximated, in the sense that $\|s_{\theta^*}^X(x, t) - \nabla_x \log p_*^{X_t}(x)\|_2$ is upper-bounded (with the bound not depending on $x$ nor $t$), then $\hat{p}_{\theta^*}^{X_0}$ has the same support as $p_*^X$, i.e. the model (assuming no discretization error) has the same support as the data. Although this result does not guarantee that diffusion models recover their target distribution, it does ensure that they recover the correct manifold, even if there is some error in the learned score function. De Bortoli (2022) further refined these results, finding an upper bound for $\mathbb{W}_1(p_*^X, p_{\theta^*}^X)$ under reasonable assumptions (recall that $p_\theta^X$ corresponds to the approximate solution of Equation 52 at time $T$). This upper bound depends on $T$, the error between the true and modelled score functions, and the step size of the discretization used to solve Equation 52; the upper bound goes to 0 as these quantities go to $\infty$, 0, and 0 at appropriate rates, respectively. All of this entails that score-based diffusion models can learn distributions on unknown manifolds.

---

[13]It is worthwhile to highlight that Kwon et al. (2022) proved a similar result with Wasserstein distance (Section 3.5), namely that the denoising score matching objective used to train diffusion models provides, up to scaling and constant factors, an upper bound of the $\mathbb{W}_2$ distance between the model and $p_*^X$ which becomes tight at optimality. Unfortunately, Kwon et al. (2022) also assume $p_*^X$ is full-dimensional, and thus their result cannot be used to justify the manifold-awareness of diffusion models.

[14]Note that even though $p_*^X$ and $\tilde{p}^{X_0}$ are not full-dimensional densities, the KL divergence between them is meaningfully defined because they have the same support (Section 3.4).

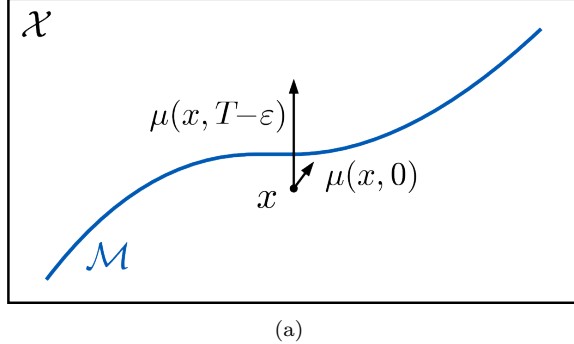
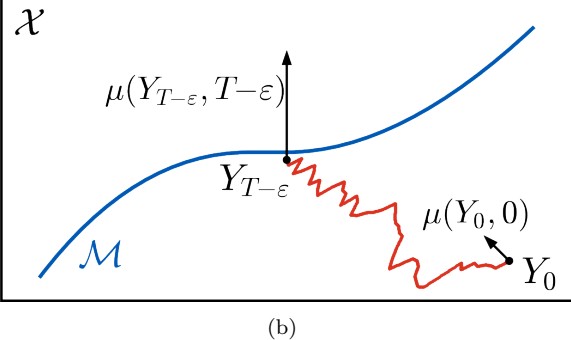

(a)                                          (b)

Figure 6: **(a)** Informal illustration of why the score function explodes. Consider $Y_t = x$ for some fixed $x$ outside of $\mathcal{M}$ as $t$ increases from 0 to $T$. Since diffusion models learn manifolds, $Y_T$ must be in $\mathcal{M}$. Thus, when $t$ gets "infinitesimally" close to $T$, the diffusion must push $x$ onto $\mathcal{M}$ by moving it in the direction of the drift term in Equation 50, i.e. $\mu(Y_t, t) := \beta(T-t)(Y_t/2 + \nabla_{y_t} \log p_*^{X_{T-t}}(Y_t))$, but using only an "infinitesimally" small step size. Since $x$ is not "infinitesimally" close to $\mathcal{M}$, it needs to move a "non-infinitesimal" amount in an "infinitesimal" time step: the only way in which this can happen is if the norm of the drift, and thus of the score as well, explodes to infinity. **(b)** In practice, the norm of the score function diverges not only for fixed points ($\|s_{\theta^*}^X(x, T-t)\|_2 \to \infty$), but also along generated trajectories ($\|s_{\theta^*}^X(Y_t, T-t)\|_2 \to \infty$).

Pidstrigach (2022) also argues that to properly learn the manifold the score $\nabla_x \log p_*^{X_{T-t}}(x)$ must explode to infinity at time $T$, i.e. $\|\nabla_x \log p_*^{X_{T-t}}(x)\|_2 \to \infty$ as $t \to T^-$ for every $x$ outside of $\mathcal{M}$. This behaviour, which we illustrate in Figure 6(a), is easy to understand intuitively: Equation 50 results in $Y_T \sim p_*^X$, which must be in $\mathcal{M}$ because, as discussed in the previous paragraph, diffusion models learn manifolds. Now, fix $x$ outside of $\mathcal{M}$. Since $Y_t$ is full-dimensional for every $t < T$, it follows that $x$ is in its support, so that $x$ is a possible value for $Y_t$. This means that when $t$ is "infinitesimally close" to $T$, the norm of the drift in Equation 50, i.e. $\beta(T-t)\|x/2 + \nabla_x \log p_*^{X_{T-t}}(x)\|_2$, must be "infinitely large" to move $x$ to $\mathcal{M}$, since $x$ is outside of $\mathcal{M}$. In other words, the score must explode to infinity. It follows that if $s_{\theta^*}^X(x, T-t)$ closely matches $\nabla_x \log p_*^{X_{T-t}}(x)$, it must also diverge as $t \to T^-$. Lu et al. (2023) formalize this intuition, showing that under some regularity conditions, not only does the score function explode when $d^* < D$, but also that it does not when $d^* = D$. This phenomenon justifies an often used parameterization of score functions: rather than directly taking $s_\theta^X$ as a neural network (which would be continuous on the compact interval $[0, T]$ and would thus achieve its maximum, making it impossible for the function to diverge and approximate the true score function), $s_\theta^X$ is taken as $s_\theta^X(x, t) = \hat{s}_\theta^X(x, t)/\sigma_t$, where $\hat{s}_\theta^X$ is the neural network. Empirically, it has been observed that diffusion models produce trajectories with exploding score functions (Figure 6(b)), i.e. that $\|s_{\theta^*}^X(Y_t, T-t)\|_2 \to \infty$ as $t \to T^-$ (Kim et al., 2022), which explains the common practice of not running Equation 52 until time $T$, and instead stopping at time $T - \varepsilon$ for some small $\varepsilon > 0$ (Vahdat et al., 2021). The results of Pidstrigach (2022) and Lu et al. (2023) showing that the score is unbounded in the manifold setting thus strongly hint at why these trajectories have exploding scores.[15]

We also highlight that there is no contradiction between diffusion models being able to learn distributions on unknown manifolds and the fact that they can be interpreted as continuous normalizing flows (which cannot do so because of manifold overfitting, see Section 4.1); the former are trained through denoising score matching instead of maximum-likelihood. We note as well that the function $v_{\theta^*}(x, t) = -\frac{\beta(t)}{2}(x + s_{\theta^*}^X(x, t))$ which allows us to interpret diffusion models as continuous NFs through the ODE in Equation 30 is not Lipschitz in $t$ when the score function blows up to infinity at time 0. In turn, there is no immediate guarantee that Equation 57 has a unique solution, so that diffusion models need not be properly defined as continuous NFs; this is of course consistent with the numerical instabilities which render their likelihoods unreliable (Section 4.1.1 and Section 4.1.3).

---

[15]Note that the result that $\|s_{\theta^*}^X(x, T-t)\|_2 \to \infty$ as $t \to T^-$ for every $x$ outside of $\mathcal{M}$ does not formally imply that $\|s_{\theta^*}^X(Y_t, T-t)\|_2 \to \infty$ with probability 1, even if $Y_t$ is outside of $\mathcal{M}$ for $t < T$, since $Y_t$ converges to $Y_T \in \mathcal{M}$. Nevertheless the score function exploding at fixed points shows it is unbounded, which is necessary for the trajectories to blow up as well.

Overall, we believe this view of score-based diffusion models through the manifold setting is particularly illuminating, as it justifies their strong empirical performance (unlike many other popular models, diffusion models can properly learn manifold-supported distributions) and a popular parameterization of the score function, while also explaining its exploding behaviour at time 0.

Lastly, we highlight that the progressive noising of data employed by diffusion models (Equation 49, or a discretized version) has been adopted in contexts beyond denoising score matching: Xiao et al. (2022) combine it with adversarial training (Section 4.2 and Section 5.2.1), and Tran et al. (2023) leverage it for manifold-aware maximum-likelihood training of DGMs.

> Here we briefly formalize some of the conclusions of Pidstrigach (2022) discussed above. The statement that $\tilde{p}^{X_0}$ and $p_*^X$ have the same support is formalized as $\tilde{\mathbb{P}}^{X_0} \ll \mathbb{P}_*^X$ and $\mathbb{P}_*^X \ll \tilde{\mathbb{P}}^{X_0}$, where $\tilde{\mathbb{P}}^{X_0}$ is the probability measure corresponding to $\tilde{p}^{X_0}$. Note that this statement implies that $\tilde{\mathbb{P}}^{X_0}$ and $\mathbb{P}_*^X$ must have the same sets of measure 0, and thus of measure 1 as well, and since the support of a distribution depends only on the sets to which it assigns probability 1 (Equation 1), these two probability measures must have the same support. Similarly, the statement that $\hat{p}_{\theta^*}^{X_0}$ and $p_*^X$ have the same support is formalized as $\hat{\mathbb{P}}_{\theta^*}^{X_0} \ll \mathbb{P}_*^X$ and $\mathbb{P}_*^X \ll \hat{\mathbb{P}}_{\theta^*}^{X_0}$, where $\hat{\mathbb{P}}_{\theta^*}^{X_0}$ is the probability measure corresponding to $\hat{p}_{\theta^*}^{X_0}$.

### 5.1.3 Conditional Flow Matching

Continuous normalizing flows (Section 4.1.3) as defined through Equation 30 were originally trained through maximum-likelihood. As discussed in Section 5.1.2, diffusion models can be interpreted as providing an alternative training objective for continuous NFs. Conditional flow matching (CFM; Liu et al., 2023; Albergo & Vanden-Eijnden, 2023; Lipman et al., 2023) provides additional alternatives, the main variant of which we cover here.

Recall that diffusion models parameterize and learn a vector field ($s_\theta^X$, approximating the score function) by attempting to regress against the true score,

$$\min_\theta \int_0^T w(t) \mathbb{E}_{X_t \sim p_*^{X_t}}[\|s_\theta^X(X_t, t) - \nabla_{x_t} \log p_*^{X_t}(X_t)\|_2^2] \mathrm{d}t, \tag{59}$$

where we will assume $w(t) = 1$ to better highlight their similarities to CFM. As discussed in Section 4.3 and Section 5.1.1, directly optimizing Equation 59 cannot be done in practice because $p_*^{X_t}$ is unknown. However, by conditioning on $X_0$, the equivalent – yet tractable – objective in Equation 55 is obtained.

Consider a "true" vector field $v_* : \mathcal{X} \times [0, T] \to \mathcal{X}$ in the sense that its corresponding forward ordinary differential equation (Equation 30) maps samples from $p_*^X$ at time $t = 0$ to samples from $p^Z$ at $t = T$. Note that even when such a vector field exists, it is not unique. Similarly to diffusion models, CFM learns a vector field $v_\theta$ by attempting to regress against $v_*$, i.e.

$$\min_\theta \int_0^T \mathbb{E}_{X_t \sim p_*^{X_t}}[\|v_\theta(X_t, t) - v_*(X_t, t)\|_2^2] \mathrm{d}t, \tag{60}$$

where $p_*^{X_t}$ now corresponds to the density of $x_t$ if $\mathrm{d}x_t = v_*(x_t, t)\mathrm{d}t$ is initialized at $x_0 = X_0 \sim p_*^X$. Again, Equation 60 cannot be directly optimized because $v_*$, and hence also $p_*^{X_t}$, are unknown. Conditioning is once again the solution, except now conditioning is done on both $X_0$ and $X_T$ which allows for explicit construction of a target vector field that maps $X_0$ to $X_T$. There are many such vector fields, but the simplest is the vector field $(X_T - X_0)/T$ (Tong et al., 2024). Since this vector field is time-independent, sampling from the corresponding $p_*^{X_t}$ becomes easy as one can take $X_t = ((T - t)X_0 + tX_T)/T$, where $X_0 \sim p_*^X$ and $X_T \sim p^Z$ are independently sampled. In summary, $v_\theta$ is trained through

$$\min_\theta \int_0^T \mathbb{E}_{X_0 \sim p_*^X, X_T \sim p^Z}\left[\left\|v_\theta\left(\frac{(T-t)X_0 + tX_T}{T}, t\right) - \frac{X_T - X_0}{T}\right\|_2^2\right] \mathrm{d}t. \tag{61}$$

Remarkably, when $p_*^X$ and $p^Z$ are both full-dimensional, and under some other regularity conditions, the conditional optimization problem in Equation 61 turns out to be equivalent to the unconditional optimization

problem in Equation 60 for a particular $v_*$. Hence, under these conditions, the vector field $v_{\theta*}$ optimized under Equation 61 can still be used for generation with the reverse ODE in Equation 32.

**Conditional flow matching through the lens of the manifold hypothesis**  We begin by pointing out that Lipman et al. (2023) add a small amount of noise to the data while applying CFM, which enforces full-dimensionality of the target density. However, if no noise is added, the known correctness guarantees for CFM break down when $p_*^X$ is supported on a low-dimensional manifold. Kingma & Gao (2023) proved that, surprisingly, the objectives in Equation 55 for diffusion models and Equation 61 for CFM are equivalent given appropriate hyperparameter choices. Since diffusion models are manifold-aware, at first glance this result might seem to imply that CFM must also learn distributions on unknown manifolds. However, this does *not* follow from the result of Kingma & Gao (2023): the manifold-awareness of diffusion models is guaranteed when the backward stochastic differential equation (Equation 52) is used to sample from the model, whereas CFM always uses an ODE (Equation 32).[16]  To the best of our knowledge, there is currently no formal analysis of CFM under the manifold hypothesis analogous to that of Pidstrigach (2022) or De Bortoli (2022) for diffusion models.[17]  Nonetheless, due to its similarities with diffusion models, we conjecture that CFM is indeed manifold-aware, which would be consistent with its strong empirical performance. We do however point out that if CFM successfully learns its target distribution in the manifold setting, then it must experience the numerical instabilities of likelihood evaluation that we established in Section 4.1.1, even if these instabilities do not manifest themselves during training since likelihoods do not appear in Equation 61. Finally, we also highlight the work of Kapusniak et al. (2024), who point out that $X_t$ linearly interpolating between data $X_0$ and noise $X_T$ has some undesirable properties; they thus propose a modification of CFM with the goal of ensuring that the interpolations remain close to geodesics in $\mathcal{M}$, which they show leads to empirical improvements.

### 5.1.4  Noisy Normalizing Flows

Several models have been proposed based on the idea of adding noise to the data, training a normalizing flow (Section 4.1.3) on this noisy data, and then having a "deflation" procedure whose goal is to sample from $p_*^X$ rather than its learned noisy version. One such model is the denoising NF of Horvat & Pfister (2021), which itself is heavily based on the theoretical results of Horvat & Pfister (2023).[18]  Motivated by the inherent struggles of flow-based architectures on manifold-supported data (described in Section 4.1 and Section 4.1.3), denoising NFs attempt to model an "inflated" version of the data distribution with added noise, and then provide conditions under which this noise can be "deflated" to recover the true on-manifold density. Similar to denoising score matching (Section 5.1.1), the inflation step of Horvat & Pfister (2021) consists of simply adding full dimensional Gaussian noise to the data, and building a density model for $p_*^{X_\sigma} := p_*^X \circledast \mathcal{N}(\cdot; 0, \sigma^2 I_D)$, where $\sigma > 0$ is a hyperparameter. On the other hand, Horvat & Pfister (2023) discuss a more theoretically-grounded inflation step, where $(D - d^*)$-dimensional Gaussian noise is added to $X \sim p_*^X$ along the *normal space* of $\mathcal{M}$ at $X$.[19]  While determining the normal space is only possible for known manifolds, the authors argue that when $d^*$ is much smaller than $D$, using full-dimensional Gaussian noise is a good approximation of $(D - d^*)$-dimensional noise in the normal space. We will shortly review the deflation step.

To model $p_*^{X_\sigma}$, Horvat & Pfister (2021) use a full-dimensional DGM $p_\theta^{X_\sigma}$. Their specific choice is a $D$-dimensional NF, albeit with a non-standard learnable latent distribution $p_\theta^Z$. For some hyperparameter $d$ meant to approximate $d^*$, the first $d$ latent coordinates are themselves modelled by a $d$-dimensional NF, and the remaining $D - d$ latent coordinates use a zero-mean Gaussian with covariance $\sigma^2 I_{D-d}$, matching the

---

[16]In turn, the result of Kingma & Gao (2023) does ensure that if the vector field learned through CFM is converted back to a score function and used alongside the backward SDE for sampling, then the resulting model will be manifold-aware. However, this is not how CFM is sampled from in practice. Note also that any result ensuring that Equation 58 can indeed produce samples from $p_*^X$ in the manifold setting would in turn guarantee the manifold-awareness of CFM.

[17]The closest analysis we are aware of is due to Gao & Zhu (2024), who provide a Wasserstein distance bound for the ODE sampler. However, this bound requires assumptions which are incompatible with the manifold setting, and thus does not ensure manifold-awareness. This stands in contrast with the analogous bound for the SDE sampler by De Bortoli (2022), which does guarantee manifold-awareness.

[18]Horvat & Pfister (2023) was released first on arXiv despite having a later publication date than Horvat & Pfister (2021).

[19]"Normal" in the geometric sense, not the Gaussian sense.

noise level of the inflation step above. The overall likelihood for their full-dimensional model is thus

$$p_\theta^{X_\sigma}(x) = p_\theta^{Z_1}(z_1)p^{Z_2}(z_2) \left| \det \nabla_x f_\theta(x) \right|, \tag{62}$$

where: $f_\theta = g_\theta^{-1}$ with $g_\theta$ being the $D$-dimensional NF, $z = (z_1, z_2) = f_\theta(x)$ with $z_1 \in \mathbb{R}^d$ and $z_2 \in \mathbb{R}^{D-d}$, and the latent density $p_\theta^Z$ is given by $p_\theta^Z(z) = p_\theta^{Z_1}(z_1)p_\theta^{Z_2}(z_2)$, where $p_\theta^{Z_1}$ is the density corresponding to the $d$-dimensional NF model, and $p^{Z_2}(\cdot) = \mathcal{N}(\cdot\,;0,\sigma^2 I_{D-d})$ with no trainable parameters. The idea is that the first $d$ latent coordinates are noise-insensitive – i.e. they are meant to *denoise* the data – while the remaining $D - d$ coordinates are noise-sensitive. The model $p_\theta^{X_\sigma}$ defined above has no explicit manifold-awareness. Horvat & Pfister (2021) thus add an injective flow construction (reviewed further in Section 5.3.3) into the mix, defining the actual (denoised) manifold-supported generative process $p_\theta^X$ by first sampling $Z_1 \sim p_\theta^{Z_1}$ and then setting $X = g_\theta((Z_1, 0))$, where $0 \in \mathbb{R}^{D-d}$ and $(\cdot, \cdot)$ denotes concatenation so that $(Z_1, 0) \in \mathbb{R}^D$. Horvat & Pfister (2021) refer to using the manifold-supported $p_\theta^X$ rather than the full-dimensional $p_\theta^{X_\sigma}$ as "deflating" $p_\theta^{X_\sigma}$.

Horvat & Pfister (2021) also follow injective flows (Equation 97) in adding a reconstruction term as a regularizer to encourage manifold-awareness of the overall objective, which can be written as

$$\max_\theta \mathbb{E}_{X_\sigma \sim p_*^{X_\sigma}} \left[ \log p_\theta^{X_\sigma}(X_\sigma) \right] - \beta\,\mathbb{E}_{X_\sigma \sim p_*^{X_\sigma}} \left[ \| X_\sigma - g_\theta\left((f_\theta(X_\sigma)_1, 0)\right) \|_2^2 \right], \tag{63}$$

where $\beta > 0$ is a hyperparameter, $f_\theta(X_\sigma)_1 \in \mathbb{R}^d$ corresponds to the first $d$ coordinates of $f_\theta(X_\sigma)$, and once again $0 \in \mathbb{R}^{D-d}$. The regularizer encourages $g_\theta$ to not need $Z_2 = f_\theta(X_\sigma)_2$ (i.e. the last $D - d$ coordinates of $f_\theta(X_\sigma)$) to perfectly reconstruct $X_\sigma$: this can be intuitively understood as providing an inductive bias which promotes capturing the added noise only through $Z_2$, thus justifying the use of $p_\theta^X$ instead of $p_\theta^{X_\sigma}$.

We now discuss the "deflation" aspect of this approach. Horvat & Pfister (2023) prove that a trained denoising normalizing flow $p_{\theta^*}^X$ recovers $p_*^X$ under the following conditions: (*i*) the noise added to $X \sim p_*^X$ is only added along the $(D - d^*)$-dimensional normal space to the true manifold $\mathcal{M}$ at $X$, (*ii*) the noise parameter $\sigma$ is sufficiently small, and (*iii*) the manifold $\mathcal{M}$ is "sufficiently smooth and disentangled" (this condition is properly formalized by Horvat & Pfister (2023)).

Condition (*i*) is not satisfied by the denoising normalizing flow, although Horvat & Pfister (2021) argue that when $D$ is much larger than $d^*$, $\mathcal{N}(\cdot\,;0,\sigma^2 I_D)$ is a good approximation for the $(D - d^*)-$dimensional Gaussian noise in the normal space; it is also worth reiterating that it would not be possible to add noise in the normal space without knowing the true manifold exactly in the first place. Condition (*iii*) is also worth discussing: it is entirely unclear if natural data observed "in the wild" is sufficiently smooth and disentangled, so we may not have any guarantee of retrieving the true manifold-supported data distribution even in the nonparametric regime. In summary, despite denoising NFs being an elegant approach, their manifold-awareness can only be ensured under potentially strong and hard-to-verify assumptions.

Postels et al. (2022) proposed a very similar approach to denoising NFs, except the latent density used to define $p_\theta^{X_\sigma}$ is fixed as a standard Gaussian, no regularizer is used during training, and the "deflation" step is done by first sampling $X_\sigma \sim p_{\theta^*}^{X_\sigma}$ and then solving

$$X = \arg\max_x \log p_{\theta^*}^{X_\sigma}(x) - \beta\|x - X_\sigma\|_2^2. \tag{64}$$

Although Postels et al. (2022) do not rigorously justify their method, the intuition behind this "deflation" step is sensible; since $p_{\theta^*}^{X_\sigma}$ should spike around $\mathcal{M}$ when $\sigma$ is small enough, Equation 64 can be informally interpreted as pulling $X_\sigma$ closer to $\mathcal{M}$.

Finally, Kim et al. (2020) use a continuum of noise levels in a manner reminiscent of diffusion models (Section 5.1.2). More specifically, they first construct a conditional normalizing flow $g_\theta : \mathcal{Z} \times [0, \sigma_{\max}] \to \mathcal{X}$, where $\sigma_{\max} > 0$ is a hyperparameter. For every $\sigma \in [0, \sigma_{\max}]$, the flow $g_\theta(\cdot, \sigma)$ must be a diffeomorphism, whose inverse we denote as $f_\theta(\cdot, \sigma)$. They then condition the NF on the amount of added noise and train through (conditional) maximum-likelihood:

$$\max_\theta \int_0^{\sigma_{\max}} \mathbb{E}_{X_\sigma \sim p_*^{X_\sigma}} \left[ \log p^Z \left( f_\theta(X_\sigma, \sigma) \right) + \log \left| \det \nabla_{x_\sigma} f_\theta(X_\sigma, \sigma) \right| \right] d\sigma. \tag{65}$$

Since the conditional NF is trained so that $X_\sigma = g_{\theta^*}(Z, \sigma)$, where $Z \sim p^Z$, is distributed according to $p_*^{X_\sigma}$, the "deflation" step here simply consists of using $\sigma = 0$ when sampling from the model, i.e. $X = g_{\theta^*}(Z, 0)$.

Finally, we highlight that although the three methods presented above entail fitting full-dimensional likelihood-based models to full-dimensional target densities, the target densities consist of the convolution of manifold-supported densities with small amounts of noise. As a result, these methods are still exposed to the numerical pathologies described in Section 4.1.1.

### 5.1.5 Spread Divergences

As previously mentioned, attempting to minimize KL divergence through maximum-likelihood in the manifold setting can result in manifold overfitting (Section 4.1), and while adding a small amount of noise to the data can circumvent the problem in theory, it might not in practice. Zhang et al. (2020a) propose to not only add noise to the data, but to add the same amount of noise to the model as well. They formalize this idea by introducing spread divergences. Here we will focus exclusively on the spread KL divergence and Gaussian noise, but highlight that the same ideas can be applied to other divergences and (potentially learnable) noise distributions. Formally, for a fixed $\sigma > 0$, the spread KL divergence $\mathbb{KL}_\sigma(p \,\|\, q)$ between probability densities $p$ and $q$ is given by

$$\mathbb{KL}_\sigma(p \,\|\, q) \coloneqq \mathbb{KL}(p_\sigma \,\|\, q_\sigma), \tag{66}$$

where $p_\sigma \coloneqq p \circledast \mathcal{N}(\,\cdot\,; 0, \sigma^2 I_D)$ and $q_\sigma \coloneqq q \circledast \mathcal{N}(\,\cdot\,; 0, \sigma^2 I_D)$, i.e. the spread KL divergence is the KL divergence between noisy versions of $p$ and $q$. The spread KL divergence has two important properties: $\mathbb{KL}_\sigma(p \,\|\, q) \geq 0$, with equality if and only if $p = q$; and it is always meaningfully defined, since the noisy versions of the original distributions are always full-dimensional (Section 3.4). Together, these properties imply that the spread KL divergence provides a mathematically sensible objective to train generative models under the manifold setting.

While the noisy model $p_\theta^X \circledast \mathcal{N}(\,\cdot\,; 0, \sigma^2 I_D)$ always has a full-dimensional density, this density cannot in general be evaluated, and thus minimizing $\mathbb{KL}_\sigma(p_*^X \,\|\, p_\theta^X)$ is not immediately trivial. Zhang et al. (2020a) propose a very similar model to a variational autoencoder (Section 4.1.2), called the $\delta$-VAE, where the conditional distribution of $X$ given $Z$ is now a point mass at $g_\theta(Z)$, instead of a Gaussian as in Equation 22. In other words, this model is like a Gaussian VAE, except no additional noise is added to $g_\theta(Z)$, i.e. all the noise in this model comes from the low-dimensional $Z$. Note that, unlike VAEs, this model is not full-dimensional. Much like VAEs are trained to minimize an upper bound of $\mathbb{KL}(p_*^X \,\|\, p_\theta^X)$ (or equivalently, maximizing the lower bound to the log-likelihood from Equation 24), $\delta$-VAEs minimize an upper bound of $\mathbb{KL}_\sigma(p_*^X \,\|\, p_\theta^X)$, which as we will see ends up amounting to a simple modification to standard VAE training. Importantly, even if the supports of $p_\theta^X$ and $p_*^X$ do not overlap, their spread KL divergence is meaningfully defined and thus provides a valid objective that enables $\delta$-VAEs to properly learn distributions on unknown manifolds. More specifically, $\delta$-VAEs use a variational posterior density $q_\phi^{Z|X_\sigma}$ and are trained through

$$\max_{\theta,\phi} \mathbb{E}_{X_\sigma \sim p_*^{X_\sigma}} \left[ \mathbb{E}_{Z \sim q_\phi^{Z|X_\sigma}(\cdot|X_\sigma)}[\log \mathcal{N}(X_\sigma; g_\theta(Z), \sigma^2 I_D)] - \mathbb{KL}\left( q_\phi^{Z|X_\sigma}(\cdot|X_\sigma) \,\Big\|\, p_Z \right) \right], \tag{67}$$

where $p_*^{X_\sigma} \coloneqq p_*^X \circledast \mathcal{N}(\,\cdot\,; 0, \sigma^2 I_D)$. Note that this objective is equivalent to that of VAEs from Equation 24, except Gaussian noise is added to the data, and the covariance of the decoder is not learnable, but rather made to match the covariance of the noise that was added to the data.

A point about $\delta$-VAEs warrants discussion. A common "cheat" when sampling from a trained Gaussian VAE model is to sample $Z \sim p^Z$, and then output the decoder mean $g_{\theta^*}(Z)$, rather than outputting a sample from a Gaussian centered at this point (i.e. the noise from the stochastic decoder is ignored). The use of the spread KL divergence elegantly justifies this common practice, since the noise (corresponding to the $\mathcal{N}(X_\sigma; g_\theta(Z), \sigma^2 I_D)$ term in Equation 67) here is added due to the spread KL divergence, rather than being part of the model itself. Zhang et al. (2020a) did not make this observation when introducing $\delta$-VAEs, and to the best of our knowledge, we are the first to point this out.

Finally, Zhang et al. (2023) aim to extend the use of the spread KL divergence beyond VAEs, and propose a procedure to use it within the context of normalizing flows (Section 4.1.3). Since the spread KL divergence

remains intractable, they introduce another bound as the training objective. Unlike other variational bounds such as the ELBO (Equation 24) or Equation 67, the bound of Zhang et al. (2023) does not become tight at optimality, so it lacks the theoretical guarantees of $\delta$-VAEs which ensure manifold-awareness.

## 5.2 Manifold-Awareness through Support-Agnostic Optimization Objectives

In Section 5.1 we showed various DGMs which learn manifolds by first adding noise to $p_*^X$, and then using a full-dimensional objective like score matching or KL minimization (i.e. maximum-likelihood). An alternative to these approaches is using an objective that is agnostic to the supports of the underlying distributions being compared. In particular, divergences between probability distributions which metrize weak convergence, such as Wasserstein distances (Section 3.5) or maximum mean discrepancy (Section 3.6), provide such support-agnostic objectives. We now cover DGMs which are trained through these objectives.

### 5.2.1 Wasserstein Generative Adversarial Networks

Arjovsky et al. (2017) proposed to minimize Wasserstein distance (Section 3.5) between $p_*^X$ and $p_\theta^X$ as a way to address the issues arising from attempting to train generative adversarial networks using various $f$-divergences outlined in Section 4.2, resulting in strong empirical performance (Karras et al., 2018; 2019; 2020). In the GAN context considered by Arjovsky et al. (2017), where $p_\theta^X$ is again given by the distribution of $X = g_\theta(Z)$, where $Z \sim p^Z$ (once more, formally, $p_\theta^X$ is given by the pushforward of $p^Z$ through $g_\theta$) and $p^Z$ is fixed (e.g. standard Gaussian), this means changing the GAN objective from Equation 41 to

$$\min_\theta \max_\phi \mathbb{E}_{X \sim p_*^X} \left[ h_\phi(X) \right] - \mathbb{E}_{Z \sim p^Z} \left[ h_\phi(g_\theta(Z)) \right], \tag{68}$$

where the neural network $h_\phi : \mathcal{X} \to \mathbb{R}$ is no longer a classifier and is now constrained to be Lipschitz. If, as with standard GANs, the network $h_\phi$ is assumed to be arbitrarily flexible, the objective reduces to minimizing the $\mathbb{W}_1$ distance between the true distribution and the model (Equation 17),

$$\min_\theta \mathbb{W}_1(p_*^X, p_\theta^X), \tag{69}$$

which as mentioned in Section 3.5, provides a support-agnostic optimization objective for training $p_\theta^X$. Arjovsky et al. (2017) enforce the Lipschitz constraint on $h_\phi$ through weight clipping, which was later improved upon by Gulrajani et al. (2017a) and by Miyato et al. (2018). The former use a regularizer encouraging $\|\nabla_x h_\phi(x)\|_2$ to be close to 1, while the latter regularizes the spectral norm of the weight parameters of $h_\phi$; both obtain much better empirical performance than weight clipping.

### 5.2.2 Wasserstein Autoencoders

As described in Section 5.2.1 for Wasserstein generative adversarial networks, Arjovsky et al. (2017) leveraged the dual formulation of $\mathbb{W}_1$ (Equation 17). Tolstikhin et al. (2018) proposed Wasserstein autoencoders (WAEs), which instead leverage the definition of $\mathbb{W}^c$ (Equation 16) – which remains a support-agnostic objective for training DGMs. In particular, they proved that when $p_\theta^X$ is given by the distribution of $X = g_\theta(Z)$ where $Z \sim p^Z$ (once again, $p_\theta^X$ formally corresponds to the pushforward of $p^Z$ through $g_\theta$), the optimal transport cost can be written as

$$\mathbb{W}^c(p_*^X, p_\theta^X) = \inf_{q^{Z|X} \in \mathcal{Q}(p_*^X, p^Z)} \mathbb{E}_{X \sim p_*^X} \left[ \mathbb{E}_{Z \sim q^{Z|X}(\cdot|X)} \left[ c(X, g_\theta(Z)) \right] \right], \tag{70}$$

where $\mathcal{Q}(p_*^X, p^Z)$ is the set of conditional (on $X$) densities on $\mathcal{Z}$ whose marginal matches $p^Z$, i.e. $\mathcal{Q}(p_*^X, p^Z) := \{q^{Z|X} : \mathcal{X} \to \Delta(\mathcal{Z}) \mid q^Z = p^Z\}$, where $q^Z := \mathbb{E}_{X \sim p_*^X}[q^{Z|X}(\cdot|X)]$. Tolstikhin et al. (2018) propose two losses to train WAEs, both inspired by this property: analoguously to variational autoencoders, a conditional distribution $q_\phi^{Z|X}$ is parameterized (e.g. Equation 26), and the first loss is given by

$$\min_{\theta,\phi} \max_{\phi'} \mathbb{E}_{X \sim p_*^X} \left[ \mathbb{E}_{Z \sim q_\phi^{Z|X}(\cdot|X)}[c(X, g_\theta(Z))] \right] + \beta \left( \mathbb{E}_{Z \sim q_\phi^Z} \left[ \log h_{\phi'}(Z) \right] + \mathbb{E}_{Z \sim p^Z} \left[ \log \left( 1 - h_{\phi'}(Z) \right) \right] \right), \tag{71}$$

where $h_{\phi'} : \mathcal{Z} \to (0,1)$, and $\beta > 0$ is a hyperparameter. The first term in the above objective can be understood as minimizing the cost in Equation 70 with no regard for the constraint that $q_\phi^{Z|X} \in \mathcal{Q}(p_*^X, p^Z)$, whereas the second term encourages this constraint to be satisfied through an adversarial loss (Equation 41) that aims to minimize $\mathbb{JS}(q_\phi^Z \| p^Z)$; thus, Equation 71 does indeed aim to minimize the optimal transport cost in Equation 70. Note that even though $q_\phi^Z$ cannot be evaluated, it is trivial to obtain a sample $Z$ from it by first sampling $X \sim p_*^X$, and then sampling $Z|X \sim q_\phi^{Z|X}(\cdot|X)$, so that optimizing Equation 71 is tractable.

We find it relevant to highlight some differences between WAEs and other DGMs. ($i$) Unlike GANs (Section 4.2) and Wasserstein GANs, WAEs use an adversarial loss on the latent space $\mathcal{Z}$ rather than on ambient space $\mathcal{X}$. This helps stabilize WAEs, as adversarial losses on ambient space can be notoriously unstable. Nonetheless, properly trained Wasserstein GANs tend to empirically outperform WAEs. ($ii$) WAEs are also very similar to variational autoencoders (Section 4.1.2) – e.g. if $c(x,y) = \|x - y\|_2^2$, the WAE and Gaussian VAE losses become extremely alike – but the KL term in the VAE loss (Equation 24) encourages $q_\phi^{Z|X}(\cdot|X)$ to match $p^Z$ for every $X$ sampled from $p_*^X$, whereas WAEs only encourage this to happen on average, i.e. $\mathbb{E}_{X \sim p_*^X}[q_\phi^{Z|X}(\cdot|X)] = q_\phi^Z = p^Z$.

The second WAE loss is completely analogous, except it uses maximum mean discrepancy (Section 3.6) instead of an adversarial loss to encourage satisfying the constraint that $q_\phi^Z = p^Z$,

$$\min_{\theta,\phi} \mathbb{E}_{X \sim p_*^X} \left[ \mathbb{E}_{Z \sim q_\phi^{Z|X}(\cdot|X)}[c(X, g_\theta(Z))] \right] + \beta \mathbb{MMD}_k^2 \left( q_\phi^Z, p^Z \right) \tag{72}$$

for some kernel $k : \mathcal{Z} \times \mathcal{Z} \to \mathbb{R}$, which results in an objective with no adversarial component. The choice of which objective to use on latent space to enforce $q_\phi^Z = p^Z$ has also been expanded in follow-up work, for example Kolouri et al. (2018) use the sliced Wasserstein distance, and Patrini et al. (2020) use relaxed (Sinkhorn) optimal transport.

Finally, we point out that Tolstikhin et al. (2018) found that using arbitrarily distributions $q_\phi^{Z|X}$ was not key for good empirical performance, and they thus restrict $\mathcal{Q}(p_*^X, p_\theta^X)$ to only contain point masses, i.e. $q_\phi^{Z|X}(\cdot|x)$ is given by a point mass at $f_\phi(x)$. This choice, which amounts to using deterministic rather than stochastic encoders, reduces the first term in Equation 71 and Equation 72 to a reconstruction error (as measured by $c$), $\mathbb{E}_{X \sim p_*^X}[c(X, g_\theta(f_\phi(X)))]$.

> Here we simply highlight that since the distributions involved need not necessarily admit Lebesgue densities, Equation 70 must be formalized by using measures instead of densities:
>
> $$\mathbb{W}^c(\mathbb{P}_*^X, \mathbb{P}_\theta^X) = \inf_{\mathbb{Q}^{Z|X} \in \mathcal{Q}(\mathbb{P}_*^X, \mathbb{P}^Z)} \mathbb{E}_{X \sim \mathbb{P}_*^X} \left[ \mathbb{E}_{Z \sim \mathbb{Q}^{Z|X}(\cdot|X)} \left[ c(X, g_\theta(Z)) \right] \right], \tag{73}$$
>
> where $\mathcal{Q}(\mathbb{P}_*^X, \mathbb{P}^Z) := \{\mathbb{Q}^{Z|X} : \mathcal{X} \to \Delta(\mathcal{Z}) \mid \mathbb{Q}^Z = \mathbb{P}^Z\}$ with $\mathbb{Q}^Z := \mathbb{E}_{X \sim \mathbb{P}_*^X}[\mathbb{Q}^{Z|X}(\cdot|X)]$.

### 5.2.3 Generative Networks Based on Maximum Mean Discrepancy

Dziugaite et al. (2015) and Li et al. (2015) propose another way to train the model $p_\theta^X$ corresponding to $X = g_\theta(Z)$ and $Z \sim p^Z$ (again, we point out that formally, this model corresponds to the pushforward of $p^Z$ through $g_\theta$): by minimizing maximum mean discrepancy (Section 3.6). Although their motivation was to avoid the adversarial training involved in generative adversarial networks (Section 4.2) rather than to model manifold-supported data, the resulting objective provides a mathematically principled way of training DGMs under the manifold setting. The training objective of generative moment matching networks is simply

$$\min_\theta \mathbb{MMD}_k^2 \left( p_*^X, p_\theta^X \right) \tag{74}$$

for a pre-specified kernel $k$. Although minimizing MMD is straightforward, generative moment matching networks are not known for achieving good empirical performance: this highlights that, even though manifold-awareness should be considered a necessary condition for strong empirical results, it is not sufficient.

To improve the empirical performance of generative moment matching networks, Li et al. (2017) propose maximum mean discrepancy generative adversarial networks (MMD GANs), where the main idea is to reintroduce adversarial training to learn the kernel. First, for a fixed auxiliary latent space $\mathcal{Z}' = \mathbb{R}^{d'}$, a given a kernel $k : \mathcal{Z}' \times \mathcal{Z}' \to \mathbb{R}$, and a neural network $h_\phi : \mathcal{X} \to \mathcal{Z}'$, Li et al. (2017) defined the kernel $k_\phi : \mathcal{X} \times \mathcal{X} \to \mathbb{R}$ as $k_\phi(x, y) := k(h_\phi(x), h_\phi(y))$. They then showed that, if some regularity conditions hold and $h_\phi$ is injective, then $\max_\phi \mathbb{MMD}^2_{k_\phi}$ metrizes weak convergence, thus making it a mathematically sensible objective to train DGMs. In order to enforce injectivity of $h_\phi$, Li et al. (2017) leverage the fact that a function $h : \mathcal{X} \to \mathcal{Z}'$ is injective on $\mathcal{X}$ if and only if it admits a left inverse $h^\dagger : \mathcal{Z}' \to \mathcal{X}$, i.e. $h^\dagger(h(x)) = x$ for all $x \in \mathcal{X}$. They thus introduce an auxiliary network $h_\phi^\dagger$ (which need not share parameters with $h_\phi$), and train MMD GANs through

$$\min_\theta \max_\phi \mathbb{MMD}^2_{k_\phi}\left(p_*^X, p_\theta^X\right) - \beta \mathbb{E}_{X \sim \frac{1}{2}p_*^X + \frac{1}{2}p_\theta^X}\left[\|X - h_\phi^\dagger\left(h_\phi(X)\right)\|_2^2\right], \tag{75}$$

where $\beta > 0$ is a hyperparameter, and the second term encourages $h_\phi$ to admit $h_\phi^\dagger$ as a left inverse on the supports of $p_*^X$ and $p_\theta^X$. Similarly to Wasserstein GANs (Section 5.2.1), the empirical performance of MMD GANs benefits from gradient regularization during training (Bińkowski et al., 2018; Arbel et al., 2018).

### 5.2.4 Generalized Energy-Based Models

Generalized energy-based models (GEBMs; Arbel et al., 2021) combine generative adversarial networks (Section 4.2) with energy-based models (Section 4.1.4). GEBMs consist of a fixed prior $p^Z$ supported on $\mathcal{Z}$, a generator $g_{\theta_1} : \mathcal{Z} \to \mathcal{X}$, and an energy function $E_{\theta_2} : \mathcal{X} \to \mathbb{R}$, where we explicitly distinguish between the parameters of these components as $\theta = (\theta_1, \theta_2)$. As in GANs, the prior along with the generator implicitly define the density $p_{\theta_1}^g$ of $X = g_{\theta_1}(Z)$, where $Z \sim p^Z$.[20] This is not a full-dimensional density, but rather it is supported on the model manifold $\mathcal{M}_{\theta_1} := g_{\theta_1}(\mathcal{Z})$.[21] GEBMs define an EBM on $\mathcal{M}_{\theta_1}$ by re-weighting $p_{\theta_1}^g$ through the use of $E_{\theta_2}$ as an energy function:

$$p_\theta^X(x) \propto p_{\theta_1}^g(x) e^{-E_{\theta_2}(x)}. \tag{76}$$

Despite $E_{\theta_2}$ being defined over $\mathcal{X}$, the above density is only defined on $\mathcal{M}_{\theta_1}$ and is thus also not a full-dimensional density.[22] The intuition behind GEBMs is that the generator can easily learn to map to $\mathcal{M}$, whereas the energy function helps correct the distribution within the learned manifold.

In order to train GEBMs, Arbel et al. (2021) define the quantity

$$\mathbb{KALE}\left(p_*^X \,\|\, p_{\theta_1}^g\right) := \sup_{E \in \mathcal{E}, \phi \in \mathbb{R}} 1 - \phi - \mathbb{E}_{X \sim p_*^X}\left[E(X)\right] - \mathbb{E}_{Z \sim p^Z}\left[e^{-E(g_{\theta_1}(Z)) - \phi}\right], \tag{77}$$

where $\mathcal{E}$ is a set of Lipschitz energy functions satisfying certain regularity conditions (which are satisfied by feed-forward neural networks). They then show that $\mathbb{KALE}(p_*^X \,\|\, p_{\theta_1}^g)$ is a meaningfully defined divergence between $p_*^X$ and $p_{\theta_1}^g$, even when their supports do not perfectly overlap (i.e. it metrizes weak convergence, more details are provided in the grey box below), so that it provides a sensible objective to train the generator. As a divergence, $\mathbb{KALE}$ is intimately related to the KL divergence: Arbel et al. (2021) also show that if $E_{\theta_2}$ achieves the supremum in Equation 77, then $\mathbb{KL}(p_*^X \,\|\, p_\theta^X) \leq \mathbb{KL}(p_*^X \,\|\, p_{\theta_1}^g)$.[23] Therefore, the re-weighting done in Equation 76 to $p_{\theta_1}^g$ indeed improves upon simply using $p_{\theta_1}^g$. Putting these properties together, Arbel et al. (2021) train GEBMs through

$$\min_{\theta_1} \max_{\theta_2, \phi} 1 - \phi - \mathbb{E}_{X \sim p_*^X}\left[E_{\theta_2}(X)\right] - \mathbb{E}_{Z \sim p^Z}\left[e^{-E_{\theta_2}(g_{\theta_1}(Z)) - \phi}\right], \tag{78}$$

---

[20]Note that in Section 4.2 we denoted $p_{\theta_1}^g$ as $p_\theta^X$, but we use different notation here as GEBMs further modify $p_{\theta_1}^g$.

[21]Formally $\mathcal{M}_{\theta_1}$ need not be a manifold, even if $g_{\theta_1}$ is smooth, as it might have points of self-intersection. The measure-theoretic formulation of GEBMs in the grey box below remains nonetheless valid.

[22]More formally, the $\propto$ symbol in Equation 76 should be understood as proportional *within* $\mathcal{M}_{\theta_1}$, only integrating over the model manifold, i.e. $p_\theta^X(x) = p_{\theta_1}^g(x) e^{-E_{\theta_2}(x)} / \int_{\mathcal{M}_{\theta_1}} p_{\theta_1}^g e^{-E_{\theta_2}} \mathrm{dvol}_{\mathcal{M}_{\theta_1}}$.

[23]Note that none of the involved densities – namely $p_*^X$, $p_\theta^X$, and $p_{\theta_1}^g$ – are full-dimensional densities, so the KL divergence between them could be infinite (Section 3.4). The inequality is thus trivially true when the support of $p_\theta^X$ and $p_{\theta_1}^g$, i.e. $\mathcal{M}_{\theta_1}$, does not match $\mathcal{M}$. Nonetheless, when the generator perfectly recovers $\mathcal{M}$, the inequality does justify the use of the energy function in GEBMs.

where the Lipschitz constraint on $E_{\theta_2}$ is enforced as in Wasserstein GANs (Section 5.2.1) and $\phi \in \mathbb{R}$ is a free auxiliary parameter.

Arbel et al. (2021) also show that in order to sample from a GEBM as defined through Equation 76, one can first sample $Z$ from the EBM $p_\theta^Z$ on $\mathcal{Z}$ given by

$$p_\theta^Z(z) \propto p^Z(z) e^{-E_{\theta_2}(g_{\theta_1}(z))}, \tag{79}$$

and then setting $X = g_{\theta_1}(Z)$ will produce a sample from $p_\theta^X$. Note that $p_\theta^Z$ is now a full-dimensional density in $\mathcal{Z}$, and it can thus be sampled through Markov chain Monte Carlo as standard in EBMs. We point out that GEBMs are intimately linked to two-step models, which we discuss in Section 5.3.

Finally, we highlight some related works. Che et al. (2020) proposed a similar model to GEBMs, but their model is trained as a standard GAN (Equation 41), and the EBM is defined *post-hoc* by using the discriminator $h_{\phi^*}$; despite the similarity with GEBMs, this procedure does not endow manifold-unaware GANs with manifold-awareness. Birrell et al. (2022) constructed a class of divergences, which $\mathbb{KALE}$ belongs to, by extending its relationship with the KL divergence to general $f$-divergences; and Gu et al. (2024) leveraged these divergences, along with optimal transport (Section 3.5), to obtain a manifold-aware adversarial training objective for continuous normalizing flows (Section 4.1.3).

---

We now formalize the presentation of GEBMs. Here we denote $p_{\theta_1}^g$ as a probability measure, $g_{\theta_1 \#} \mathbb{P}^Z$, instead of as a density. The model distribution $\mathbb{P}_\theta^X$ is then defined through its Radon-Nikodym derivative with respect to $g_{\theta_1 \#} \mathbb{P}^Z$,

$$p_\theta^X(x) = \frac{\mathrm{d}\mathbb{P}_\theta^X}{\mathrm{d}g_{\theta_1 \#}\mathbb{P}^Z}(x) := \frac{e^{-E_{\theta_2}(x)}}{\mathbb{E}_{X \sim g_{\theta_1 \#}\mathbb{P}^Z}[e^{-E_{\theta_2}(X)}]}, \tag{80}$$

where the expectation is assumed to be finite. Arbel et al. (2021) proved under mild conditions that: (*i*) $\mathbb{KALE}(\mathbb{P}_*^X \,\|\, g_{\theta_1 \#}\mathbb{P}^Z) \geq 0$ with equality if and only if $\mathbb{P}_*^X = g_{\theta_1 \#}\mathbb{P}^Z$; and that (*ii*) for a sequence of generators $(g_{\theta_{1,t}})_{t=1}^\infty$, $\mathbb{KALE}(\mathbb{P}_*^X \,\|\, g_{\theta_{1,t} \#}\mathbb{P}^Z) \to 0$ as $t \to \infty$ if and only if $g_{\theta_{1,t} \#}\mathbb{P}^Z \xrightarrow{\omega} \mathbb{P}_*^X$ as $t \to \infty$.

---

### 5.2.5 Principal Component Flows

Principal component flows (PCFs; Cunningham et al., 2022) are a variant on standard normalizing flows (Section 4.1.3) that seek to uncover manifold structure in a manner analogous to principal component analysis (PCA) and related to disentanglement (Bengio et al., 2013). If $g_\theta : \mathcal{Z} \to \mathcal{X}$ is a normalizing flow, the eigenvectors $\{\nu_1(x), \ldots, \nu_D(x)\}$ of $\nabla_z g_\theta(z) \nabla_z g_\theta(z)^\top$ are said to be the *principal components of $g_\theta$ at $x = g_\theta(z)$* and can be ordered using the corresponding eigenvalues as in standard PCA. The principal components at $x$ represent the principal axes of variation in the flow's density $p_\theta^X$. In a manifold-learning context, principal axes with small eigenvalues represent off-manifold directions, while those with the highest eigenvalues represent primary directions of variation along the manifold, as illustrated in Figure 7(a): we highlight that here $p_*^X$ is assumed full-dimensional and to concentrate around $\mathcal{M}$, rather than being strictly manifold-supported.

We now discuss a special case of PCFs as an introduction; for a presentation of the method in more generality, see the work of Cunningham et al. (2022). PCFs involve the notion of *contour log-likelihoods*, which here can be interpreted as the likelihood along the $i$th coordinate curve of the flow,

$$\log p^{Z_i}(z_i) - \log \left( \nabla_z g_\theta(x)_{ii} \right) \tag{81}$$

for $i \in \{1, 2, \ldots, D\}$, where $p^Z(z) = \prod_{i=1}^D p^{Z_i}(z_i)$ is a coordinatewise factorization of the latent prior (which is typically Gaussian), and $\nabla_x g_\theta(x)_{ii}$ is the $i$th entry on the diagonal of $\nabla_x g_\theta(x)$.

The goal of training a PCF is to align the principal components of $g_\theta$ with its latent coordinates on the manifold (Figure 7(b)). One of the key insights of Cunningham et al. (2022) is that the difference between the model's log density $\log p_\theta^X(x)$ and the sum of its contour log-likelihoods measures the diagonality of

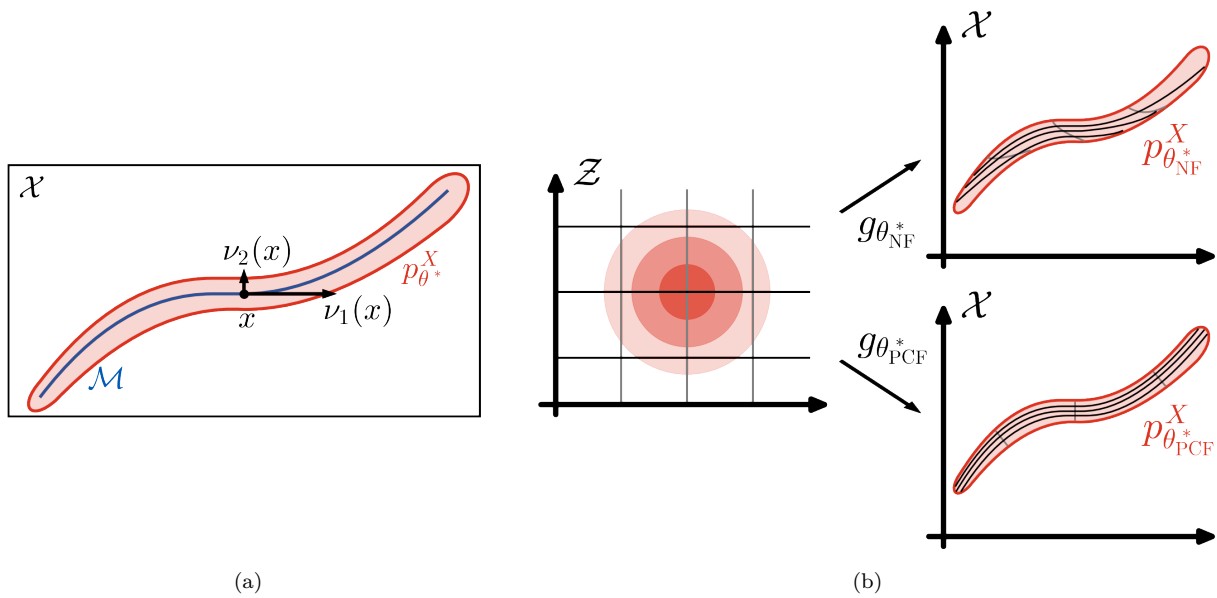

(a)                                                      (b)

Figure 7: The motivation behind PCFs. **(a)** The principal components, $\nu_1(x)$ and $\nu_2(x)$, of a trained NF with density $p_{\theta^*}^X$, taken at a point $x \in \mathcal{X}$. These principal components are scaled according to their eigenvalues: here, $\nu_1(x)$ is the primary direction of variation along the manifold $\mathcal{M}$. **(b)** A comparison between the contours of two NFs: $p_{\theta_{\mathrm{NF}}^*}^X$ and $p_{\theta_{\mathrm{PCF}}^*}^X$, trained as a standard NF and as a PCF, respectively. Contours are visualized by the way each NF maps the grid lines of $\mathcal{Z}$. Since the contours mapped by $g_{\theta_{\mathrm{PCF}}^*}$ correspond to the principal components of $p_{\theta_{\mathrm{PCF}}^*}^X$, the NF model $p_{\theta_{\mathrm{PCF}}^*}^X$ is formally a principal component flow.

$\nabla_z g_\theta(z) \nabla_z g_\theta(z)^\top$ and hence how well the model's latent coordinates are aligned with its principal components. From this, a regularizer can be derived,

$$\mathcal{I}(\theta) := \mathbb{E}_{X \sim p_*^X}\left[\log p_\theta^X(X) - \sum_{i=1}^{D} \log p^{Z_i}\left(f_\theta(X)_i\right) + \log\left(\nabla_x f_\theta(X)_{ii}\right)\right], \tag{82}$$

where $f_\theta(X)_i$ denotes the $i$th coordinate of $f_\theta(X)$.[24] $\mathcal{I}(\theta)$ is a non-positive quantity such that $\mathcal{I}(\theta) = 0$ if and only if the latent coordinates are perfectly aligned with the flow's principal components at each point $x \in \mathcal{X}$. Maximizing $\mathcal{I}(\theta)$ as a regularizer while maximizing the likelihood aligns the flow's latent coordinates with the *principal manifolds* of the data. We highlight that while this objective was derived to provide a useful inductive bias when manifolds are involved, it remains nonetheless based on full-dimensional likelihoods, and is thus subject to the corresponding pathologies (Section 4.1 and Section 4.1.1).

Canonical manifold flows (CMFs; Flouris & Konukoglu, 2023) take a related approach which directly penalizes off-diagonal elements of $\nabla_z g_\theta(f_\theta(X)) \nabla_z g_\theta(f_\theta(X))^\top$. This results in a potentially looser regularizer which is designed to achieve the same goal at optimality as PCFs. Both PCFs and CMFs have been shown to naturally represent $\mathcal{M}$ using a subset of the flow's latent coordinates, meaning they automatically discover an estimate $d$ of the dimension $d^*$ of $\mathcal{M}$ without the practitioner having to set it as a hyperparameter.

## 5.3   Manifold-Awareness through Two-Step Models

Except for generative adversarial networks, all the DGMs described in Section 4 are manifold-unaware as a direct consequence of misspecified dimensionality: the data is $d^*$-dimensional, whereas the model is $D$-dimensional. The methods described in Section 5.1 address this misspecification by adding noise to the data, making it $D$-dimensional, whereas those from Section 5.2 use manifold-appropriate losses. Another

---

[24]Note that the meaning of $f_\theta(X)_1$ is different here than in Section 5.1.4, where it refers to the first $d$ coordinates of $f_\theta(X)$.

approach to enable manifold-awareness, which we cover here, is to instead reduce the ambient dimension of the data to match its intrinsic dimension before learning its distribution. Two-step approaches do this by separating the overall generative modelling process into two distinct steps. Manifold learning (step 1) typically involves some form of encoding-decoding, which uncovers a lower-dimensional representation space $\mathcal{Z}$ whose dimension $d$ ideally matches the intrinsic dimension $d^*$ of the given data. Distribution learning (step 2) is then carried out on the obtained manifold, which often takes the form of generative modelling of the $d$-dimensional representations learned in the previous step. Importantly, this two-step procedure aims to remove the dimensionality mismatch between the data and the model, and thus circumvent any woes caused by it, such as manifold overfitting (Section 4.1). When discussing two-step models, it will be useful to distinguish between the generative parameters of each step, and we thus write $\theta = (\theta_1, \theta_2)$, where $\theta_1$ are the generative parameters required for the first step, and $\theta_2$ those for the second one. We now outline these two steps in more detail:

- **Manifold learning** Most two-step models use autoencoder-based methods for manifold learning (Section 3.2) as in Equation 6,

$$\min_{\theta_1, \phi} \mathbb{E}_{X \sim p_*^X} \left[ \|X - g_{\theta_1}(f_\phi(X))\|_2^2 \right], \tag{83}$$

  or any variant such as variational autoencoders (Section 4.1.2), although we will see in Section 5.4.1 that non-autoencoder-based choices are also possible. Importantly, the goal in this step is to perform manifold learning rather than generative modelling, so e.g. even if a VAE is used, it is interpreted as a regularized autoencoder rather than a generative model.

- **Distribution learning** This step consists of learning the distribution on the manifold obtained in the previous step. In the standard setup where manifold learning is performed with an autoencoder-based model, a pre-trained encoder $f_{\phi^*}$ and decoder $g_{\theta_1^*}$ pair is available. The encoder defines a distribution $q_{\phi^*}^Z$ of encoded data $f_{\phi^*}(X)$ where $X \sim p_*^X$ (formally, $q_{\phi^*}^Z$ is the pushforward density of $p_*^X$ through $f_{\phi^*}$), which can be learned by instantiating a DGM $p_{\theta_2}^Z$ on $\mathcal{Z}$, and training it with any of the methods covered in this survey (but changing the target distribution from the $D$-dimensional $p_*^X$ to the $d$-dimensional $q_{\phi^*}^Z$), while keeping the encoder $f_{\phi^*}$ and decoder $g_{\theta_1^*}$ frozen. Then, once this low-dimensional DGM is trained, resulting in $p_{\theta_2^*}^Z$, the distribution of the two-step model on the learned manifold can be sampled through $X = g_{\theta_1^*}(Z)$, where $Z \sim p_{\theta_2^*}^Z$ (i.e. $p_{\theta^*}^X$ is formally given by the pushforward of $p_{\theta_2^*}^Z$ through $g_{\theta_1^*}$).

Importantly, by solving the generative modelling task in $\mathcal{Z}$ instead of $\mathcal{X}$, the support of the target distribution $q_{\phi^*}^Z$ is $f_{\phi^*}(\mathcal{M}) \subseteq \mathcal{Z}$, whose dimension should intuitively be given by $\min(d, d^*)$. If the latent dimension $d$ is chosen properly (i.e. $d = d^*$), we should then expect $q_{\phi^*}^Z$ to be full-dimensional within $\mathcal{Z}$. This full-dimensionality stands in contrast to the case discussed in Section 4.1 where the target distribution $p_*^X$ is $d^*$-dimensional but its corresponding ambient space $\mathcal{X}$ is $D$-dimensional. Furthermore, even if $d$ is overspecified as $d^* < d < D$, the "dimensionality gap" – i.e. the ambient dimension of the model minus that of the true manifold – of the second-step model is $d - d^*$, which is smaller than $D - d^*$.[25] Thus, we should intuitively expect any manifold-related woes arising from dimensionality mismatch to be milder than the corresponding full-dimensional issues.

Many two-step models have been proposed in the literature, sometimes with the manifold setting in mind, and some other times simply for tractability, as training DGMs on a low-dimensional latent space is cheaper than doing so in high-dimensional ambient space. Loaiza-Ganem et al. (2022a) provided a theoretical justification for all of these models, proving that under mild regularity conditions, when $d = d^*$ and $\mathbb{E}_{X \sim p_*^X}[\|X - g_{\theta_1^*}(f_{\phi^*}(X))\|_2^2] = 0$ (i.e. perfect reconstructions), then: (*i*) $q_{\phi^*}^Z$ is indeed full-dimensional, and thus $p_{\theta_2}^Z$ can be any DGM, even if manifold-unaware (e.g. trained through maximum-likelihood), and still learn $q_{\phi^*}^Z$; and (*ii*) transforming samples from $p_{\theta_2^*}^Z$ through $g_{\theta_1^*}$ is equivalent to sampling from $p_*^X$, i.e.

---

[25]The case where $d$ is underspecified, i.e. $d < d^*$, is less interesting, as in this case $\mathcal{M}$, and thus $p_*^X$, cannot be recovered.

two-step models recover $p_*^X$. This result, which we discuss further in the next grey box, implies that two-step models learn $\mathcal{M}$, which is an interesting observation since autoencoders by themselves need not – see Figure 2 and the discussion in Section 3.2. In particular, when $p_{\theta_2^*}^Z$ is perfectly trained it must be supported on $f_{\phi^*}(\mathcal{M})$, and since $g_{\theta_1^*}(f_{\phi^*}(\mathcal{M})) = \mathcal{M}$, it follows that together, the support of $p_{\theta_2^*}^Z$ and the decoder $g_{\theta_1^*}$ jointly characterize $\mathcal{M}$.

Here we briefly summarize two-step models which are straightforward combinations of an autoencoder-based model with any other generative model on latent space; two-step models warranting additional discussion are covered in Section 5.3.2 and Section 5.3.3. Dai & Wipf (2019) use a VAE for manifold learning and another VAE for distribution learning on latent space. Xiao et al. (2019) use an autoencoder and a normalizing flow (Section 4.1.3), as do Boehm & Seljak (2022). Ghosh et al. (2020) use an autoencoder with an added regularization term, along with a Gaussian mixture model (which, while not deep, remains a generative model and thus fits the two-step framework). Li et al. (2015) use an autoencoder and then train a generative moment matching network (Section 5.2.3) on the recovered latents, and Dao et al. (2023) use a regularized VAE along with conditional flow matching (Section 5.1.3). The improved generative performance reported in many of these works compared to their full-dimensional counterparts may be attributed to the reduction of dimension mismatch.

We also point out that generalized energy-based models (Section 5.2.4) are very similar to autoencoder-based two-step models, since GEBMs instantiate an energy-based model (Section 4.1.4) on a low-dimensional latent space, whose samples are then mapped through a decoder. GEBMs are nonetheless not autoencoder-based two-step models, the differences being that: GEBMs do not require an encoder, the distribution used by GEBMs on $\mathcal{Z}$ (Equation 79) shares parameters with the decoder, GEBMs are trained end-to-end, and they are limited to using EBMs on the latent space.

**End-to-end training** Although the two-step models described above are manifold-aware, the objective in the first step does not necessarily promote representations that are conducive to distribution learning in the second step. Therefore, one might expect that training these models in an end-to-end manner could further improve their performance. Albeit not always inspired by this motivation, several works have proposed end-to-end objectives, some in the context of variational autoencoders, and some in the context of injective normalizing flows; we discuss the former here and the latter in Section 5.3.3. VAEs as described in Section 4.1.2 assume a fixed prior $p^Z$ on the latent space $\mathcal{Z}$. However, the prior $p_{\theta_2}^Z$ can be made trainable, in which case maximizing the ELBO (Equation 24), i.e.

$$\max_{\theta,\phi} \mathbb{E}_{X \sim p_*^X}\left[\mathbb{E}_{Z \sim q_\phi^{Z|X}(\cdot|X)}[\log p_{\theta_1}^{X|Z}(X|Z)] - \mathbb{KL}\left(q_\phi^{Z|X}(\cdot|X) \,\Big\|\, p_{\theta_2}^Z\right)\right], \tag{84}$$

remains a valid objective (assuming $p_*^X$ is full-dimensional) for end-to-end training of $p_{\theta_1}^{X|Z}$ and $p_{\theta_2}^Z$. Depending on the choice of $p_{\theta_2}^Z$ additional computational tricks might be required to efficiently optimize the ELBO. Tomczak & Welling (2018) instantiate $p_{\theta_2}^Z$ as a Gaussian mixture model; Sønderby et al. (2016), Vahdat & Kautz (2020), and Child (2021) use learnable hierarchical priors; Chen et al. (2017) use normalizing flows; Pang et al. (2020) use energy-based models; and Vahdat et al. (2021) use diffusion models (Section 5.1.2). When $p_{\theta_1}^{X|Z}$ is a flexible enough full-dimensional density as in Equation 22, all these models remain susceptible to the manifold overfitting issues discussed in Section 4.1 and Section 4.1.2, despite directly encouraging the encoder to learn representations whose distribution can be easily recovered by $p_{\theta_2}^Z$.

Even though designing an end-to-end objective for training in a manifold-aware fashion is intuitively desirable, doing so is not always straightforward. To see why, consider a two-step model whose first-step loss is given by Equation 83, and whose second-step model is trained by minimizing $\mathbb{D}(q_{\phi^*}^Z, p_{\theta_2}^Z)$ over $\theta_2$ for some divergence $\mathbb{D}$ between probability distributions. Let us further assume that $\mathbb{D}(q_{\phi^*}^Z, p_{\theta_2}^Z)$ cannot be computed without evaluating $q_{\phi^*}^Z$, but that $\mathbb{D}(q_{\phi^*}^Z, p_{\theta_2}^Z) = \mathcal{L}(p_{\theta_2}^Z; q_{\phi^*}^Z) + c(q_{\phi^*}^Z)$, where $\mathcal{L}(p_{\theta_2}^Z; q_{\phi^*}^Z)$ can be computed without evaluating $q_{\phi^*}^Z$, and where $c(q_{\phi^*}^Z)$ does not depend on $p_{\theta_2}^Z$. Divergences with these properties are prevalent; the KL divergence (Section 3.4) is an instance, where $\mathcal{L}(p_{\theta_2}^Z; q_{\phi^*}^Z) = -\mathbb{E}_{Z \sim q_{\phi^*}^Z}[\log p_{\theta_2}^Z(Z)]$ and $c(q_{\phi^*}^Z) = \mathbb{E}_{Z \sim q_{\phi^*}^Z}[\log q_{\phi^*}^Z(Z)]$, as well as the Fisher divergence (Section 4.3) which underpins score matching

and thus diffusion models. In this case, the second-step model is trained by using the loss $\mathcal{L}(p_{\theta_2}^Z; q_{\phi^*}^Z)$, which is equivalent to minimizing $\mathbb{D}(q_{\phi^*}^Z, p_{\theta_2}^Z)$. Naïvely combining the losses of this two-step model into a single loss for end-to-end training would result in the objective

$$\min_{\theta,\phi} \mathbb{E}_{X \sim p_*^X} \left[ \|X - g_{\theta_1}(f_\phi(X))\|_2^2 \right] + \beta \mathcal{L}\left(p_{\theta_2}^Z; q_\phi^Z\right) \tag{85}$$

for some $\beta > 0$. This naïve objective ignores $c(q_\phi^Z)$, so that it is *not* equivalent to

$$\min_{\theta,\phi} \mathbb{E}_{X \sim p_*^X} \left[ \|X - g_{\theta_1}(f_\phi(X))\|_2^2 \right] + \beta \mathbb{D}\left(q_\phi^Z, p_{\theta_2}^Z\right). \tag{86}$$

In short, Equation 85 does not provide a valid objective for manifold-aware end-to-end training since $c(q_\phi^Z)$ cannot be ignored when $\phi$ is not fixed after the first step of training. Unfortunately, although Equation 86 specifies a principled objective to address the issue, it remains intractable when $c(q_\phi^Z)$ cannot be computed.

---

More formally, a trained autoencoder-based two-step model is given by a distribution $\mathbb{P}_{\theta_2^*}^Z$ on $\mathcal{Z}$ along with a decoder $g_{\theta_1^*} : \mathcal{Z} \to \mathcal{X}$, and the model distribution is given by $\mathbb{P}_{\theta^*}^X = g_{\theta_1^*} \# \mathbb{P}_{\theta_2^*}^Z$. The result of Loaiza-Ganem et al. (2022a) justifying two-step models states that, under mild regularity conditions, if $d = d^*$ and $\mathbb{E}_{X \sim \mathbb{P}_*^X}[\|X - g_{\theta_1^*}(f_{\phi^*}(X))\|_2^2] = 0$, then:

- $\mathbb{Q}_{\phi^*}^Z \ll \lambda_d$, where $\mathbb{Q}_{\phi^*}^Z = f_{\phi^*} \# \mathbb{P}_*^X$ is the distribution of encoded data.

- $g_{\theta_1^*} \# \mathbb{Q}_{\phi^*}^Z = \mathbb{P}_*^X$.

The first point ensures $\mathbb{Q}_{\phi^*}^Z$ admits a density with respect to $\lambda_d$, so that it is full-dimensional. The second point ensures that if the target distribution $\mathbb{Q}_{\phi^*}^Z$ is properly learned during the second step then two-step models recover the true data-generating distribution, i.e. if $\mathbb{P}_{\theta_2^*}^Z = \mathbb{Q}_{\phi^*}^Z$ then $\mathbb{P}_{\theta^*}^X = \mathbb{P}_*^X$.

---

### 5.3.1 Two-Step Models Minimize Wasserstein Distance

Before continuing our review of existing two-step models (Section 5.3), we highlight that these models can be interpreted through an optimal transport lens (Section 3.5). To the best of our knowledge, this observation has not been made in the literature, and constitutes a novel contribution of our work. Here we still consider the model $p_\theta^X$ given by the two learnable components $g_{\theta_1}$ and $p_{\theta_2}^Z$, and will use the notation introduced for Wasserstein autoencoders (Section 5.2.2). Key to our insight is Equation 70, which, for the model $p_\theta^X$ considered here, can be rewritten as

$$\mathbb{W}^c(p_*^X, p_\theta^X) = \inf_{q^{Z|X} \in \mathcal{Q}(p_*^X, p_{\theta_2}^Z)} \mathbb{E}_{X \sim p_*^X} \left[ \mathbb{E}_{Z \sim q^{Z|X}(\cdot|X)} \left[ c(X, g_{\theta_1}(Z)) \right] \right]. \tag{87}$$

This equality then implies that

$$\mathbb{W}^c(p_*^X, p_\theta^X) \leq \inf_{f \in \mathcal{F}(p_*^X, p_{\theta_2}^Z)} \mathbb{E}_{X \sim p_*^X} \left[ c\left( X, g_{\theta_1}(f(X)) \right) \right], \tag{88}$$

where $\mathcal{F}(p_*^X, p_{\theta_2}^Z)$ is the set of functions $f : \mathcal{X} \to \mathcal{Z}$ such that if $X \sim p_*^X$, then $f(X) \sim p_{\theta_2}^Z$. To see that Equation 87 indeed implies Equation 88, simply note that if $f \in \mathcal{F}(p_*^X, p_{\theta_2}^Z)$, then the conditional (on $X = x$) distribution on $\mathcal{Z}$ given by the point mass at $f(x)$ is in $\mathcal{Q}(p_*^X, p_{\theta_2}^Z)$. Equation 88 is used to justify the use of deterministic encoders within WAEs: doing so minimizes an upper bound of $\mathbb{W}^c(p_*^X, p_\theta^X)$. We also point out that, assuming $c(x, y)$ is minimal if and only if $x = y$, the bound becomes tight at optimality as long as perfect reconstructions are achievable with a deterministic autoencoder (i.e. $X = g_{\theta_1^*}(f_{\phi^*}(X))$, see Section 3.2 and Section 5.4 for discussions of when this is possible with continuous autoencoders).

For an encoder $f_\phi$, we let $q_\phi^Z$ be the distribution of $f_\phi(X)$ where $X \sim p_*^X$ (formally, $q_\phi^Z$ is the pushforward density of $p_*^X$ through $f_\phi$), and note that $f_\phi \in \mathcal{F}(p_*^X, p_{\theta_2}^Z)$ is equivalent to $q_\phi^Z = p_{\theta_2}^Z$. In turn, Equation 88

justifies training the model $p_\theta^X$ by minimizing an upper bound of its optimal transport cost through

$$\min_{\theta,\phi} \mathbb{E}_{X \sim p_*^X} \left[ c\left(X, g_{\theta_1}\left(f_\phi(X)\right)\right)\right] \tag{89}$$
$$\text{subject to } q_\phi^Z = p_{\theta_2}^Z.$$

The key difference between this objective and that of WAEs is that the distribution on latent space is now learnable instead of being fixed. While this distinction with WAEs might seem conceptually trivial, it enables minimizing optimal transport cost through a two step process: in the first step, $\theta_1$ and $\phi$ are trained to minimize the reconstruction error, $\mathbb{E}_{X \sim p_*^X}[c(X, g_{\theta_1}(f_\phi(X)))]$, with no regard for the constraint. This step results in a now fixed distribution $q_{\phi^*}^Z$ on $\mathcal{Z}$, which of course need not match $p_{\theta_2}^Z$. Thanks to $p_{\theta_2}^Z$ being learnable and $\theta_2$ not appearing in the first-step objective, this mismatch can be addressed in the second step, where any objective over $\theta_2$ to match $p_{\theta_2}^Z$ and $q_{\phi^*}^Z$ can be used. For example, if the second-step model were trained through maximum-likelihood, its objective would be[26]

$$\min_{\theta_2} \mathbb{KL}\left(q_{\phi^*}^Z \parallel p_{\theta_2}^Z\right), \tag{90}$$

which indeed satisfies the constraint in Equation 89 at optimality. In other words, two-step models solve Equation 89 by optimizing an unconstrained version of the objective during the first step, and then ensuring the constraint is actually satisfied during the second step. Crucially, the second step does not affect the optimality of the first step because $\mathbb{E}_{X \sim p_*^X}[c(X, g_{\theta_1}(f_\phi(X)))]$ does not depend on $\theta_2$, so that two-step models indeed provide a valid way of solving Equation 89.

We now make some additional observations. ($i$) When $c$ is given by the squared Euclidean distance, the corresponding loss for the first-step model is exactly that of a standard autoencoder (Equation 6 and Equation 83). When the first-step model is trained through a different autoencoder-based objective, e.g. Equation 24, we can simply interpret the model as a regularized autoencoder as long as it encourages perfect reconstructions at optimality. ($ii$) Although two-step models are often trained using deterministic encoders, using arbitrarily flexible stochastic encoders would imply that $\mathbb{W}^c(p_*^X, p_\theta^X)$ is being minimized rather than an upper bound.

In summary, we have justified the manifold-awareness of autoencoder-based two-step models through optimal transport. We believe that this result is not only interesting on its own, but also hope that by establishing a connection between seemingly unrelated manifold-aware model classes – namely two-step models and those which are trained through support-agnostic optimization objectives (Section 5.2), such as WAEs – it will enable future improvements to both.

Equation 88 is formalized as follows:

$$\mathbb{W}^c(\mathbb{P}_*^X, \mathbb{P}_\theta^X) \leq \inf_{f \in \mathcal{F}(\mathbb{P}_*^X, \mathbb{P}^Z)} \mathbb{E}_{X \sim p_*^X}\left[c\left(X, g_\theta(f(X))\right)\right], \tag{91}$$

where $\mathcal{F}(\mathbb{P}_*^X, \mathbb{P}^Z) \coloneqq \{f : \mathcal{X} \to \mathcal{Z} \mid f \text{ is measurable, and } f_\# \mathbb{P}_*^X = \mathbb{P}^Z\}$. As a technical point, note that the unconstrained version of the right hand side of this equation – upon which the interpretation of two-step models as Wasserstein distance minimizers is based – i.e.

$$\inf_{f \in \mathcal{F}} \mathbb{E}_{X \sim \mathbb{P}_*^X}\left[c\left(X, g_{\theta_1}\left(f(X)\right)\right)\right], \tag{92}$$

where $\mathcal{F} \coloneqq \{f : \mathcal{X} \to \mathcal{Z} \mid f \text{ is measurable}\}$, involves an infimum over measurable functions (which is then minimized over $\theta_1$ during the first step). In the nonparametric regime (Section 2.2) we assume that neural networks are flexible enough to approximate any continuous function arbitrarily well, but this property need not *a priori* extend to measurable functions. Intuitively this should however not be a problem thanks to Lusin's theorem – which, informally, states that in certain settings any measurable function can be approximated by a continuous one. It is nonetheless pertinent to show that the infimum in Equation 92 can actually be replaced by a corresponding infimum over

---

[26]Note that the second-step objective could itself require additional auxiliary parameters but we omit this possibility for notational simplicity.

continuous functions, which we do below.

**Proposition 1.** *Let $\mathbb{P}_*^X$ be a probability measure on $\mathcal{X}$, $g_{\theta_1} : \mathcal{Z} \to \mathcal{X}$ be measurable, and $c : \mathcal{X} \times \mathcal{X} \to \mathbb{R}$ be measurable and such that there exists $C > 0$ such that*

$$\sup_{(x,y) \in \mathcal{X} \times \mathcal{X}} |c(x,y)| < C. \tag{93}$$

*Then,*

$$\inf_{f \in \mathcal{F}} \mathbb{E}_{X \sim \mathbb{P}_*^X} \left[ c\left( X, g_{\theta_1}\left( f(X) \right) \right) \right] = \inf_{f \in \mathcal{C}} \mathbb{E}_{X \sim \mathbb{P}_*^X} \left[ c\left( X, g_{\theta_1}\left( f(X) \right) \right) \right], \tag{94}$$

*where $\mathcal{F} := \{ f : \mathcal{X} \to \mathcal{Z} \mid f \text{ is measurable} \}$ and $\mathcal{C} := \{ f : \mathcal{X} \to \mathcal{Z} \mid f \text{ is continuous} \}$.*

*Proof.* See Appendix B.2. $\qquad\square$

This result allows us to formally interpret two-step models as minimizers of an upper bound of the Wasserstein distance in the nonparametric regime when the assumption in Equation 93 holds. We point out that this is a mild regularity condition, as it is always satisfied in the common case where $\mathcal{X}$ is compact and $c$ is continuous (since continuous functions always achieve their supremums over compact sets).

Finally, we point out that Patrini et al. (2020) claimed that, as long as $\mathbb{P}_*^X$ is non-atomic, then the inequality in Equation 91 is actually an equality. This would allow us to interpret two-step models as minimizing Wasserstein distance – not an upper bound – even when using deterministic encoders. However, Lee et al. (2024) found an error in the proof of Patrini et al. (2020), so that only the upper bound interpretation remains valid.

### 5.3.2 Latent Diffusion Models

Latent diffusion models (Rombach et al., 2022; Peebles & Xie, 2023; Zhang et al., 2024) are another class of two-step models (Section 5.3). They first train a regularized autoencoder, which combines a Gaussian variational autoencoder objective (Section 4.1.2) with various potential regularizers (Larsen et al., 2016; Higgins et al., 2017; van den Oord et al., 2017). Once this autoencoder-based model is trained and the corresponding low-dimensional representations obtained, a diffusion model (Section 5.1.2) $s_{\theta_2}^Z : \mathcal{Z} \times (0, T] \to \mathcal{Z}$ is trained on them as the second-step model.

As previously discussed, diffusion models can learn manifolds, but their score function must diverge to infinity at the end of the backward process (Equation 52) as a consequence of the mismatch between the intrinsic and ambient dimensions of the data. To test that latent diffusion models are not as sensitive to this numerical pathology, we trained a diffusion model and a latent diffusion model on the CIFAR-10 dataset (Krizhevsky & Hinton, 2009), with all experimental details provided in Appendix C. We plot the average squared Euclidean norm of the score functions, normalized by their dimension,[27] along generated paths in Figure 8: it is evident that the score function of diffusion models on latent space exhibits much better numerical behaviour than when these models are trained on ambient space. This result is suggested by theory, and to the best of our knowledge, we are the first to empirically confirm it.

Currently, latent diffusion models are amongst the best performing DGMs empirically, and the manifold lens provides a convincing explanation for this: (*i*) they can learn manifolds; (*ii*) they alleviate the numerical issues of diffusion models; and (*iii*) they are robust to misspecification of the dimension of the latent space, in the sense that even if $d^* < d$ (i.e. the latent dimension is specified as larger than the true intrinsic dimension), they still learn their target distribution – albeit with an exploding score function. To see this, simply note that setting $d^* < d$ results in a second-step model whose target distribution $q_{\phi^*}^Z$ is still supported

---

[27]Note that normalizing squared Euclidean norm by dimension is the most natural way of enabling comparisons across dimensions. To see this consider a constant vector with all entries equal to 1, which has a squared $\ell_2$ norm equal to its dimension; or consider a standard Gaussian vector, whose expected squared Euclidean norm also matches its dimension.

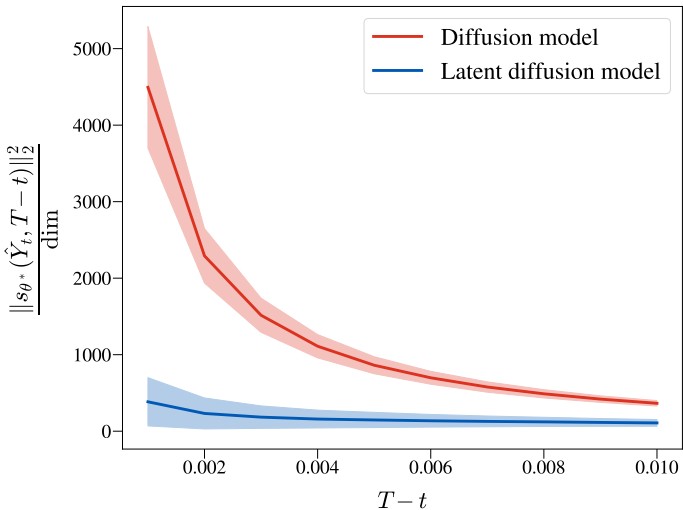

Figure 8: Average squared norm of the learned score function on CIFAR-10 over 100 generated paths from Equation 52, normalized by dimension (i.e. $\texttt{dim} = D = 3072$ for diffusion models, and $\texttt{dim} = d = 256$ for latent diffusion models); the shaded area corresponds to one standard deviation. The paths are stopped at time $T - \varepsilon$, with $\varepsilon = 0.001$.

on a manifold $f_{\phi^*}(\mathcal{M})$ of lower-than-ambient dimension, which the diffusion model from the second step can still learn. Although in this case the score function would also diverge to infinity, the fact that the difference between the ambient and intrinsic dimensions for the latent model $(d - d^*)$ remains much smaller than for a model on ambient space $(D - d^*)$ intuitively suggests that the numerical issues should nonetheless be diminished for latent diffusion models. Indeed, despite the latent diffusion model shown in Figure 8 using $d = 256$, which is likely larger than the true intrinsic dimension $d^*$ of CIFAR-10 (Pope et al., 2021), it has a numerically much better behaved score function than the diffusion model on ambient space.

### 5.3.3 Injective Normalizing Flows

Ordinary normalizing flows (Section 4.1.3) are full-dimensional density models with $D$-dimensional latent spaces, conflicting with the $d^*$-dimensional nature of $p_*^X$. To correct this mismatch, a line of research started by Kumar et al. (2020) has proposed to shrink the NF's latent space dimensionality to $d < D$, allowing the model to represent densities on a low-dimensional submanifold of $\mathcal{X}$. Whereas standard NF architectures must be bijective, $g_{\theta_1}$ now cannot be bijective because it maps from $d$ to $D$ dimensions. The most one can ask is that $g_{\theta_1}$ be *injective*, resulting in the injective normalizing flow (INF).

Brehmer & Cranmer (2020) enforce injectivity architecturally, by constructing $g_{\theta_1}$ as a zero-padding operation followed by a $D$-dimensional NF, in which case the left inverse $f_{\theta_1}$ is given by inverting this NF and applying a projection operation; this is the same construction as the one used by denoising NFs (Section 5.1.4).[28] Injectivity enables density evaluation through the injective change-of-variables formula (Equation 9),

$$p_\theta^X(x) = p_{\theta_2}^Z(z) \left| \det \left( \nabla_z g_{\theta_1}(z)^\top \nabla_z g_{\theta_1}(z) \right) \right|^{-\frac{1}{2}}, \tag{95}$$

where $z = f_{\theta_1}(x)$. Unlike standard NFs, INFs cannot be naïvely trained through maximum-likelihood for two main reasons: $(i)$ the involved determinant is much more computationally challenging to compute and optimize than in standard NFs, and more importantly $(ii)$ Brehmer & Cranmer (2020) showed that doing so would result in pathological solutions. To understand why, recall that Equation 95 is only valid when $x \in g_{\theta_1}(\mathcal{Z})$. When $x$ lies outside $g_{\theta_1}(\mathcal{Z})$, the right hand side of Equation 95 evaluates to $p_\theta^X(\hat{x}_{\theta_1}(x))$, where $\hat{x}_{\theta_1}(x) \coloneqq g_{\theta_1}(f_{\theta_1}(x))$ can be thought of as a projection of $x$ onto $g_{\theta_1}(\mathcal{Z})$. Since $g_{\theta_1}(\mathcal{Z})$ need not perfectly match the data

---

[28]Like in standard NFs, here the encoder $f_{\theta_1}$ is determined by the decoder $g_{\theta_1}$, and it is thus parameterized by the same parameters (i.e. $\theta_1$), so no auxiliary parameters $\phi$ are needed to parameterize it.

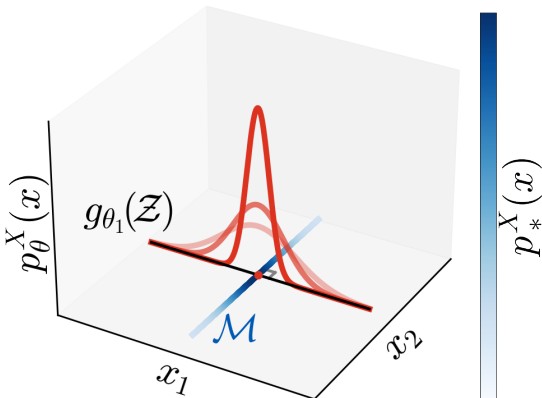

Figure 9: Pathology of naïvely maximizing Equation 95. In this case, the true density is a standard Gaussian along $\mathcal{M} = \{(0, x_2) \in \mathbb{R}^2 \mid x_2 \in \mathbb{R}\}$, here shown in blue with darker values indicating higher density. The model can maximize likelihood by aligning $g_{\theta_1}(\mathcal{Z})$ perpendicular to $\mathcal{M}$, and then learning a density along $g_{\theta_1}(\mathcal{Z})$ that becomes infinitely peaked at the projection $\hat{x}_{\theta_1}(x)$ onto $\mathcal{M}$; in this example the projection will always lie at the origin for any $x \in \mathbb{R}^2$. In the figure, we show $g_{\theta_1}(\mathcal{Z}) = \{(x_1, 0) \in \mathbb{R}^2 \mid x_1 \in \mathbb{R}\}$ with the black line, the projection with the red dot, and increasing peakedness of $p_\theta^X$ with increasing opacity.

manifold $\mathcal{M}$, attempting to maximize $\mathbb{E}_{X \sim p_*^X}[\log p_{\theta_2}^Z(f_{\theta_1}(X)) - \frac{1}{2}\log|\det(\nabla_z g_{\theta_1}(f_{\theta_1}(X))^\top \nabla_z g_{\theta_1}(f_{\theta_1}(X)))|]$ over $\theta$ would thus result in maximizing the likelihood of projected data. As illustrated in Figure 9, this objective can admit pathological solutions where the projections collapse onto a single point whose likelihood is sent to infinity.

To circumvent this issue, Brehmer & Cranmer (2020) propose to train INFs as two-step models (Section 5.3), where $g_{\theta_1}$ and $f_{\theta_1}$ are trained to minimize an $\ell_2$ reconstruction error as in Equation 83; this training procedure also obviates the need to optimize through the determinant in Equation 95. Brehmer & Cranmer (2020) then instantiate $p_{\theta_2}^Z$ as a $d$-dimensional NF, which is trained on encoded data $f_{\theta_1^*}(X)$, where $X \sim p_*^X$. Kothari et al. (2021) follow up on this work by proposing a more efficient architecture for $g_{\theta_1}$. Kumar et al. (2020) originally proposed relaxed INFs, for which the encoder $f_\phi$ is parameterized separately (and encouraged to invert the decoder on $\mathcal{M}$ through a reconstruction error), and where injectivity is instead encouraged by regularizing the singular values of $\nabla_z g_{\theta_1}(f_\phi(X))$ for $X \sim p_*^X$. They then take the second-step density $p_{\theta_2}^Z$ as a Gaussian mixture model. By virtue of being two-step models, all these DGMs are manifold-aware.

**End-to-end training** As mentioned in Section 5.3, it is intuitively desirable to find an end-to-end objective to train two-step models, and INFs provide particularly interesting opportunities. To circumvent the pathological behaviour of naïve maximum-likelihood training of INFs outlined above, several works encourage perfect reconstructions with the goal of ensuring that the model manifold matches the true data manifold:

$$\max_\theta \mathbb{E}_{X \sim p_*^X}\left[\log p_{\theta_2}^Z(f_{\theta_1}(X)) - \frac{1}{2}\log\left|\det\left(\nabla_z g_{\theta_1}(f_{\theta_1}(X))^\top \nabla_z g_{\theta_1}(f_{\theta_1}(X))\right)\right|\right]$$
$$\text{subject to } \mathbb{E}_{X \sim p_*^X}\left[\|X - g_{\theta_1}(f_{\theta_1}(X))\|_2^2\right] = 0. \tag{96}$$

In practice, the constraint can be encouraged by adding an $\ell_2$ regularization term to the likelihood,

$$\max_\theta \mathbb{E}_{X \sim p_*^X}\left[\log p_{\theta_2}^Z(f_{\theta_1}(X)) - \frac{1}{2}\log\left|\det\left(\nabla_z g_\theta(f_{\theta_1}(X))^\top \nabla_z g_{\theta_1}(f_{\theta_1}(X))\right)\right| - \beta\|X - g_{\theta_1}(f_{\theta_1}(X))\|_2^2\right], \tag{97}$$

where $\beta > 0$ is a hyperparameter. Much like how normalizing flows focus on tractability of the log-det-Jacobian term, a central theme of end-to-end injective flows is tractability of the second term in Equation 97, which we will refer to as the "log-det-$J^\top J$" term. One approach is to impose structural constraints on $g_{\theta_1}$ to make the log-det-$J^\top J$ term easily computable, albeit at the cost of expressiveness, for example using conformal embeddings (Ross & Cresswell, 2021). A concurrent approach, pursued by Caterini et al. (2021b), is to approximate the gradient of the log-det-$J^\top J$ term with respect to $\theta_1$ through a combination of Hutchinson's

estimator (Hutchinson, 1989) and various tricks from linear algebra and automatic differentiation (Baydin et al., 2018). Despite these approximations, tractability remains an issue with this technique and it struggles to scale to datasets of higher ambient dimensionality than CIFAR-10 (Krizhevsky & Hinton, 2009).

Denoising NFs (Section 5.1.4) perform single-step training of a flow with an injective component for generation, but with a full-dimensional model $p_\theta^{X_\sigma}$ of a noised-out data density $p_*^{X_\sigma} := p_*^X \circledast \mathcal{N}(\,\cdot\,; 0, \sigma^2 I_D)$. The objective for denoising NFs (Equation 63) ends up quite similar to Equation 97, although with two important differences: $(i)$ the expectation is over $p_*^{X_\sigma}$ rather than $p_*^X$, and $(ii)$ the formulation of $p_\theta^{X_\sigma}$ as a model for $p_*^{X_\sigma}$ eliminates the need to optimize over the costly log-det-$J^\top J$ term. However, the computational benefit comes at the cost of introducing a disconnect between the injective generator and the full-dimensional density model. In particular, the numerical instabilities described in Section 4.1.1 do not apply to INFs trained through Equation 97 since no full-dimensional densities are involved, whereas they do apply denoising NFs.

Cunningham et al. (2022) and Flouris & Konukoglu (2023) both propose injective variants of the flow models described in Section 5.2.5 using similar objectives to Caterini et al. (2021b). Cunningham et al. (2022) in particular use a regularizer that, with a certain hyperparameter setting, cancels out the log-det-$J^\top J$ term in the likelihood, making likelihood-based optimization of injective flows more efficient.

Meanwhile, several works parameterize the encoder $f_\phi$ separately, as Kumar et al. (2020) did. Zhang et al. (2020b) proposed a similar objective to Equation 97 in the context of variational autoencoders (Section 4.1.2). Sorrenson et al. (2024b) argue that the objective in Equation 97 is subject to similar pathologies to those outlined by Brehmer & Cranmer (2020) for naïve maximum-likelihood training if the encoder and decoder are flexible enough. More specifically, they contend that Equation 97 can be made pathologically large by learning a manifold with arbitrarily large curvature rather than $\mathcal{M}$. This is interesting as it highlights that despite being directly motivated to account for the low-dimensional structure of the data, INFs trained through Equation 97 can nonetheless still fail to learn manifolds. Sorrenson et al. (2024b) propose an intuitively well-motivated but theoretically ad-hoc modification to the objective, along with an improved gradient estimator over that of Caterini et al. (2021b), for end-to-end training of INFs. Their model no longer falls under the umbrella of two-step models, and hence its ability to learn the manifold is unclear, but it does inherit computational benefits as it circumvents calculation of the challenging log-det-$J^\top J$ term (Brehmer & Cranmer, 2020). Finally, Nazari et al. (2023) proposed an autoencoder for manifold learning (not generative modelling) which penalizes the variance of the log-det-$J^\top J$ term during training, and which results in a smoother and more interpretable latent space as compared to standard autoencoders.

## 5.4 Overcoming Topological Obstacles to Manifold Learning

As mentioned in Section 5.3, one might want to set the latent dimension $d$ of an autoencoder to $d^*$. Yet, as discussed in Section 3.2, when $d = d^*$, perfect manifold learning is not always achievable through bottleneck methods for topological reasons. We illustrate this problem, which can cause downstream issues with density estimation, in Figure 10(a). In particular, if $\mathcal{M}$ has any non-trivial topological properties such as holes or disconnected components, any attempt to model $\mathcal{M}$ as the image of a decoder $g_\theta$ will cause numerical instability in $g_\theta$ (Cornish et al., 2020; Salmona et al., 2022). This problem was originally identified in the context of normalizing flows (Section 4.1.3), with a line of work that first appends additional dimensions to $\mathcal{X}$, and then trains a normalizing flow on the augmented space (Dupont et al., 2019; Chen et al., 2020; Huang et al., 2020). While these techniques indeed increase the stability and expressiveness of the flow itself, they still produce full-support densities that can never truly model non-trivial topological structures in data. Other works have leveraged tools from the field of topological data analysis (Chazal & Michel, 2021; Barannikov et al., 2022) to explicitly regularize autoencoders with the goal of encouraging $f_\phi(\mathcal{M})$ to share topological properties with $\mathcal{M}$ (Moor et al., 2020; Trofimov et al., 2023). Empirically, these methods have been shown to decrease topological mismatch; yet, as argued in the grey box in Section 3.2, completely eliminating this mismatch is theoretically impossible when the root cause of the problem is the non-existence of a topological embedding of $\mathcal{M}$ into $\mathcal{Z}$, rather than the loss used to train the autoencoder. Alternatively, to more faithfully tackle topological issues, some works have proposed to restructure the manifold-learning step to better reflect the ways manifolds are defined in theory; we cover these approaches in detail below.

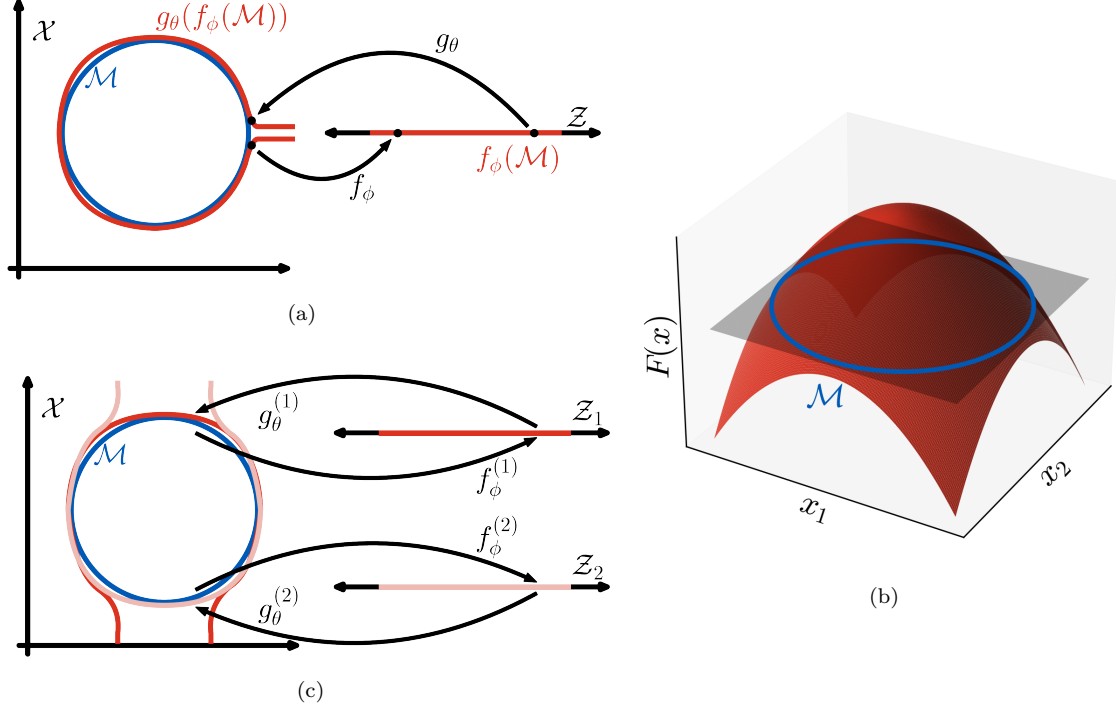

Figure 10: Models for $\mathcal{M} = \{(x_1, x_2) \mid x_1^2 + x_2^2 = 1\}$, the unit circle in $\mathcal{X} = \mathbb{R}^2$. **(a)** Using a single encoder-decoder pair to model the circle. The pair approaches numerical non-invertibility since the encoder $f_\phi$ must map two nearby points in $\mathcal{M}$ (in black) to two distant latent points in $\mathcal{Z}$ (also in black). **(b)** Illustration of implicit manifolds. Here, $\mathcal{M}$ is characterized as the 0-level set of $F : \mathbb{R}^2 \to \mathbb{R}$. Specifically, $F(x_1, x_2) = 1 - (x_1^2 + x_2^2)$ is shown in red, and the grey plane corresponds to $\{x \in \mathbb{R}^3 \mid x_3 = 0\}$. Neural implicit manifolds use no autoencoders, and instead attempt to learn $F_\theta$ so that its 0-level set, $F_\theta^{-1}(\{0\})$, matches $\mathcal{M}$. **(c)** The problem depicted in (a) can also be circumvented by employing multiple encoder-decoder pairs, each one using its own latent space and covering a different part of $\mathcal{M}$.

### 5.4.1   Neural Implicit Manifolds

Under some conditions, manifolds with non-trivial topologies can be defined using level sets of functions. In particular, a level set of a smooth function $F : \mathbb{R}^D \to \mathbb{R}^{D-d^*}$ represents a $d^*$-dimensional manifold if its Jacobian has full rank on that level set (Lee, 2012). We illustrate such an implicitly defined manifold – which cannot be characterized with a single encoder-decoder pair – in Figure 10(b).

Neural implicit manifold learning (Ross et al., 2023) operationalizes this fact by modelling $\mathcal{M}$ as the zero set of a neural network $F_{\theta_1} : \mathcal{X} \to \mathbb{R}^{D-d}$. The network $F_{\theta_1}$ is trained to align its zero set $F_{\theta_1}^{-1}(\{0\})$ with $\mathcal{M}$ while being regularized to have full rank on $\mathcal{M}$. The following loss is used,

$$\min_{\theta_1} \mathbb{E}_{X \sim p_*^X, Y \sim q_{\theta_1'}^X, V \sim \mathcal{U}(\,\cdot\,; \mathcal{S}^{D-d-1})} \left[ \|F_{\theta_1}(X)\|_2 - \alpha \|F_{\theta_1}(Y)\|_2 + \beta \big( \eta - \|V^\top \nabla_x F_{\theta_1}(X)\|_2 \big)_+^2 \right], \tag{98}$$

where: $Y$ is independent of $X$; $q_{\theta_1'}^X(y) \propto e^{-\|F_{\theta_1'}(y)\|_2^2}$ with $\theta_1' = \texttt{stopgrad}(\theta_1)$ (as in energy-based models, see Section 4.1.4); $\mathcal{U}(\,\cdot\,; \mathcal{S}^{D-d-1})$ is the uniform distribution over $\mathcal{S}^{D-d-1} \coloneqq \{y \in \mathbb{R}^{D-d} \mid \|y\|_2 = 1\}$, the $(D-d-1)$-sphere in $\mathbb{R}^{D-d}$; $\alpha > 0$, $\beta > 0$, and $\eta > 0$ are hyperparameters; and $(\,\cdot\,)_+ \coloneqq \max(\,\cdot\,, 0)$. In this loss, the first term ensures that $F_{\theta_1}^{-1}(\{0\})$ contains $\mathcal{M}$, the second term prevents $F_{\theta_1}^{-1}(\{0\})$ from containing off-manifold samples, and the third term regularizes the Jacobian of $F_{\theta_1}$ to have full rank on $\mathcal{M}$.

Importantly, $F_{\theta_1}$ here is not an encoder; it is best interpreted as defining $D - d$ non-linear constraints on the data, thus leaving $d$ degrees of freedom for the data manifold. The full-rank requirement can then be

interpreted as ensuring none of these constraints is redundant, thereby ensuring the learned manifold has the correct dimensionality. This constraint-based learning procedure for $\mathcal{M}$ makes implicit manifold learning unusual among two-step models (Section 5.3) in that it is not autoencoder-based. The lack of encoder in this method means there is no latent space, which presents a challenge for learning the distribution on the model manifold, $F_{\theta_1^*}^{-1}(\{0\})$.

Ross et al. (2023) propose to learn this distribution with the constrained energy-based model, which represents an EBM constrained to the learned manifold, $E : F_{\theta_1^*}^{-1}(\{0\}) \to \mathbb{R}$. This is parameterized in practice by a neural network $E_{\theta_2} : \mathcal{X} \to \mathbb{R}$, for which values are ignored outside of the learned manifold $F_{\theta_1^*}^{-1}(\{0\})$. To sample from constrained EBMs, constrained Langevin dynamics (Brubaker et al., 2012) is used to generate samples from $E_{\theta_2}$ constrained to the manifold. This allows for likelihood maximization via Equation 38, as with ordinary EBMs – except constrained EBMs are manifold-supported.

### 5.4.2 Multi-Chart Manifolds

Typical techniques model the manifold globally using a single encoder-decoder pair. In general, however, manifolds can consist of a patchwork of many *charts*: mathematical objects that each serve roughly the same function as a single encoder-decoder pair (for a formal definition, please see Lee (2012)). Some manifolds may thus require many encoder-decoder pairs $(f_\phi^{(i)}, g_\theta^{(i)})_{i=1}^n$ – each using its latent space $\mathcal{Z}_i$ to locally describe some subset of the manifold – rather than a single one: we illustrate this fact in Figure 10(c).

Several works have taken this route to learn the data manifold. Schonsheck et al. (2019) first proposed a multi-chart latent space using multiple encoder-decoder pairs in a non-generative modelling context. For each incoming datapoint, the correct chart is selected dynamically using a prediction head for the encoder-decoder pair with the smallest reconstruction error. Kalatzis et al. (2021) propose multi-chart flows, in which likelihoods are defined using a mixture of injective normalizing flows (Section 5.3.3), with mixture weights again computed using a prediction head. On the other hand, Sidheekh et al. (2022) propose a mixture of INFs in which chart membership for an incoming datapoint is computed using discrete latent assignments.

Modelling a manifold with multiple charts imposes drawbacks. For one, each encoder-decoder pair has its own latent space, so for a given datapoint $x \in \mathcal{M}$, choosing the correct encoder can be a challenge. This makes it unclear how to perform tasks involving the manipulation of latent representations, such as interpolation. In many cases, there is no single correct encoder, as the images of various decoders need to overlap to correctly define topologically complex manifolds (such as in Figure 10(c)). The ambiguity of choosing the correct encoder is underlined by how differently each of the aforementioned methods attempt to do so.

### 5.4.3 Disconnected Manifolds

Another common source of topological complexity in the data manifold is when it consists of more than one connected component. This situation occurs, for example, in datasets with multiple disjoint classes. In this context, theoretical analyses have shown that any decoder-based model will suffer from training instability (Salmona et al., 2022) and poor sample quality (Luzi et al., 2020).[29] One technique to improve sample quality is to avoid sampling from latent regions where the network $g_{\theta^*}$ is unstable. Tanielian et al. (2020) propose, in the context of generative adversarial networks (Section 4.2 and Section 5.2.1), to reject samples $X = g_{\theta^*}(Z)$, where $Z \sim p^Z$, for which $\nabla_z g_{\theta^*}(Z)$ has a high Frobenius norm, which they show indicates an off-manifold sample. Other work, discussed below, seeks to avoid instability entirely during training.

A few general techniques have been proposed for modelling disconnected manifolds. One way is to use disconnected (or near-disconnected) latent distributions, which aims to match the topology of the support of $p_\theta^Z$ with that of $\mathcal{M}$. This is typically done with a Gaussian mixture model for $p_\theta^Z$ and has been proposed for multiple classes of generative model (Nalisnick et al., 2016; Dilokthanakul et al., 2016; Jiang et al., 2017; Ben-Yosef & Weinshall, 2018; Izmailov et al., 2020).

---

[29]Note that the theoretical analysis of Salmona et al. (2022) shows numerical instability when $p_*^X$ is multimodal, in which case its support $\mathcal{M}$ can be considered as numerically disconnected.

A related approach is to use multiple decoder networks in a similar manner to the aforementioned multi-chart methods from Section 5.4.2. The model then becomes a mixture $p_\theta^X(x) = \sum_{i=1}^n \pi_i p_{\theta,i}^X(x)$ of generative submodels $p_{\theta,i}^X$, where $\pi_1, \ldots, \pi_n$ are the mixture weights (sometimes trainable). For example, Arora et al. (2017) propose to directly train a mixture of generative adversarial networks to stabilize training, wherein the entire mixture is learned with backpropagation. Cornish et al. (2020) use a hierarchical continuously-indexed mixture of normalizing flows (Section 4.1.3). Other work uses techniques based on expectation-maximization (Dempster et al., 1977) to train the mixture (Banijamali et al., 2017; Locatello et al., 2018). A key challenge in this line of work is to encourage different submodels to model distinct parts of the distributions. This can be done by partitioning the data beforehand, by class (Luzi et al., 2020) or through unsupervised clustering (Brown et al., 2023), and training a model on each partition. A more flexible approach is to backpropagate through an ancillary classification model to encourage each submodel to generate data from distinct manifolds (Hoang et al., 2018; Khayatkhoei et al., 2018; Ghosh et al., 2018).

While all the models mentioned above can properly account for some topological features of $\mathcal{M}$ such as disconnectedness, we highlight that most are nonetheless manifold-unaware. For example, full-dimensional models trained through maximum-likelihood remain exposed to the corresponding pathologies (Section 4.1), even when they are mixture models.

# 6  Discrete Deep Generative Models

Since this survey's focus is on the manifold hypothesis, all of the models presented thus far are for continuous distributions. Nonetheless, many DGMs assume that the ambient space $\mathcal{X}$ is discrete. For example, images can be modelled as having pixels which take only finitely many different values, rather than a continuum of them. In this case, $p_*^X$ and $p_\theta^X$ are both probability mass functions over $\mathcal{X}$, and $\mathcal{M} \subset \mathcal{X}$ denotes the support of $p_*^X$. Formally, in this setting $\mathcal{X}$ is a 0-dimensional manifold, so that even when $\mathcal{M}$ is a strict subset of $\mathcal{X}$, it remains a 0-dimensional submanifold. In other words, there can be no dimensionality mismatch for discrete data since $D = d^* = 0$. In turn, this implies that mathematically, discrete likelihood-based DGMs are not exposed to problems such as manifold overfitting (Section 4.1) which arise from dimensionality mismatch. This view of discrete DGMs through the manifold lens is useful, since it suggests that whenever a manifold-unaware DGM admits a straightforward discrete analogue, the latter should be preferred as it will be unaffected by manifold-related woes. Indeed, as mentioned in Section 4.1.2, discrete variational autoencoders (Gulrajani et al., 2017b; Vahdat & Kautz, 2020; Vahdat et al., 2021) empirically outperform their continuous variants. Similarly, discrete incarnations of likelihood-based autoregressive DGMs (Germain et al., 2015; van den Oord et al., 2016; Salimans et al., 2017; Parmar et al., 2018) outperform continuous ones (Uria et al., 2013). In contrast, discrete versions of diffusion models (Austin et al., 2021; Campbell et al., 2022; Meng et al., 2022) do not outperform their manifold-aware continuous counterparts (Section 5.1.2) when modelling images.

Discrete and continuous DGMs nonetheless have similarities, despite the differences outlined above. As discussed in Section 1, a key motivation behind the manifold hypothesis is to capture the intuition that $\mathcal{M}$, the support of $p_*^X$, is somehow sparse within $\mathcal{X}$. This intuition often remains true in the discrete case: using images as an example once again, there are $256^D$ possible discrete images (assuming each pixel entry takes one of 256 possible values), yet the subset of natural images is vanishingly small in comparison and contains orders of magnitude fewer elements. The main idea of continuous two-step models (Section 5.3), namely to first approximate the support of $p_*^X$ and then learn the distribution within, remains equally sensible in the discrete case. van den Oord et al. (2017) proposed an autoencoder which recovers discrete representations over which they train a discrete DGM; this idea that has been further developed, with strong empirical results (Razavi et al., 2019; Esser et al., 2021; Ramesh et al., 2021; Chang et al., 2022). We finish by pointing out that the discussion in Section 5.3.1 applies to all these discrete two-step models, so that they can be interpreted as minimizing a potentially regularized upper bound of the Wasserstein distance between $p_*^X$ and $p_\theta^X$ which becomes tight at optimality, because in the discrete case, perfect reconstructions are always achievable given enough capacity of the encoder and decoder.

# 7 Conclusions and Future Outlook

**Conclusions** In this survey we have carried out a review of deep generative models through the lens of the manifold hypothesis. This viewpoint presents a mathematically elegant perspective of DGMs, and suggests that manifold-awareness is an important necessary condition for strong empirical performance. We thus encourage researchers who are developing new DGMs to consider manifold-awareness as a desideratum, and ask themselves: *Can my deep generative model learn distributions supported on unknown low-dimensional manifolds?* When the answer is yes, demonstrating this fact will strengthen the work's motivation; and when the answer is no, this suggests that the DGM can be improved by endowing it with manifold-awareness – either through a model-specific fix, or at least by training it on latent space as a two-step model (Section 5.3). We also showed that numerical instabilities of likelihood-evaluation are unavoidable in the manifold setting (Section 4.1.1) and that two-step models can be interpreted as minimizing a (potentially regularized) upper bound of the Wasserstein distance objective (Section 5.3.1).

**Future outlook** Finally, we outline a non-exhaustive list of research directions involving deep generative models and their interplay with the manifold hypothesis. We believe these lines of inquiry are interesting, and mostly unexplored at the time of writing:

- **Further understanding dimensionality mismatch** The effects of using a full-dimensional model when the ground truth distribution is manifold-supported are well understood for likelihood-based models (Section 4.1) and diffusion models (Section 5.1.2), yet our grasp of the interplay between DGMs and the manifold hypothesis remains incomplete. For example, a theoretical understanding of score matching (Section 4.3) and conditional flow matching (Section 5.1.3) under misspecified dimension is lacking, as is the effect of using lower-bounded energy functions in energy-based models (Section 4.1.4).

- **Improved training of DGMs with two-step architectures** Two-step models as presented in Section 5.3 are manifold-aware. Yet, as also discussed in Section 5.3, two-step training does not encourage the encoder from the first step to represent the data in a way conducive to distribution learning in the second step. Intuitively, this means there is room for improvement in how these models are trained, and since the end-to-end approaches described at the end of Section 5.3 are in general manifold-unaware, several avenues remain open. For example, despite the existence of regularizers for training autoencoders (Larsen et al., 2016; Higgins et al., 2017; Nazari et al., 2023), there is very little work explicitly designing autoencoders for two-step training. The only work we are aware of in this direction is by Hu et al. (2023), who propose to split the first step into two sub-steps: in the first sub-step the encoder is trained along with a low-capacity decoder, and in the second sub-step the encoder is frozen and a more flexible decoder is trained. Another avenue is finding an end-to-end objective to train this type of model in a manifold-aware fashion. Current end-to-end methods are manifold-unaware, despite providing a desirable inductive bias – an exception being generalized energy-based models (Section 5.2.4) which cannot be readily extended beyond using energy-based models (Section 4.1.4) as the latent distribution. We thus hypothesize that any end-to-end, or improved two-step, manifold-aware procedure which can train diffusion models in latent space while scaling to massive datasets (Schuhmann et al., 2022) is likely to improve upon current commercial versions of latent diffusion models (Section 5.3.2).

- **Extracting and leveraging manifold information** Any manifold-aware DGM which succeeds at learning its target distribution $p_*^X$ must have learned its support $\mathcal{M}$ as well, albeit perhaps implicitly. Extracting information about $\mathcal{M}$ from a trained DGM is thus a natural problem, as is leveraging this information for any practical use. Various works have shown that trained DGMs induce Riemannian metrics over the learned manifolds (Shao et al., 2018; Arvanitidis et al., 2018; Chadebec & Allassonnière, 2022; Sorrenson et al., 2024a), which can in turn be leveraged for interpolating between datapoints and for improved sampling procedures. Several works have also shown that DGMs can be used to estimate the intrinsic dimension of $\mathcal{M}$ (Tempczyk et al., 2022; Zheng et al., 2022; Horvat & Pfister, 2024; Kamkari et al., 2024b; Stanczuk et al., 2024), and these quantities have already proven useful for unsupervised out-of-distribution detection and to identify memorized

samples (Kamkari et al., 2024a; Ross et al., 2024; Humayun et al., 2024). The field of topological data analysis (Chazal & Michel, 2021) aims to extract topological and geometric features of $\mathcal{M}$ – such as intrinsic dimension – from an observed dataset, conventionally without the use of DGMs. Another fruitful direction for future research will involve further exploiting DGMs for topological data analysis.

- **Finite-sample convergence rates** All the analyses presented here assumed the nonparametric regime (Section 2.2). As we have seen throughout our survey, this simplifying assumption enables a useful and practical understanding of DGMs through the lens of the manifold hypothesis. Yet, this assumption remains unrealistic since in practice expectations with respect to $p_*^X$ cannot be computed; $p_*^X$ must thus be approximated via its empirical distribution – i.e. a mixture of (equally weighted) point masses at the (finitely many) observed datapoints. Formally, the empirical distribution is supported on a 0-dimensional submanifold of $\mathcal{X}$ – namely, the observed dataset – so that any flexible enough and sufficiently well optimized manifold-aware DGM should simply memorize its entire training dataset. Evidently manifold-awareness remains a desirable property – statistical consistency under the manifold hypothesis is impossible without it – but the fact that state-of-the-art DGMs do not suffer from total memorization cannot be explained while assuming the nonparametric regime. Thus, understanding what drives DGMs to generalize rather than memorize remains a relevant problem. Kadkhodaie et al. (2024) study these questions for diffusion models through the lens of the inductive biases provided through the architecture of the score network. More formal explanations of generalization are provided by statistical learning theory in the form of finite-sample convergence rates. Although these results often do not assume the manifold hypothesis, a recent line of work has, obtaining in turn much faster convergence rates which depend on intrinsic rather than ambient dimension (Schreuder et al., 2021; Huang et al., 2022; Dahal et al., 2022; Tang & Yang, 2023; Chae et al., 2023; Chen et al., 2023; Oko et al., 2023; Chakraborty & Bartlett, 2024b;a; Hu et al., 2024; Vardanyan et al., 2024; Tang & Yang, 2024). We believe that this research direction provides a challenging but highly promising avenue for a full theoretical understanding of DGMs.

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

# A   Weak Convergence Primer

We now provide a brief summary of weak convergence of probability measures. We do not use a grey box around this section due to its length, despite the content being fairly technical. As in the main text, all the measures we consider here will be defined on $\mathcal{X}$ along with its Borel $\sigma$-algebra. Given a sequence of probability measures $(\mathbb{P}^X_{\theta_t})^\infty_{t=1}$, we would like to define what it means for the sequence to converge to some probability measure $\mathbb{P}^X_\dagger$. As we will see, there are several ways to define convergence of probability measures; we would like to use one which captures the intuition that $\mathbb{P}^X_{\theta_t}$ "learns" $\mathbb{P}^X_\dagger$ in the sense that $\mathbb{P}^X_{\theta_t}$ converges to $\mathbb{P}^X_\dagger$ as $t \to \infty$ if and only if samples from $\mathbb{P}^X_{\theta_t}$ become progressively harder to distinguish from those of $\mathbb{P}^X_\dagger$ as $t$ becomes larger, becoming indistinguishable in the limit. When $\mathbb{P}^X_{\theta_t}$ represents a DGM, this is precisely the form of convergence we would hope to observe as we optimize its parameters $\theta_t$.

**Strong convergence**   The seemingly most natural way to define convergence is to say that $\mathbb{P}^X_{\theta_t}$ converges to $\mathbb{P}^X_\dagger$ as $t \to \infty$ if $\mathbb{P}^X_{\theta_t}(B) \to \mathbb{P}^X_\dagger(B)$ as $t \to \infty$ for every Borel set $B$. This type of convergence is called *strong convergence*. As the name suggests, this type of convergence is too strong, to the point where it does not properly capture the intended intuition of "$\mathbb{P}^X_{\theta_t}$ converges to $\mathbb{P}^X_\dagger$ if and only if $\mathbb{P}^X_{\theta_t}$ learns $\mathbb{P}^X_\dagger$".

Let us illustrate why this is the case with two examples. First, let $\mathbb{P}^X_{\theta_t}$ be Gaussian with mean 0 and covariance matrix $1/t\, I_D$. Intuitively, this sequence learns a point mass at 0, $\delta_0$, yet it does not strongly converge to it: $\{0\}$ is a Borel set, and $\mathbb{P}^X_{\theta_t}(\{0\}) = 0 \to 0$ as $t \to \infty$, yet $\delta_0(\{0\}) = 1 \neq 0$. As a second example, consider $\mathbb{P}^X_{\theta_t} = \delta_{x_t}$, where $(x_t)^\infty_{t=1}$ is a fixed sequence converging to 0 with $\{x_t\}^\infty_{t=1} \cap \{0\} = \emptyset$. Intuitively this sequence also learns $\delta_0$, but similarly to the previous example, $\mathbb{P}^X_{\theta_t}(\{0\}) = 0$ for every $t$ and thus the sequence does not strongly converge to $\delta_0$ either.

We highlight that these are not overly-contrived examples in the DGM setting. The first example illustrates a common scenario where a sequence of full-dimensional models $\mathbb{P}^X_{\theta_t} \ll \lambda_D$ "learn" a distribution $\mathbb{P}^X_\dagger$ supported on a low-dimensional embedded submanifold $\mathcal{M}$ of $\mathcal{X}$, without strongly converging to $\mathbb{P}^X_\dagger$. In this case, $\mathcal{M}$ is also a Borel set, and $\mathbb{P}^X_{\theta_t}(\mathcal{M}) = 0 \to 0$ as $t \to \infty$ even though $\mathbb{P}^X_\dagger(\mathcal{M}) = 1 \neq 0$. The second example illustrates how a sequence of models whose supports do not overlap with that of their target distribution can "learn" it without strongly converging to it. Indeed, we need a laxer definition of convergence of probability measures to properly convey the idea that a sequence of models $\mathbb{P}^X_{\theta_t}$ "learns" $\mathbb{P}^X_\dagger$.

**Weak convergence**   Weak – rather than strong – convergence provides a more appropriate notion of convergence to convey "learning". We say that $\mathbb{P}^X_{\theta_t}$ *converges weakly* to $\mathbb{P}^X_\dagger$ if $\mathbb{E}_{X \sim \mathbb{P}^X_{\theta_t}}[h(X)] \to \mathbb{E}_{X \sim \mathbb{P}^X_\dagger}[h(X)]$ as $t \to \infty$ for every bounded and continuous function $h : \mathcal{X} \to \mathbb{R}$. As mentioned in Section 2.1, we write $\mathbb{P}^X_{\theta_t} \xrightarrow{\omega} \mathbb{P}^X_\dagger$ as $t \to \infty$ to denote weak convergence. Intuitively, $\mathbb{P}^X_{\theta_t}$ converges weakly to $\mathbb{P}^X_\dagger$ if, as $t \to \infty$, it becomes arbitrarily difficult to distinguish between samples from $\mathbb{P}^X_{\theta_t}$ and samples from $\mathbb{P}^X_\dagger$ by using a bounded and continuous function; weak convergence matches the intuition of $\mathbb{P}^X_{\theta_t}$ "learning" $\mathbb{P}^X_\dagger$ much better than strong convergence.

There are many equivalent definitions of weak convergence, with the standard one being the one presented above. The result establishing the equivalence of these definitions is called the Portmanteau Lemma. We present a reduced version of this lemma below – which we will use to prove the Likelihood Instability Theorem in Appendix B.1 – where only one of these equivalences is stated. Before stating the lemma, we define the continuity sets of a probability measure.

**Definition 1** (Continuity Set). *Let $\mathbb{P}^X_\dagger$ be a probability measure on $\mathcal{X}$ and $B \subset \mathcal{X}$ a Borel set. We say that $B$ is a* continuity set *of $\mathbb{P}^X_\dagger$ if $\mathbb{P}^X_\dagger(\partial_\mathcal{X} B) = 0$, where $\partial_\mathcal{X} B$ denotes the topological boundary of $B$ on $\mathcal{X}$.*

**Lemma 1** (Portmanteau). *Let $\mathbb{P}^X_\dagger$ be a probability measure on $\mathcal{X}$, and let $(\mathbb{P}^X_{\theta_t})^\infty_{t=1}$ be a sequence of probability measures on $\mathcal{X}$. Then, $\mathbb{P}^X_{\theta_t} \xrightarrow{\omega} \mathbb{P}^X_\dagger$ as $t \to \infty$ if and only if $\mathbb{P}^X_{\theta_t}(B) \to \mathbb{P}^X_\dagger(B)$ as $t \to \infty$ for every continuity set $B$ of $\mathbb{P}^X_\dagger$.*

Let us consider once again the example where $\mathbb{P}^X_{\theta_t}$ is Gaussian with mean 0 and covariance matrix $1/t\, I_D$. It is not difficult to prove that $\mathbb{P}^X_{\theta_t}(B) \to \delta_0(B)$ as $t \to \infty$ for every Borel set $B$ such that $0 \notin \partial_\mathcal{X} B$ (i.e.

$\delta_0(\partial_\mathcal{X} B) = 0$), so that as intended, $\mathbb{P}^X_{\theta_t} \xrightarrow{\omega} \delta_0$ as $t \to \infty$. Similarly, it is not difficult to prove the same in the example where $\mathbb{P}^X_{\theta_t} = \delta_{x_t}$. The fact that these sequences converge weakly but not strongly to $\delta_0$ illustrates that weak convergence does indeed provide the right tool to talk about a sequence of models "learning" their target distribution.

**Metrizing weak convergence** Finally, we say that a metric $\mathbb{D} : \Delta(\mathcal{X}) \times \Delta(\mathcal{X}) \to \mathbb{R}$ on the space of probability measures on $\mathcal{X}$ *metrizes weak convergence* if $\mathbb{D}(\mathbb{P}^X_{\theta_t}, \mathbb{P}^X_\dagger) \to 0$ as $t \to 0$ holds if and only if $\mathbb{P}^X_{\theta_t} \xrightarrow{\omega} \mathbb{P}^X_\dagger$ as $t \to \infty$. Throughout the main manuscript, we often abuse language and use the term "metrizing weak convergence" even when $\mathbb{D}$ is only a divergence rather than a metric, as this is enough to ensure that minimizing $\mathbb{D}(\mathbb{P}^X_\theta, \mathbb{P}^X_*)$ over $\theta$ is a sensible training objective for a DGM $\mathbb{P}^X_\theta$ – even if $\mathbb{P}^X_*$ has low-dimensional support.

# B Proofs

As in Appendix A, we omit the use of a grey box despite the use of technical language.

## B.1 The Likelihood Instability Theorem

We start by restating the Likelihood Instability Theorem for convenience before discussing it.

**Theorem 1** (Likelihood Instability of Deep Generative Models). *Let $M \subset \mathcal{X}$ be a Borel set such that $\lambda_D(\mathrm{cl}_\mathcal{X}(M)) = 0$, and let $\mathbb{P}^X_\dagger$ be a probability measure on $\mathcal{X}$ such that $\mathbb{P}^X_\dagger(M) = 1$ and $\mathrm{supp}(\mathbb{P}^X_\dagger) = \mathrm{cl}_\mathcal{X}(M)$. Let $(\mathbb{P}^X_{\theta_t})^\infty_{t=1}$ be a sequence of probability measures on $\mathcal{X}$ such that $\mathbb{P}^X_{\theta_t} \xrightarrow{\omega} \mathbb{P}^X_\dagger$ as $t \to \infty$ and $\mathbb{P}^X_{\theta_t} \ll \lambda_D$, with corresponding densities $p^X_{\theta_t}$. Then:*

- $\liminf_{t \to \infty} p^X_{\theta_t}(x) = 0$, $\lambda_D$-*almost-everywhere on* $\mathcal{X} \setminus \mathrm{cl}_\mathcal{X}(M)$.

- $\sup_{x' \in B_\varepsilon(x)} p^X_{\theta_t}(x') \to \infty$ *as* $t \to \infty$ *for every* $x \in \mathrm{cl}_\mathcal{X}(M)$ *and every* $\varepsilon > 0$, *where* $B_\varepsilon(x) := \{x' \in \mathcal{X} \mid \|x' - x\|_2 < \varepsilon\}$.

As mentioned in Section 4.1.1 we begin with an example satisfying the assumptions of the Likelihood Instability Theorem for which it does not hold that $p^X_{\theta_t}(x) \to \infty$ for $x \in \mathrm{cl}_\mathcal{X}(M)$, thus highlighting that the theorem cannot be "trivially strengthened". Consider $\mathcal{X} = \mathbb{R}^2$, $M = \{(x_1, 0) \in \mathbb{R}^2 \mid 0 < x_1 < 1\}$, let $\mathbb{P}^X_\dagger$ be uniform on $M$, and $\mathbb{P}^X_{\theta_t}$ be uniform on $M_t$, where $M_t = \{(x_1, x_2) \in \mathbb{R}^2 \mid 0 < x_1 < 1 \text{ and } 1/(t+1) < x_2 < 1/t\}$. Finally, take the corresponding densities as

$$p^X_{\theta_t}(x) = \frac{\mathbb{1}(x \in M_t)}{\lambda_2(M_t)} = t(t+1)\mathbb{1}(x \in M_t), \tag{99}$$

where $\mathbb{1}(\cdot)$ denotes an indicator function. Figure 11 illustrates this example. Here, it holds that $\mathbb{P}^X_{\theta_t} \xrightarrow{\omega} \mathbb{P}^X_\dagger$ – so that the assumptions of the Likelihood Instability Theorem are satisfied – yet $p^X_{\theta_t}(x) \to 0$ as $t \to \infty$ for every $x \in \mathcal{X}$, and thus in particular for every $x \in \mathrm{cl}_\mathcal{X}(M)$ as well. Nonetheless, when $x \in \mathrm{cl}_\mathcal{X}(M)$, $\sup_{x' \in B_\varepsilon(x)} p^X_{\theta_t}(x') \to \infty$ as $t \to \infty$ does hold for every $\varepsilon > 0$, as concluded by the theorem.

Before proving the Likelihood Instability Theorem, we state and prove three lemmas, all of which we will rely on. We will heavily use Continuity Sets and will leverage the Portmanteau Lemma; see Appendix A for a reminder on these topics.

**Lemma 2.** *Let $\mathbb{P}^X_\dagger$ be a probability measure on $\mathcal{X}$, $x \in \mathcal{X}$, and $\varepsilon > 0$. Then, there exists $\varepsilon' \in (0, \varepsilon)$ such that $B_{\varepsilon'}(x)$ is a continuity set of $\mathbb{P}^X_\dagger$, where $B_\varepsilon(x) = \{x' \in \mathcal{X} \mid \|x' - x\|_2 < \varepsilon\}$.*

*Proof.* We proceed by contradiction: let $x \in \mathcal{X}$ and $\varepsilon > 0$, and assume that $\mathbb{P}^X_\dagger(\partial_\mathcal{X} B_{\varepsilon'}(x)) > 0$ for every $\varepsilon' \in (0, \varepsilon)$. Since we can countably partition $(0, 1]$ as $\cup^\infty_{n=2}(1/n, 1/(n-1)]$, it follows that uncountably

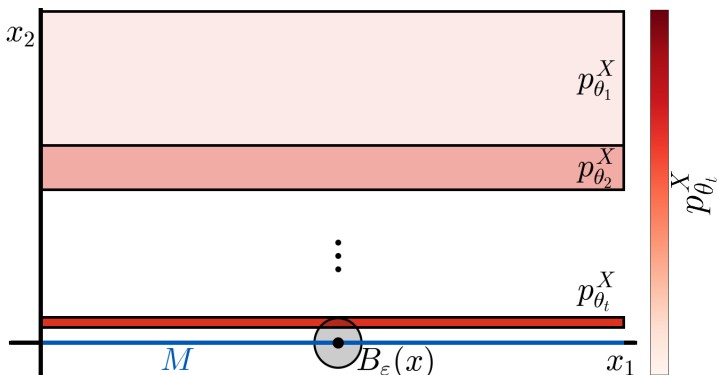

Figure 11: Visualization of the sequence of densities from Equation 99, which converge weakly to a uniform distribution on $M$. For $x \in \mathrm{cl}_{\mathcal{X}}(M)$ it always holds that $p_{\theta_t}^X(x) = 0$ because the support of $p_{\theta_t}^X$ does not overlap with $\mathrm{cl}_{\mathcal{X}}(M)$, so that $p_{\theta_t}^X(x) \to \infty$ as $t \to \infty$ does *not* hold. Nonetheless, for any fixed $\varepsilon > 0$, the support of $p_{\theta_t}^X$ always overlaps with $B_\varepsilon(x)$ for large enough $t$, and thus $\sup_{x' \in B_\varepsilon(x)} p_{\theta_t}^X(x') \to \infty$ as $t \to \infty$.

many elements from $\{\mathbb{P}_\dagger^X(\partial_{\mathcal{X}} B_{\varepsilon'}(x))\}_{\varepsilon' \in (0,\varepsilon)}$ belong to an element of the partition. Thus, in particular there exists an integer $n'$ and distinct numbers $\varepsilon_1', \varepsilon_2', \ldots, \varepsilon_{n'}'$ in $(0, \varepsilon)$ such that $\mathbb{P}_\dagger^X(\partial_{\mathcal{X}} B_{\varepsilon_i'}(x)) > 1/n'$ for every $i = 1, 2, \ldots, n'$. Then, because $\partial_{\mathcal{X}} B_{\varepsilon_i'}(x) \cap \partial_{\mathcal{X}} B_{\varepsilon_j'}(x) = \emptyset$ whenever $i \neq j$, we have that

$$\mathbb{P}_\dagger^X \left( \bigcup_{i=1}^{n'} \partial_{\mathcal{X}} B_{\varepsilon_i'}(x) \right) = \sum_{i=1}^{n'} \mathbb{P}_\dagger^X \left( \partial_{\mathcal{X}} B_{\varepsilon_i'}(x) \right) > \sum_{i=1}^{n'} \frac{1}{n'} = 1, \tag{100}$$

which is clearly a contradiction since $\mathbb{P}_\dagger^X(\mathcal{X}) = 1$, thus finishing the proof. $\qquad\square$

**Lemma 3.** *Let $M \subset \mathcal{X}$, and let $\mathbb{P}_\dagger^X$ be a probability measure on $\mathcal{X}$ such that $\mathbb{P}_\dagger^X(M) = 1$. Then for every $\delta > 0$, the set $M_\delta := \{x \in \mathcal{X} \mid \inf_{x' \in M} \|x' - x\|_2 < \delta\}$ is open in $\mathcal{X}$, and is a continuity set of $\mathbb{P}_\dagger^X$.*

*Proof.* $M_\delta$ is open because it can be written as a union of open sets:

$$M_\delta = \bigcup_{x' \in M} B_\delta(x'). \tag{101}$$

Then, we have:

$$\mathbb{P}_\dagger^X (\partial_{\mathcal{X}} M_\delta) = \mathbb{P}_\dagger^X (\partial_{\mathcal{X}} M_\delta \cap M) = \mathbb{P}_\dagger^X ((\mathrm{cl}_{\mathcal{X}}(M_\delta) \setminus \mathrm{int}_{\mathcal{X}}(M_\delta)) \cap M) = \mathbb{P}_\dagger^X ((\mathrm{cl}_{\mathcal{X}}(M_\delta) \setminus M_\delta) \cap M) \tag{102}$$
$$= \mathbb{P}_\dagger^X(\emptyset) = 0, \tag{103}$$

where $\mathrm{int}_{\mathcal{X}}(M)$ denotes the topological interior of $M$ in $\mathcal{X}$, and the first equality follows from $\mathbb{P}_\dagger^X(M) = 1$, the second one from the definition of boundary, the third one from $M_\delta$ being open, and the fourth one from $M \subset M_\delta$. Thus $M_\delta$ is indeed a continuity set of $\mathbb{P}_\dagger^X$. $\qquad\square$

**Lemma 4.** *Let $M \subset \mathcal{X}$, and let $\mathbb{P}_\dagger^X$ be a probability measure on $\mathcal{X}$ such that $\mathrm{supp}(\mathbb{P}_\dagger^X) = \mathrm{cl}_{\mathcal{X}}(M)$. Then $\mathbb{P}_\dagger^X(B_\varepsilon(x)) > 0$ for every $x \in \mathrm{cl}_{\mathcal{X}}(M)$ and every $\varepsilon > 0$, where $B_\varepsilon(x) := \{x' \in \mathcal{X} \mid \|x' - x\|_2 < \varepsilon\}$.*

*Proof.* Since $\mathbb{P}_\dagger^X$ is a Borel measure and $\mathcal{X}$ is separable, $\mathbb{P}_\dagger^X(\mathrm{supp}(\mathbb{P}_\dagger^X)) = 1$ (Bogachev, 2007, Proposition 7.2.9).[30] It follows that $\mathrm{cl}_{\mathcal{X}}(M) = \mathrm{supp}(\mathbb{P}_\dagger^X)$ is nonempty. Let $x \in \mathrm{cl}_{\mathcal{X}}(M)$ and $\varepsilon > 0$. We proceed by contradiction, and assume that $\mathbb{P}_\dagger^X(B_\varepsilon(x)) = 0$. Since $\mathbb{P}_\dagger^X(\mathrm{cl}_{\mathcal{X}}(M)) = 1$, we have that

---

[30]Note that in general, $\mathbb{P}(\mathrm{supp}(\mathbb{P})) = 1$ need not hold, see for example Schilling & Kühn (2021, Examples 6.2 and 6.3).

$\mathbb{P}_{\dagger}^{X}(\operatorname{cl}_{\mathcal{X}}(M) \setminus B_{\varepsilon}(x)) = 1$. Since $B_{\varepsilon}(x)$ is open, $\operatorname{cl}_{\mathcal{X}}(M) \setminus B_{\varepsilon}(x)$ is closed. Then, by definition of support (Equation 1) and because $\operatorname{supp}(\mathbb{P}_{\dagger}^{X}) = \operatorname{cl}_{\mathcal{X}}(M)$, it follows that $\operatorname{cl}_{\mathcal{X}}(M) \cap B_{\varepsilon}(x) = \emptyset$. This is clearly a contradiction since $x \in \operatorname{cl}_{\mathcal{X}}(M) \cap B_{\varepsilon}(x)$. $\qquad\square$

*Proof of Theorem 1.* We first prove that $\liminf_{t \to \infty} p_{\theta_t}^{X}(x) = 0$, $\lambda_D$-almost-surely on $\mathcal{X} \setminus \operatorname{cl}_{\mathcal{X}}(M)$. Let $x \in \mathcal{X} \setminus \operatorname{cl}_{\mathcal{X}}(M)$, and let $U_x$ be an open neighbourhood of $x$ such that $\operatorname{cl}_{\mathcal{X}}(U_x) \cap \operatorname{cl}_{\mathcal{X}}(M) = \emptyset$, which exists because $\mathcal{X}$ is regular. Clearly $\mathbb{P}_{\dagger}^{X}(\partial_{\mathcal{X}} U_x) = 0$ (i.e. $U_x$ is a continuity set of $\mathbb{P}_{\dagger}^{X}$) and $\mathbb{P}_{\dagger}^{X}(U_x) = 0$ since $\operatorname{cl}_{\mathcal{X}}(U_x) \cap \operatorname{cl}_{\mathcal{X}}(M) = \emptyset$ and $\mathbb{P}_{\dagger}^{X}(\operatorname{cl}_{\mathcal{X}}(M)) = 1$.

By the Portmanteau Lemma, $\mathbb{P}_{\theta_t}^{X}(U_x) \to \mathbb{P}_{\dagger}^{X}(U_x)$ as $t \to \infty$, and we have the following implications:

$$\lim_{t \to \infty} \mathbb{P}_{\theta_t}^{X}(U_x) = \mathbb{P}_{\dagger}^{X}(U_x) \iff \lim_{t \to \infty} \int_{U_x} p_{\theta_t}^{X}(x') \mathrm{d}\lambda_D(x') = \mathbb{P}_{\dagger}^{X}(U_x) = 0 \tag{104}$$

$$\implies \liminf_{t \to \infty} \int_{U_x} p_{\theta_t}^{X}(x') \mathrm{d}\lambda_D(x') = 0. \tag{105}$$

Then, by Fatou's lemma,

$$\int_{U_x} \liminf_{t \to \infty} p_{\theta_t}^{X}(x') \mathrm{d}\lambda_D(x') \leq 0. \tag{106}$$

Since $\liminf_{t \to \infty} p_{\theta_t}^{X}(x) \geq 0$, $\lambda_D$-almost-everywhere on $\mathcal{X}$, it follows that

$$\int_{U_x} \liminf_{t \to \infty} p_{\theta_t}^{X}(x') \mathrm{d}\lambda_D(x') = 0, \tag{107}$$

and thus $\liminf_{t \to \infty} p_{\theta_t}^{X}(x) = 0$, $\lambda_D$-almost-everywhere on $U_x$.

We still need to extend the result from $\lambda_D$-almost-everywhere on $U_x$ to $\lambda_D$-almost-everywhere on $\mathcal{X} \setminus \operatorname{cl}_{\mathcal{X}}(M)$. Clearly $\{U_x\}_{x \in \mathcal{X} \setminus \operatorname{cl}_{\mathcal{X}}(M)}$ is an open cover of $\mathcal{X} \setminus \operatorname{cl}_{\mathcal{X}}(M)$. Since $\mathcal{X}$ is second countable and every subspace of a second countable space is second countable, it follows that $\mathcal{X} \setminus \operatorname{cl}_{\mathcal{X}}(M)$ is second countable. Then, by Lindelöf's lemma, $\{U_x\}_{x \in \mathcal{X} \setminus \operatorname{cl}_{\mathcal{X}}(M)}$ has a countable subcover $\{U_{x_i}\}_{i=1}^{\infty}$ of $\mathcal{X} \setminus \operatorname{cl}_{\mathcal{X}}(M)$. The result then holds $\lambda_D$-almost-everywhere on $U_{x_i}$ for $i = 1, 2, \dots$, and because any countable union of sets of measure 0 has measure 0, it also holds $\lambda_D$-almost-everywhere on

$$\bigcup_{i=1}^{\infty} U_{x_i} = \mathcal{X} \setminus \operatorname{cl}_{\mathcal{X}}(M), \tag{108}$$

which finishes this part of the proof.

Now, let $x \in \operatorname{cl}_{\mathcal{X}}(M)$ and $\varepsilon > 0$, and we will prove that $\sup_{x' \in B_{\varepsilon}(x)} p_{\theta_t}^{X}(x') \to \infty$ as $t \to \infty$.

First, note that $\sup_{x' \in B_{\varepsilon}(x)} p_{\theta_t}^{X}(x')$ is increasing in $\varepsilon$ for every $t$. If $B_{\varepsilon}(x)$ is not a continuity set of $\mathbb{P}_{\dagger}^{X}$, by Lemma 2 we could always find $\varepsilon' \in (0, \varepsilon)$ such that $B_{\varepsilon'}(x)$ is a continuity set of $\mathbb{P}_{\dagger}^{X}$, and if we managed to prove that $\sup_{x' \in B_{\varepsilon'}(x)} p_{\theta_t}^{X}(x') \to \infty$ as $t \to \infty$, the same result would immediately follow for $\varepsilon$. We can thus assume without loss of generality that $\varepsilon$ is such that $B_{\varepsilon}(x)$ is a continuity set of $\mathbb{P}_{\dagger}^{X}$.

Now, let $M_{\delta} := \{x \in \mathcal{X} \mid \inf_{x' \in M} \|x' - x\|_2 < \delta\}$ and let $U_{\varepsilon,\delta}(x) := M_{\delta} \cap B_{\varepsilon}(x)$. From basic topology we have that $\partial_{\mathcal{X}}(M_{\delta} \cap B_{\varepsilon}(x)) \subset \partial_{\mathcal{X}} M_{\delta} \cup \partial_{\mathcal{X}} B_{\varepsilon}(x)$. Since $M_{\delta}$ and $B_{\varepsilon}(x)$ are continuity sets of $\mathbb{P}_{\dagger}^{X}$ by Lemma 3 and by assumption, respectively, it follows that $U_{\varepsilon,\delta}(x)$ is a continuity set of $\mathbb{P}_{\dagger}^{X}$, since $\mathbb{P}_{\dagger}^{X}(\partial_{\mathcal{X}} U_{\varepsilon,\delta}(x)) \leq \mathbb{P}_{\dagger}^{X}(\partial_{\mathcal{X}} M_{\delta}) + \mathbb{P}_{\dagger}^{X}(\partial_{\mathcal{X}} B_{\varepsilon}(x)) = 0$. Similarly, $B_{\varepsilon}(x) \setminus M_{\delta}$ is a continuity set of $\mathbb{P}_{\dagger}^{X}$ because $\partial_{\mathcal{X}}(B_{\varepsilon}(x) \setminus M_{\delta}) \subset \partial_{\mathcal{X}} B_{\varepsilon}(x) \cup \partial_{\mathcal{X}} M_{\delta}$. We then write:

$$\mathbb{P}_{\theta_t}^{X}(B_{\varepsilon}(x)) = \mathbb{P}_{\theta_t}^{X}(U_{\varepsilon,\delta}(x)) + \mathbb{P}_{\theta_t}^{X}(B_{\varepsilon}(x) \setminus M_{\delta}) = \int_{U_{\varepsilon,\delta}(x)} p_{\theta_t}^{X}(x') \mathrm{d}\lambda_D(x') + \mathbb{P}_{\theta_t}^{X}(B_{\varepsilon}(x) \setminus M_{\delta}) \tag{109}$$

$$\leq \lambda_D(U_{\varepsilon,\delta}(x)) \sup_{x' \in U_{\varepsilon,\delta}(x)} p_{\theta_t}^{X}(x') + \mathbb{P}_{\theta_t}^{X}(B_{\varepsilon}(x) \setminus M_{\delta}) \tag{110}$$

$$\leq \lambda_D(U_{\varepsilon,\delta}(x)) \sup_{x' \in B_{\varepsilon}(x)} p_{\theta_t}^{X}(x') + \mathbb{P}_{\theta_t}^{X}(B_{\varepsilon}(x) \setminus M_{\delta}). \tag{111}$$

Since $B_\varepsilon(x)$ is open, as is $M_\delta$ by Lemma 3, then $U_{\varepsilon,\delta}(x)$ is open as well. In turn $\lambda_D(U_{\varepsilon,\delta}(x)) > 0$, and it follows that

$$\frac{\mathbb{P}^X_{\theta_t}(B_\varepsilon(x)) - \mathbb{P}^X_{\theta_t}(B_\varepsilon(x) \setminus M_\delta)}{\lambda_D(U_{\varepsilon,\delta}(x))} \leq \sup_{x' \in B_\varepsilon(x)} p^X_{\theta_t}(x'). \tag{112}$$

By the Portmanteau Lemma and since $\mathbb{P}^X_\dagger(B_\varepsilon(x) \setminus M_\delta) = 0$, taking the limit as $t \to \infty$ of the left hand side of the above equation yields

$$\lim_{t \to \infty} \frac{\mathbb{P}^X_{\theta_t}(B_\varepsilon(x)) - \mathbb{P}^X_{\theta_t}(B_\varepsilon(x) \setminus M_\delta)}{\lambda_D(U_{\varepsilon,\delta}(x))} = \frac{\mathbb{P}^X_\dagger(B_\varepsilon(x))}{\lambda_D(U_{\varepsilon,\delta}(x))}. \tag{113}$$

Thus, taking $\liminf$ as $t \to \infty$ on both sides of Equation 112 implies that

$$\frac{\mathbb{P}^X_\dagger(B_\varepsilon(x))}{\lambda_D(U_{\varepsilon,\delta}(x))} \leq \liminf_{t \to \infty} \sup_{x' \in B_\varepsilon(x)} p^X_{\theta_t}(x'). \tag{114}$$

Since $U_{\varepsilon,\delta'}(x) \subset U_{\varepsilon,\delta}(x)$ whenever $\delta' < \delta$, we have that

$$\lim_{\delta \to 0^+} \lambda_D(U_{\varepsilon,\delta}(x)) = \lambda_D\left(\bigcap_{\delta > 0} U_{\varepsilon,\delta}(x)\right) = \lambda_D\left(\bigcap_{\delta > 0}(M_\delta \cap B_\varepsilon(x))\right) = \lambda_D\left(\left(\bigcap_{\delta > 0} M_\delta\right) \cap B_\varepsilon(x)\right) \tag{115}$$

$$= \lambda_D(\mathrm{cl}_{\mathcal{X}}(M) \cap B_\varepsilon(x)) = 0, \tag{116}$$

where we used that $(i)$ $\bigcap_{\delta > 0} M_\delta = \mathrm{cl}_{\mathcal{X}}(M)$, which holds because $\mathrm{cl}_{\mathcal{X}}(M)$ is the set of points which are arbitrarily close to $M$, and that $(ii)$ $\lambda_D(\mathrm{cl}_{\mathcal{X}}(M)) = 0$ by assumption. Finally, by Lemma 4, $\mathbb{P}^X_\dagger(B_\varepsilon(x)) > 0$, so that taking the limit as $\delta \to 0^+$ on both sides of Equation 114 yields that $\liminf_{t \to \infty} \sup_{x' \in B_\varepsilon(x)} p^X_{\theta_t}(x') = \infty$, which in turn implies that $\sup_{x' \in B_\varepsilon(x)} p^X_{\theta_t}(x') \to \infty$ as $t \to \infty$, finishing the proof. $\quad\square$

## B.2 Formalizing the Link between Two-Step Models and Optimal Transport

For convenience, we restate Proposition 1 below.

**Proposition 1.** *Let $\mathbb{P}^X_*$ be a probability measure on $\mathcal{X}$, $g_{\theta_1} : \mathcal{Z} \to \mathcal{X}$ be measurable, and $c : \mathcal{X} \times \mathcal{X} \to \mathbb{R}$ be measurable and such that there exists $C > 0$ such that*

$$\sup_{(x,y) \in \mathcal{X} \times \mathcal{X}} |c(x,y)| < C. \tag{93}$$

*Then,*

$$\inf_{f \in \mathcal{F}} \mathbb{E}_{X \sim \mathbb{P}^X_*}[c(X, g_{\theta_1}(f(X)))] = \inf_{f \in \mathcal{C}} \mathbb{E}_{X \sim \mathbb{P}^X_*}[c(X, g_{\theta_1}(f(X)))], \tag{94}$$

*where $\mathcal{F} := \{f : \mathcal{X} \to \mathcal{Z} \mid f \text{ is measurable}\}$ and $\mathcal{C} := \{f : \mathcal{X} \to \mathcal{Z} \mid f \text{ is continuous}\}$.*

*Proof.* Since $\mathcal{C} \subset \mathcal{F}$, it follows that

$$\inf_{f \in \mathcal{F}} \mathbb{E}_{X \sim \mathbb{P}^X_*}[c(X, g_{\theta_1}(f(X)))] \leq \inf_{f \in \mathcal{C}} \mathbb{E}_{X \sim \mathbb{P}^X_*}[c(X, g_{\theta_1}(f(X)))]. \tag{117}$$

Since both infimums are finite due to the assumption from Equation 93, it is enough to show that, for every $\varepsilon > 0$, there exists $f_\varepsilon \in \mathcal{C}$ such that

$$\inf_{f \in \mathcal{F}} \mathbb{E}_{X \sim \mathbb{P}^X_*}[c(X, g_{\theta_1}(f(X)))] > \mathbb{E}_{X \sim \mathbb{P}^X_*}[c(X, g_{\theta_1}(f_\varepsilon(X)))] - \varepsilon. \tag{118}$$

Let $\varepsilon > 0$, and let $f_* \in \mathcal{F}$ be such that

$$\inf_{f \in \mathcal{F}} \mathbb{E}_{X \sim \mathbb{P}^X_*}[c(X, g_{\theta_1}(f(X)))] > \mathbb{E}_{X \sim \mathbb{P}^X_*}[c(X, g_{\theta_1}(f_*(X)))] - \frac{\varepsilon}{2}. \tag{119}$$

It is thus enough to show that there exists $f_\varepsilon \in \mathcal{C}$ such that

$$\mathbb{E}_{X \sim \mathbb{P}_*^X} \left[ c\left( X, g_{\theta_1}\left( f_*(X) \right) \right) \right] > \mathbb{E}_{X \sim \mathbb{P}_*^X} \left[ c\left( X, g_{\theta_1}\left( f_\varepsilon(X) \right) \right) \right] - \frac{\varepsilon}{2}, \tag{120}$$

as this would imply Equation 118 holds. Since $\mathbb{P}_*^X$ is a Borel measure and $\mathcal{X}$ is Polish, $\mathbb{P}_*^X$ is a Radon measure. Additionally, $\mathbb{P}_*^X(\mathcal{X}) < \infty$, $\mathcal{X}$ is locally compact, $\mathcal{Z}$ is second countable, and $f_*$ is measurable; so it follows by Lusin's theorem that there exists a Borel set $E \subset \mathcal{X}$ and a continuous function $f_\varepsilon : \mathcal{X} \to \mathcal{Z}$ such that $f_\varepsilon(x) = f_*(x)$ for every $x \in E$, and $\mathbb{P}_*^X(\mathcal{X} \setminus E) < \varepsilon/(4C)$. Then, we have:

$$\mathbb{E}_{X \sim \mathbb{P}_*^X} \left[ c\left( X, g_{\theta_1}\left( f_\varepsilon(X) \right) \right) \right] - \mathbb{E}_{X \sim \mathbb{P}_*^X} \left[ c\left( X, g_{\theta_1}\left( f_*(X) \right) \right) \right] \tag{121}$$

$$= \int_{\mathcal{X}} c\left( x, g_{\theta_1}\left( f_\varepsilon(x) \right) \right) - c\left( x, g_{\theta_1}\left( f_*(x) \right) \right) \mathrm{d}\mathbb{P}_*^X(x) = \int_{\mathcal{X} \setminus E} c\left( x, g_{\theta_1}\left( f_\varepsilon(x) \right) \right) - c\left( x, g_{\theta_1}\left( f_*(x) \right) \right) \mathrm{d}\mathbb{P}_*^X(x) \tag{122}$$

$$\leq \int_{\mathcal{X} \setminus E} \left| c\left( x, g_{\theta_1}\left( f_\varepsilon(x) \right) \right) - c\left( x, g_{\theta_1}\left( f_*(x) \right) \right) \right| \mathrm{d}\mathbb{P}_*^X(x) \leq \int_{\mathcal{X} \setminus E} 2C \, \mathrm{d}\mathbb{P}_*^X(x) = 2C \, \mathbb{P}_*^X(\mathcal{X} \setminus E) < \frac{\varepsilon}{2}, \tag{123}$$

which in turn implies Equation 120 holds, thus finishing the proof. $\qquad\square$

## C   Experimental Details

Here we provide details on the experiments from Section 5.3.2. Our code is available at `https://github.com/layer6ai-labs/dgm_manifold_survey`.

**First-step objective for latent diffusion models**   Following Rombach et al. (2022), we trained latent diffusion models by first using a regularized variational autoencoder (Section 4.1.2) loss (Larsen et al., 2016; Higgins et al., 2017):

$$\min_{\theta_1, \phi} \max_{\phi'} \mathbb{E}_{X \sim p_*^X} \left[ \mathbb{E}_{Z \sim q_\phi^{Z|X}(\cdot|X)} \left[ \| X - g_{\theta_1}(Z) \|_2^2 \right] \right] + \beta_1 \mathbb{KL} \left( q_\phi^{Z|X}(\cdot|X) \,\|\, p^Z \right)$$
$$+ \beta_2 \mathbb{E}_{X \sim p_*^X} \left[ \mathbb{E}_{Z \sim q_\phi^{Z|X}(\cdot|X)} \left[ \log h_{\phi'}(X) + \log\left(1 - h_{\phi'}(g_{\theta_1}(Z))\right) \right] \right], \tag{124}$$

where $\beta_1 > 0$ and $\beta_2 > 0$ are hyperparameters; $p^Z$ is a standard Gaussian; $q_\phi^{Z|X}(\cdot|x) = \mathcal{N}(\cdot\,; f_\phi(x), \Sigma_\phi^{Z|X}(x))$ with $\Sigma_\phi^{Z|X}(x)$ being diagonal for every $x \in \mathcal{X}$; and $h_{\phi'} : \mathcal{X} \to (0,1)$ is a binary classifier inspired by generative adversarial networks (Section 4.2), whose objective is to distinguish between real samples $X \sim p_*^X$ and their stochastic reconstructions $g_{\theta_1}(Z)$, where $Z \sim q_\phi^{Z|X}(\cdot|X)$. Rather than fixing $\beta_2$, we dynamically update it throughout training to ensure that the first and third terms in Equation 124 have roughly the same magnitude; when taking a gradient step, this is achieved by computing the ratio of the values of the first to third term in the previous gradient step, and setting $\beta_2$ to the absolute value of this ratio.

**Training objective for diffusion models**   We train all diffusion models, latent or not, exactly as described in Section 5.1.2, i.e. through Equation 55. The integral with respect to $t$ is approximated by sampling $t$ uniformly at random in $[0, T]$ during training (one such $t$ is sampled for every element in the batch).

**Hyperparameters**   We use the Adam optimizer (Kingma & Ba, 2015) with a batch size of 128 throughout, and train all models until there is no improvement on the validation metric for 50 epochs; we keep the models with the best validation performance. For the VAE of latent diffusion models we use the squared reconstruction error $\mathbb{E}_{X \sim p_*^X}[\| X - g_{\theta_1}(f_\phi(X)) \|_2^2]$ rather than Equation 124 as the validation metric, $\beta_1 = 10^{-6}$, a learning rate of $10^{-4}$ with cosine annealing, and take two gradient steps on $\phi'$ for every gradient step on $(\theta_1, \phi)$. For the diffusion models, both on ambient and latent space, we use Equation 55 as the validation metric, a learning rate of $5 \times 10^{-5}$ without cosine annealing, $T = 1$, $\beta_{\min} = 0.1$, $\beta_{\max} = 20$, $w(t) = \sigma_t^2$, and an Euler-Maruyama discretization scheme with 1000 steps to generate the paths in Figure 8.

Table 1: Configuration used for neural networks. The architecture of the auxiliary network $h_{\phi'}$ for the VAE mimics that of the encoder, and the decoder $g_{\theta_1}$ is given by simply "reversing" the architecture of the encoder. See text for additional details.

| PARAMETER | AMBIENT SCORE $\hat{s}_\theta^X$ | VAE ENCODER $q_\phi^{Z|X}$ | LATENT SCORE $\hat{s}_{\theta_2}^Z$ |
|---|---|---|---|
| n_channels | 64 | 64 | 256 |
| ch_mults | $(1, 2, 2, 4)$ | $(1, 2, 2)$ | $(1, 2)$ |
| is_attn | $(\mathrm{False}, \mathrm{False}, \mathrm{True}, \mathrm{True})$ | - | $(\mathrm{True}, \mathrm{True})$ |
| n_blocks | 2 | 2 | 2 |

**Architectures** For a fair comparison between diffusion models on ambient and latent space, we attempt to instantiate them in such a way that their overall architectures are as similar as possible. The configurations of the architectures we used are given in Table 1, which we now describe. We parameterize both the ambient and latent score networks as the output of a neural network divided by $\sigma_t$, as mentioned in Section 5.1.2 (this is equivalent to the so-called "$\varepsilon$ parameterization" of Ho et al. (2020) with $-\varepsilon$ being parameterized instead of $\varepsilon$). For the diffusion model on ambient space, we use a U-Net architecture (Ronneberger et al., 2015) with residual connections (He et al., 2016) and an additional attention mechanism (Vaswani et al., 2017), as implemented in the labml package (Jayasiri & Wijerathne, 2020), which uses a sinusoidal positional embedding for the scalar input $t$. The ambient space U-Net takes $3 \times 32 \times 32$ images, and progressively downsamples them to a shape of $1024 \times 4 \times 4$ before upscaling them back to their original size. For the latent diffusion model, we attempt to copy the aforementioned U-Net as much as possible; the VAE mimics the U-Net up until the $8 \times 8$ resolution, and adds a convolutional layer to produce outputs of shape $4 \times 8 \times 8$, so that $d = 256$. To ensure the VAE obtains low-dimensional representations through a proper bottleneck, we remove the U-Net skip connections between the encoder and decoder, resulting in a purely residual architecture. The stochastic encoder $q_\phi^{Z|X}$ of the VAE consists of a single neural network which takes $x \in \mathcal{X}$ and produces a $2d$-dimensional output – the first $d$ dimensions correspond to $f_\phi(x)$, and the remaining ones to the diagonal of $\Sigma_\phi^{Z|X}(x)$. The auxiliary network $h_{\phi'}$ has the same architecture as the encoder $f_\phi$, except a final linear layer is added to ensure the output is a scalar. The score network of the diffusion on latent space is given another U-Net which further downsamples to a $4 \times 4$ resolution before upsampling.

**Data preprocessing** Recall that raw image data is integer-valued, with possible values ranging from 0 to 255. We dequantize the data before training the diffusion model on ambient space and the VAE for the latent diffusion model, i.e. we add independent uniform $[0, 1]$ noise to every pixel (Theis et al., 2016), so that the resulting data now has entries in $[0, 256]$. We then linearly scale the data so that every coordinate lies in $[-1, 1]$. For latent diffusion models, once the VAE is trained, we also scale its encodings to lie in $[-1, 1]^d$ before training the diffusion model on latent space.

