# OpenReview forum: "Deep Generative Models through the Lens of the Manifold Hypothesis: A Survey and New Connections"
_TMLR — Accepted by TMLR_

### Review · Reviewer_uivD · 2024-05-12

**Summary Of Contributions:**

This paper surveys deep generative models with insights from the manifold hypothesis. Specifically, the authors describe how the existing generative model works and explain why, based on the mathematical intuitions from the manifold hypothesis, those generative models would be desirable/undesirable. The authors also present some future research directions about deep generative models and their potential interplay with the manifold hypothesis.

The paper starts by introducing mathematical notations, formal setup, and preliminary materials. The preliminary materials describe the common learning objectives used for training deep generative models: KL divergence, Wasserstein distances, and maximum mean discrepancy (MMD). Particularly, it is pointed out that minimizing KL divergence is likely to fail to maximize the likelihood correctly if we consider the manifold hypothesis.

The next two sections (Section 4 and Section 5) cover deep generative models for continuous distributions. The authors divide such models into two categories based on their manifold awareness. Section 4 focuses on the manifold-unaware variants of deep generative models. The authors introduce their own results about why the likelihood-based approaches might fail in those deep generative models. Then, the authors go through the examples of existing generative models, e.g., variational autoencoders, normalizing flows, energy-based models, generative adversarial networks, and score matching, explaining how the authors’ results may affect those models. On the contrary, Section 5 concentrates on the manifold aware deep generative models. Specifically, the authors classify various methods by how each method achieves the manifold-awareness: adding noise, support-agnostic optimization objectives, and two-step aproaches that model the latent space. In each subsection, the authors introduce the deep generative models that use each approach to enable the manifold-awareness and explain them through the lens of manifold hypothesis. In particular, Section 5.3 introduces another result that helps us understand what the two-step models do; they minimize the upper bound of the Wasserstein distance under a specific assumption.

Section 6  briefly describes deep generative models for discrete distribution cases. In these cases, there is no dimensionality mismatch (between the dimension of the underlying data manifold and the dimension of the manifold that the model generates), so the models will not be affected by the problems mentioned in the previous sections.

Finally, the authors summarize their contributions and introduce future research directions with a focus on the manifold hypothesis.

**Audience:**

Yes

**Broader Impact Concerns:**

I don’t see a particular broader impact concern regarding this paper.

**Claims And Evidence:**

Yes

**Requested Changes:**

I don’t see a specific change that is required for this paper.

**Strengths And Weaknesses:**

### Strength
1. The paper is well-organized, covers various deep generative models, and provides insights into how generative models relate to manifold hypotheses.
2. Not only that the authors summarize existing works, but they also present their own contributions: one regarding the instability of likelihood-based training of deep generative models and another regarding two-step models. I don’t think that a review paper necessarily has such contributions, but the results provide more insights into understanding the existing deep generative models’ performance.
3. The authors considered readers from diverse backgrounds by separating advanced topics into boxes. As the advanced topics are separated, this paper becomes more accessible to people without a strong mathematical background, and those people who are familiar with mathematical details can always delve into the advanced topics in separate boxes.

### Weaknesses
1. This is a well-written paper, and I cannot find a specific weakness to criticize.

---

> ### Author Response · Authors · 2024-07-04
> **Rebuttal**
>
> Thank you very much for your review; we sincerely appreciate the time and effort that went into it, and are delighted that you “cannot find a specific weakness to criticize” in our paper. We have updated the manuscript to address points raised by other reviewers and incorporate a few additional citations; please let us know if you have any additional feedback.

---

### Review · Reviewer_Ydqc · 2024-05-25

**Summary Of Contributions:**

This is mostly a review paper on generative models, with the focus on understanding the consequence of the manifold hypothesis on various deep generative models (DGMs). It provides an overview of methods that are compatible with the manifold hypothesis (manifold aware) and those that are not (manifold unaware). Along the way, the authors provide some new perspectives on these models, including a rigorous understanding on why DGMs based on maximum likelihood fail under the manifold hypothesis (Theorem 1) and an understanding of two-step models (i.e., models that treat the manifold learning and distribution learning steps separately) in terms of minimising an upper bound on the Wasserstein distance between the true and approximate distributions.

**Audience:**

Yes

**Broader Impact Concerns:**

This is mostly a review article and as far as I can see, there are no ethical implications of the work.

**Claims And Evidence:**

Yes

**Requested Changes:**

- Can you please define $\mathtt{stopgrad}$ in page 19 (also in page 44)?
- In Equation (84), the $\theta$ that is minimised should be $\theta_1$. It would also be helpful to explicitly state somewhere that $q_\phi^Z$ is the pushforward measure $q_\phi^Z = f_\sharp p_*^X$.

**Strengths And Weaknesses:**

**Strengths:**

The paper is well-organised and the various DGMs considered are presented with full clarity, with appropriate references. The arguments made for and against each model to learn distributions on a lower dimensional manifold are mostly sound, presented at both an intuitive level and also at a more rigorous level in grey boxes. Indeed, the capability of generative models to learn distributions under the manifold hypothesis helps to explain the success of certain models over others and is thus important to understand the interplay between the two.
Overall, I can see the work being a useful reference for an overview of DGMs and how "aware" they are about the manifold hypothesis. The insights that the paper provide could be useful for designing new generative model architecture in the future.

**Weaknesses:**

While most of the arguments presented in the paper were satisfactory, there were a few parts that were unclear to me and would be improved with some additional explanations.
- The reasoning provided for why score matching is not manifold-aware is not fully satisfactory compared to explanations provided for other models. When the authors state "we should intuitively expect score matching to fail ... due to $p_*^X$ not being a full-dimensional density", could you please explain why that should be the case?
- The explanation for the exploding gradients in diffusion models under the manifold hypothesis was also confusing to me (pages 25-26). As the samples $Y_t$ approach the data manifold, shouldn't we expect the gradient to decrease with the time step? I am just not sure if the argument made here makes sense unless we are taking adaptive time steps that decrease as $t \rightarrow T$.
- I am not sure if I understand the logic behind the statement "if CFM successfully learns its target distribution in the manifold setting, then it must experience the numerical instabilities of likelihood estimation" (page 28).
- I don't immediately see how the loss (68) is derived from (67).

---

> ### Author Response · Authors · 2024-07-04
> **Rebuttal**
>
> Thank you very much for your review; we sincerely appreciate the time and effort that went into it, and are happy that you found our work useful both as a reference of manifold-awareness of various generative models, as well as for “designing new generative model architectures in the future”. Below we address the points you raised. Please see the updated manuscript for the corresponding changes (in blue):
>
> > The reasoning provided for why score matching is not manifold-aware is not fully satisfactory compared to explanations provided for other models. When the authors state "we should intuitively expect score matching to fail ... due to $p_*^X$ not being a full-dimensional density", could you please explain why that should be the case?
>
>
> - Thank you for raising this point; we agree that the explanation in our initial submission was not fully fleshed-out. The point here is that the model is full-dimensional, whereas the ground truth is manifold supported. The densities are thus fundamentally different types of objects, and so are the corresponding score functions. Thus, Eq 43 should not be expected to be a meaningful objective in this setting, since matching the score functions is just as meaningless as attempting to match dimensionally-mismatched densities.
>
> > The explanation for the exploding gradients in diffusion models under the manifold hypothesis was also confusing to me (pages 25-26). As the samples $Y_t$ approach the data manifold, shouldn't we expect the gradient to decrease with the time step? I am just not sure if the argument made here makes sense unless we are taking adaptive time steps that decrease as $t \rightarrow T$.
>
>
> - Thank you as well for raising this point, we also agree that our explanation on this point can be improved. In short, the result of Pidstrigach (2022) and Lu et al. (2023) that we discuss shows that, for a fixed point outside the data manifold, the score function blows up. This does not immediately imply that the sampled trajectories explode as well (as has been observed in practice), since the trajectories converge to a point on the manifold as a consequence of diffusion models being manifold-aware. The blow up of the score function of course still shows why the score function is unbounded, which is a necessary condition for the trajectories to blow up in the first place. We have now a detailed discussion about this in the paper (pg 27).
>
> > I am not sure if I understand the logic behind the statement "if CFM successfully learns its target distribution in the manifold setting, then it must experience the numerical instabilities of likelihood estimation" (page 28).
>
>
> - Note that we are not claiming that the numerical instabilities appear during training, just for likelihood evaluation. This is because of the first bullet point in section 4.1.1, “Numerical instability of likelihood evaluation”.  We have clarified this distinction in the updated manuscript.
>
> > I don't immediately see how the loss (68) is derived from (67).
>
>
> - We had included an explanation as to why this is the case, “ The first term in…”, but we have moved this explanation closer to the equation for added clarity. In short, one term corresponds to the transport cost without concern for the constraint, and the other terms aim to enforce the constraint. Please note that in the updated manuscript, these are now eqs 69 and 70, rather than 67 and 68.
>
> > Can you please define $\mathtt{stopgrad}$ in page 19 (also in page 44)?
>
>
> - Thank you for identifying this missing definition, which we have now added to the manuscript.
>
> > In Equation (84), the $\theta$ that is minimised should be $\theta_1$. It would also be helpful to explicitly state somewhere that $q_\phi^Z$ is the pushforward measure $q_\phi^Z = f_\sharp p_*^X$.
>
>
> - Note that there is no typo in eq 88 (formerly eq 84): the optimization problem is over $\theta=(\theta_1, \theta_2)$ even though $\theta_2$ appears only in the constraint. In the updated manuscript we have also explicitly stated that $q_\phi^Z$ is the pushforward of $p_*^X$ through $f_\phi$.
>
> We have updated our manuscript to address the points above, as well as points raised by other reviewers, and to include some additional citations. Please let us know if you have any additional feedback.

---

### Review · Reviewer_txjV · 2024-06-24

**Summary Of Contributions:**

This manuscript provides a survey of the modern deep generative models from the perspective of the manifold hypothesis. Namely, the paper assumes that the target data distribution lies on a low-dimensional manifold embedded in a high-dimensional ambient space. Furthermore, this low-dimensional manifold is unknown analytically (although, the paper quickly touches the generative models where the data-manifold is known analytically, e.g. data on a sphere) but given implicitly, i.e. can be recovered from the empirical distribution of the data.

Under the manifold perspective, the paper divides all the existing generative models into two categories: the ones that operate under the assumption that the dimensionality of the data manifold matches the dimensionality of the ambient space (manifold-unaware) and the ones that assume that the dimensionality of the manifold is much smaller than of the ambient space (manifold-aware). The paper goes rigorously over the major existing approaches (to the best of my knowledge over all practical approaches), classifies them according to the proposed dichotomy, and pinpoints potential issues.

For the manifold-unaware models, the authors consider the conventional approaches to generative modeling (Variational Autoencoders, Normalizing Flows, Energy-based models, Generative Adversarial Networks (GANs), Score Matching) and demonstrate the potential issues related to the manifold hypothesis. Moreover, the authors present one of their results - the Likelihood Instability theorem, which builds on top of the result from [1]. Intuitively, this theorem shows that, for the models with full-dimensional support, the maximization of the likelihood is numerically unstable and is incapable of recovering the ground true measure. The discussion of other models (Score Matching and GANs) is more intuitive and empirical.

For the manifold-aware models, the authors outline three major approaches to overcoming issues related to the manifold hypothesis: adding noise to the data or model outputs, using objectives that are properly defined for the manifold-supported data, learning the models by the two-step procedure (learning the manifold at the first step and then learning a generative model under the assumption of the full-dimensional support). Besides careful consideration and analysis of different design choices, the authors propose a novel result (based on the results from [2]) stating that the two-step learning procedure with the learning of an autoencoder at the first stage and a generative model in the latent space minimizes an upper bound on the Wasserstein distance.

Finally, the authors conclude with a short discussion of how discrete-space generative models relate to the manifold hypothesis and an outline of major challenges left in the field.

[1] Gabriel Loaiza-Ganem, Brendan Leigh Ross, Jesse C Cresswell, and Anthony L Caterini. Diagnosing and fixing manifold overfitting in deep generative models. Transactions on Machine Learning Research, 2022a.
[2] Ilya Tolstikhin, Olivier Bousquet, Sylvain Gelly, and Bernhard Schölkopf. Wasserstein auto-encoders. In International Conference on Learning Representations, 2018.

**Audience:**

Yes

**Broader Impact Concerns:**

This paper does not raise any broader impact concerns.

**Claims And Evidence:**

Yes

**Requested Changes:**

I would suggest to rework the parts of the paper that raised my concerns. However, it is not strictly necessary for the acceptance of the manuscript.

**Strengths And Weaknesses:**

The paper does not raise any major concerns. It is well-written, the presented results are technically sound, and the literature review is extensive. The paper has most of its value as a survey encompassing numerous generative models from the perspective of the manifold supported data (which is a ubiquitous setting in machine learning). Furthermore, part of its value comes from the novel results regarding the likelihood-based models (which remain one of the main interests of the community) and the two-step learning procedure (which is the state-of-the-art approach to generative modeling). The paper is of great interest to a major part of the community: novice researchers would find it valuable as a work encompassing main approaches while senior members of the community would benefit from a coherent picture of one of the main problems in the field.

The list of minor concerns:
- Some parts of the paper read as a "laundry list". However, this might be unavoidable for a survey paper.
- Throughout the paper, the authors refer to numerical instabilities but do not give any (formal or informal) definition. Especially, this worth clarifying in the context of VAEs and EBMs whose training indeed has issues but is not usually described as numerical instabilities.
- Page 1. The wording "high-dimensional likelihoods" might be confusing for the reader.
- Page 2. I suggest the authors avoid phrases that are excessively emotional "one would have to wait forever" and technically imprecise "any machine learning researcher...".
- Page 15. The unfixability of maximum likelihood requires a more detailed explanation. Namely, the claim that no such regularizer exists that can fix the issues of the maximum likelihood is either too vague or too strong. Indeed, couldn't I just take a regularizer that changes the maximization objective, i.e. cancels out the likelihood and substracts the Wasserstein distance?
- Page 22. Denoising Score Matching could be considered as a part of Score-Based Diffusion models since it is usually not used independently.
- Page 32. "WAEs only encourage this to happen on average" might be confusing when speaking about measures.
- Page 42. The noised-out data density lacks parameter $\sigma$ in its definition.

---

> ### Author Response · Authors · 2024-07-04
> **Rebuttal**
>
> Thank you very much for your review; we sincerely appreciate the time and effort that went into it, and are happy that you found our work “well-written”, “technically sound”, “extensive”, and that it is of “great interest to a major part of the community”. Below we address the points you raised. Please see the updated manuscript for the corresponding changes (in blue):
>
> > Some parts of the paper read as a "laundry list". However, this might be unavoidable for a survey paper.
>
>
>  - While we believe that our paper is cohesive thanks to its presented taxonomy of generative models, we agree that the “laundry list” presentation is to a certain extent unavoidable for a survey paper.
>
> > Throughout the paper, the authors refer to numerical instabilities but do not give any (formal or informal) definition. Especially, this worth clarifying in the context of VAEs and EBMs whose training indeed has issues but is not usually described as numerical instabilities.
>
>
> - Thank you very much for pointing this out; we agree that making the link between theory and practice more explicit was lacking from our original submission. We have modified our manuscript accordingly - specifically see Section 4.1.2 (pg 17) for VAEs and Section 4.1.4 (pg 20) for EBMs. We do however point out that Theorem 1 (pg 15) does provide a mathematically precise meaning to the instabilities we refer to.
>
> > The wording "high-dimensional likelihoods" might be confusing for the reader.
>
> - We have now rephrased this sentence in the manuscript.
>
> > I suggest the authors avoid phrases that are excessively emotional "one would have to wait forever" and technically imprecise "any machine learning researcher...".
>
> - We have now also rephrased these sentences in the manuscript.
>
> > The unfixability of maximum likelihood requires a more detailed explanation. Namely, the claim that no such regularizer exists that can fix the issues of the maximum likelihood is either too vague or too strong. Indeed, couldn't I just take a regularizer that changes the maximization objective, i.e. cancels out the likelihood and substracts the Wasserstein distance?
>
> - We agree that adding such a term would indeed “fix” the problem during training (but not for likelihood evaluation, see the “Numerical instability of likelihood evaluation” paragraph in section 4.1.1), but we believe that calling such a term a regularizer would be a stretch, as it would be a complete change of objective rather than an actual regularizer. Nonetheless, we have now specified in the manuscript that the discussion refers to regularizers which do not obviate the need to compute likelihoods (or its surrogates) during training.
>
> > Denoising Score Matching could be considered as a part of Score-Based Diffusion models since it is usually not used independently.
>
> - While we agree that these are indeed usually not used independently, we believe our current organization is preferable to combining these sections since there are mathematical differences between them.
>
> > "WAEs only encourage this to happen on average" might be confusing when speaking about measures.
>
> - Please see the updated manuscript, where we believe the statement is now clearer.
>
> > The noised-out data density lacks parameter $\sigma$ in its definition.
>
> - Thank you for catching this, we have fixed it in the updated manuscript.
>
> We have updated our manuscript to address the points above, as well as points raised by other reviewers, and to include some additional citations. Please let us know if you have any additional feedback.

---

### Author Response · Authors · 2024-07-04
**Rebuttal**

We thank all reviewers for their careful and detailed reviews. We have updated the manuscript to address all the points raised in the reviews. We also made a few additional updates to cite some papers that we were made aware of after the original submission, as well as to better discuss end-to-end training of two-step models. For the convenience of the reviewers, we have highlighted all the updated parts of the manuscript in blue.

---

### Decision · Action_Editor_QwnN · 2024-07-25

**Recommendation:** Accept as is

**Comment:**

The paper is well-written, and all reviewers agree that it is valuable for researchers interested in understanding how the manifold hypothesis is linked to existing deep generative models.

**Audience:**

yes

**Claims And Evidence:**

yes